# Identifiability of Deep Polynomial Neural Networks

**Konstantin Usevich**[*]**, Ricardo Borsoi, Clara Dérand, Marianne Clausel**
Université de Lorraine, CNRS, CRAN
Nancy, F-54000, France
`firstname.lastname@univ-lorraine.fr`[†]

## Abstract

Polynomial Neural Networks (PNNs) possess a rich algebraic and geometric structure. However, their identifiability—a key property for ensuring interpretability—remains poorly understood. In this work, we present a comprehensive analysis of the identifiability of deep PNNs, including architectures with and without bias terms. Our results reveal an intricate interplay between activation degrees and layer widths in achieving identifiability. As special cases, we show that architectures with non-increasing layer widths are generically identifiable under mild conditions, while encoder-decoder networks are identifiable when the decoder widths do not grow too rapidly compared to the activation degrees. Our proofs are constructive and center on a connection between deep PNNs and low-rank tensor decompositions, and Kruskal-type uniqueness theorems. We also settle an open conjecture on the dimension of PNN's *neurovarieties*, and provide new bounds on the activation degrees required for it to reach the expected dimension.

## 1 Introduction

Neural network architectures which use polynomials as activation functions—*polynomial neural networks* (PNN)—have emerged as architectures that combine competitive experimental performance (capturing high-order interactions between input features) while allowing a fine grained theoretical analysis. On the one hand, PNNs have been employed in many problems in computer vision [1–3], image representation [4], physics [5] and finance [6], to name a few. On the other hand, the geometry of function spaces associated with PNNs, called *neuromanifolds*, can be analyzed using tools from algebraic geometry. Properties of such spaces, such as their dimension, shed light on the impact of a PNN architecture (layer widths and activation degrees) on the *expressivity* of feedforward, convolutional and self-attention PNN architectures [7–11]. They also determine the landscape of their loss function and the dynamics of their training process [7, 12, 13].

Moreover, PNNs are also closely linked to low-rank tensor decompositions [14–18], which play a fundamental role in the study of latent variable models due to their *identifiability* properties [19]. In fact, single-output 2-layer PNNs are equivalent to low-rank symmetric tensors [7]. Identifiability—whether the parameters and, consequently, the hidden representations of a NN can be determined from its response up to some equivalence class of trivial ambiguities such as permutations of its neurons—is a key question in NN theory [20–32]. Identifiability is critical to ensure interpretability in representation learning [33–35], to provably obtain disentangled representations [36], and in the study of causal models [37]. It is also critical to understand how the architecture affects the inference process and to support manipulation or "stitching" of pretrained models and representations [35, 38, 39]. Moreover, it has important links to learning and optimization of PNNs [40, 9, 13].

---

[*]Corresponding author

[†]In this version, the appendices have been reworked for better readability. Appendix E explains the changes between the submitted and the camera-ready version.

39th Conference on Neural Information Processing Systems (NeurIPS 2025).

Identifiability of deep PNNs is intimately linked to the dimension of their so-called *neurovarieties*: when this dimension reaches the effective parameter count, the number of possible parametrizations is finite, which means the model is *finitely identifiable* and the neurovariety is said to be *non-defective*. In addition, many PNN architectures admit only a single parametrization (i.e., they are *globally identifiable*).This has been investigated for specific types of self-attention [9] and convolutional [8] layers, and feedforward PNNs without bias [11]. However, current results for feedforward networks only show that finite identifiability holds for very high activation degrees, or for networks with the same widths in every layer [11]. A standing conjecture is that this holds for any PNN with degrees at least quadratic and non-increasing layer widths [11], which parallels identifiability results of ReLU networks [29]. However, a general theory of identifiability of deep PNNs is still missing.

## 1.1 Our contribution

We provide a comprehensive analysis of the identifiability of deep PNNs considering monomial activation functions. We prove that an $L$-layer PNN is finitely identifiable if every 2-layer block composed by a pair of two successive layers is finitely identifiable for some subset of their inputs. This surprising result tightly links the identifiability of shallow and deep polynomial networks, which is a key challenge in the general theory of NNs. Moreover, our results reveal an intricate interplay between activation degrees and layer widths in achieving identifiability.

As special cases, we show that architectures with non-increasing layer widths (i.e., pyramidal nets) are generically identifiable, while encoder-decoder (bottleneck) networks are identifiable when the decoder widths do not grow too rapidly compared to the activation degrees. We also show that the minimal activation degrees required to render a PNN identifiable (which is equivalent to its *activation thresholds*) is only *linear* in the layer widths, compared to the quadratic bound in [11, Theorem 18]. These results not only settle but generalize conjectures stated in [11]. Moreover, we also address the case of PNNs with biases (which was overlooked in previous theoretical studies) by leveraging a *homogenization* procedure.

Our proofs are constructive and are based on a connection between deep PNNs and partially symmetric canonical polyadic tensor decompositions (CPD). This allows us to leverage Kruskal-type uniqueness theorems for tensors to obtain identifiability results for 2-layer networks, which serve as the building block in the proof of the finite identifiability of deep nets, which is performed by induction. Our results also shed light on the geometry of the *neurovarieties*, as they lead to conditions under which its dimension reaches the expected (maximum) value.

## 1.2 Related works

**Polynomial NNs:** Several works studied PNNs from the lens of algebraic geometry using their associated *neuromanifolds* and *neurovarieties* [7] (in the emerging field of *neuroalgebraic geometry* [41]) and their close connection to tensor decompositions. Kileel et al. [7] studied the expressivity or feedforward PNNs in terms of the dimension of their neurovarieties. An analysis of the neuromanifolds for several architectures was presented in [10]. Conditions under which training losses do not exhibit bad local minima or spurious valleys were also investigated [13, 12, 42]. The links between training 2-layer PNNs and low-rank tensor approximation [13] as well as the biases of gradient descent [43] have been established.

Recent work computed the dimensions of neuromanifolds associated with special types of self-attention [9] and convolutional [8] architectures, and also include identifiability results. For feedforward PNNs, finite identifiability was demonstrated for networks with the same widths in every layer [11], while stronger results are available for the 2-layer case with more general polynomial activations [44]. Finite identifiability also holds when the activation degrees are larger than a so-called *activation degree threshold* [11]. Recent work studied the singularities of PNNs with activations consisting of the sum of monomials with very high activation degrees [45]. PNNs are also linked to *factorization machines* [46]; this led to the development of efficient tensor-based learning algorithms [47, 48]. Note that other types of non-monomial polynomial-type activations [49, 50, 5, 51] have shown excellent performance; however, the geometry of these models is not well known.

**NN identifiability:** Many studies focused on the identifiability of 2-layer NNs with tanh, odd, and ReLU activation functions [20–23]. Moreover, algorithms to learn 2-layer NNs with unique parameter recovery guarantees have been proposed (see, e.g., [52, 53]), however, their extension to NNs with 3 or more layers is challenging and currently uses heuristics [54]. Identifiability of deep NNs under

weak genericity assumptions was first studied in the pioneering work of Fefferman [24] for the case of the tanh activation function through the study of its singularities. Recent work extended this result to more general sigmoidal activations [25, 26]. Various works focused on deep ReLU nets, which are piecewise linear [28]; they have been shown to be generically identifiable if the number of neurons per layer is non-increasing [29]. Recent work studied the local identifiability of ReLU nets [30–32]. Identifiability has also been studied for latent variable/causal modeling, leveraging different types of assumptions (e.g., sparsity, statistical independence, etc.) [55–60]. Note that although some of these works tackle deep NNs, their proof techniques are completely different from our approach and do not apply to the case of polynomial activation functions.

**Tensors and NNs:** Low-rank tensor decompositions had widespread practical impact in the compression of NN weights [61–65]. Moreover, their properties also played a key role in the theory of NNs [18]. This includes the study of the expressivity of convolutional [66] and recurrent [67, 68] NNs, and the sample complexity of reinforcement learning parametrized by low-rank transition and reward tensors [69, 70]. The decomposability of low-rank symmetric tensors was also paramount in establishing conditions under which 2-layer NNs can (or cannot [71]) be learned in polynomial time and in the development of algorithms with identifiability guarantees [52, 72, 73]. It was also used to study identifiability of some deep *linear* networks [74]. However, the use of tensor decompositions in the studying the identifiability of deep *nonlinear* networks has not yet been investigated.

## 2 Setup and background

### 2.1 Polynomial neural networks: with and without bias

Polynomial neural networks are functions $\mathbb{R}^{d_0} \to \mathbb{R}^{d_L}$ represented as feedforward networks with bias terms and activation functions of the form $\rho_r(\cdot) = (\cdot)^r$. Our results hold for both the real and complex valued case ($\mathbb{F} = \mathbb{R}, \mathbb{C}$), thus, and we prefer to keep the real notation for simplicity. Note that we allow the activation functions to have a different degree $r_\ell$ for each layer.

**Definition 1** (PNN). *A **polynomial neural network (PNN) with biases** and architecture ($\boldsymbol{d} = (d_0, d_1, \ldots, d_L), \boldsymbol{r} = (r_1, \ldots, r_{L-1})$) is a map $\mathbb{R}^{d_0} \to \mathbb{R}^{d_L}$ given by a feedforward neural network*

$$\mathrm{PNN}_{\boldsymbol{d}, \boldsymbol{r}}[\boldsymbol{\theta}] = \mathrm{PNN}_{\boldsymbol{r}}[\boldsymbol{\theta}] := f_L \circ \rho_{r_{L-1}} \circ f_{L-1} \circ \rho_{r_{L-2}} \circ \cdots \circ \rho_{r_1} \circ f_1 \,, \qquad (1)$$

*where $f_i(\boldsymbol{x}) = \boldsymbol{W}_i \boldsymbol{x} + \boldsymbol{b}_i$ are affine maps, with $\boldsymbol{W}_i \in \mathbb{R}^{d_i \times d_{i-1}}$ being the weight matrices and $\boldsymbol{b}_i \in \mathbb{R}^{d_i}$ the biases, and the activation functions $\rho_r : \mathbb{R}^d \to \mathbb{R}^d$, defined as $\rho_r(\boldsymbol{z}) := (z_1^r, \ldots, z_d^r)$ are monomial. The parameters $\boldsymbol{\theta}$ are given by the entries of the weights $\boldsymbol{W}_i$ and biases $\boldsymbol{b}_i$, i.e.,*

$$\boldsymbol{\theta} = (\boldsymbol{w}, \boldsymbol{b}), \ \boldsymbol{w} = (\boldsymbol{W}_1, \boldsymbol{W}_2, \ldots, \boldsymbol{W}_L), \ \boldsymbol{b} = (\boldsymbol{b}_1, \boldsymbol{b}_2, \ldots, \boldsymbol{b}_L). \qquad (2)$$

*The vector of degrees $\boldsymbol{r}$ is called the* activation degree *of $\mathrm{PNN}_{\boldsymbol{r}}[\boldsymbol{\theta}]$ (we often omit the subscript $\boldsymbol{d}$ if it is clear from the context).*

PNNs are algebraic maps and are polynomial vectors, where the *total degree* is $r_{total} = r_1 \cdots r_{L-1}$, that is, they belong to the polynomial space $(\mathscr{P}_{d, r_{total}})^{\times d_L}$, where $\mathscr{P}_{d, r}$ denotes the space of $d$-variate polynomials of degree $\leq r$. Most previous works analyzed the simpler case of PNNs without bias, which we refer to as *homogeneous*. Due to its importance, we consider it explicitly.

**Definition 2** (hPNN). *A PNN is said to be a **homogenous** PNN (hPNN) when it has no biases ($\boldsymbol{b}_\ell = 0$ for all $\ell = 1, \ldots, L$), and is denoted as*

$$\mathrm{hPNN}_{\boldsymbol{d}, \boldsymbol{r}}[\boldsymbol{w}] = \mathrm{hPNN}_{\boldsymbol{r}}[\boldsymbol{w}] := \boldsymbol{W}_L \circ \rho_{r_{L-1}} \circ \boldsymbol{W}_{L-1} \circ \rho_{r_{L-2}} \circ \cdots \circ \rho_{r_1} \circ \boldsymbol{W}_1. \qquad (3)$$

*Its parameter set is given by $\boldsymbol{w} = (\boldsymbol{W}_1, \boldsymbol{W}_2, \ldots, \boldsymbol{W}_L)$.*

It is well known that such PNNs are in fact homogeneous polynomial vectors and belong to the polynomial space $(\mathscr{H}_{d_0, r_{total}})^{\times d_L}$, where $\mathscr{H}_{d, r} \subset \mathscr{P}_{d, r}$ denotes the space of homogeneous $d$-variate polynomials of degree $r$. hPNNs are also naturally linked to tensors and tensor decompositions, whose properties can be used in their theoretical analysis.

**Example 3** (Running example). *Consider an hPNN with $L = 2$, $\boldsymbol{r} = (2)$ and $\boldsymbol{d} = (3, 2, 2)$. In such a case the parameter matrices are given as*

$$\boldsymbol{W}_2 = \begin{bmatrix} b_{11} & b_{12} \\ b_{21} & b_{22} \end{bmatrix}, \quad \boldsymbol{W}_1 = \begin{bmatrix} a_{11} & a_{12} & a_{13} \\ a_{21} & a_{22} & a_{23} \end{bmatrix},$$

*and the hPNN $\boldsymbol{p} = \mathrm{hPNN}_{\boldsymbol{r}}[\boldsymbol{w}]$ is a vector polynomial that admits the expression*

$$\boldsymbol{p}(\boldsymbol{x}) = \boldsymbol{W}_2 \rho_2(\boldsymbol{W}_1 \boldsymbol{x}) = \begin{bmatrix} b_{11} \\ b_{21} \end{bmatrix} (a_{11}x_1 + a_{12}x_2 + a_{13}x_3)^2 + \begin{bmatrix} b_{12} \\ b_{22} \end{bmatrix} (a_{21}x_1 + a_{22}x_2 + a_{23}x_3)^2.$$

*the only monomials that can appear are of the form $x_1^i x_2^j x_3^k$ with $i + j + k = 2$ thus $\boldsymbol{p}$ is a vector of degree-2 homogeneous polynomials in 3 variables (in our notation, $\boldsymbol{p} \in (\mathscr{H}_{3,2})^2$).*

## 2.2 Equivalent PNN representations

It is known that the PNNs admit equivalent representations (i.e., several parameters $\boldsymbol{\theta}$ leading to the same function). Indeed, for each hidden layer we can (a) permute the hidden neurons, and (b) rescale the input and output to each activation function since for any $a \neq 0$, $(at)^r = a^r t^r$. These transformations lead to different sets of parameters that leave the PNN unchanged. We can characterize all such equivalent representations in the following lemma (provided in [7] for the case without biases).

**Lemma 4.** *Let $\mathrm{PNN}_{\boldsymbol{d},\boldsymbol{r}}[\boldsymbol{\theta}]$ be a PNN with $\boldsymbol{\theta}$ as in (2). Let also $\boldsymbol{D}_\ell \in \mathbb{R}^{d_\ell \times d_\ell}$ be any invertible diagonal matrices and $\boldsymbol{P}_\ell \in \mathbb{Z}^{d_\ell \times d_\ell}$ ($\ell = 1, \ldots, L-1$) be permutation matrices, and define the transformed parameters as*

$$\boldsymbol{W}'_\ell \leftarrow \boldsymbol{P}_\ell \boldsymbol{D}_\ell \boldsymbol{W}_\ell \boldsymbol{D}_{\ell-1}^{-r_{\ell-1}} \boldsymbol{P}_{\ell-1}^\mathsf{T}, \qquad \boldsymbol{b}'_\ell \leftarrow \boldsymbol{P}_\ell \boldsymbol{D}_\ell \boldsymbol{b}_\ell,$$

*with $\boldsymbol{P}_0 = \boldsymbol{D}_0 = \boldsymbol{I}$ and $\boldsymbol{P}_L = \boldsymbol{D}_L = \boldsymbol{I}$ by convention. Then the modified parameters $\boldsymbol{W}'_\ell, \boldsymbol{b}'_\ell$ define exactly the same network, i.e. $\mathrm{PNN}_{\boldsymbol{d},\boldsymbol{r}}[\boldsymbol{\theta}] = \mathrm{PNN}_{\boldsymbol{d},\boldsymbol{r}}[\boldsymbol{\theta}']$ for the parameter vector*

$$\boldsymbol{\theta}' = ((\boldsymbol{W}'_1, \boldsymbol{W}'_2, \ldots, \boldsymbol{W}'_L), (\boldsymbol{b}'_1, \boldsymbol{b}'_2, \ldots, \boldsymbol{b}'_L)).$$

*If $\boldsymbol{\theta}$ and $\boldsymbol{\theta}'$ are linked with such a transformation, they are called equivalent (denoted $\boldsymbol{\theta} \sim \boldsymbol{\theta}'$).*

**Example 5** (Example 3, continued)**.** *In Example 3 we can take any $\alpha$, $\beta \neq 0$ to get*

$$\mathrm{hPNN}_{\boldsymbol{d},\boldsymbol{r}}[\boldsymbol{w}] = \begin{bmatrix} \alpha^{-2}b_{11} \\ \alpha^{-2}b_{21} \end{bmatrix} (\alpha a_{11}x_1 + \alpha a_{12}x_2 + \alpha a_{13}x_3)^2 + \begin{bmatrix} \beta^{-2}b_{12} \\ \beta^{-2}b_{22} \end{bmatrix} (\beta a_{21}x_1 + \beta a_{22}x_2 + \beta a_{23}x_3)^2.$$

*which correspond to rescaling rows of $\boldsymbol{W}_1$ and corresponding columns of $\boldsymbol{W}_2$. If we additionally permute them, we get $\boldsymbol{W}'_1 = \boldsymbol{P}\boldsymbol{D}\boldsymbol{W}_1$, $\boldsymbol{W}'_2 = \boldsymbol{W}_2\boldsymbol{D}^{-2}\boldsymbol{P}^\mathsf{T}$ with $\boldsymbol{D} = \begin{bmatrix} \alpha & 0 \\ 0 & \beta \end{bmatrix}$ and $\boldsymbol{P} = \begin{bmatrix} 0 & 1 \\ 1 & 0 \end{bmatrix}$.*

This characterization of equivalent representations allows us to define when a PNN is *unique*.

**Definition 6** (Unique and finite-to-one representation)**.** *The PNN $\boldsymbol{p} = \mathrm{PNN}_{\boldsymbol{d},\boldsymbol{r}}[\boldsymbol{\theta}]$ (resp. hPNN $\boldsymbol{p} = \mathrm{hPNN}_{\boldsymbol{d},\boldsymbol{r}}[\boldsymbol{w}]$) with parameters $\boldsymbol{\theta}$ (resp. $\boldsymbol{w}$) is said have a **unique** representation if every other representation satisfying $\boldsymbol{p} = \mathrm{PNN}_{\boldsymbol{d},\boldsymbol{r}}[\boldsymbol{\theta}']$ (resp. $\boldsymbol{p} = \mathrm{hPNN}_{\boldsymbol{d},\boldsymbol{r}}[\boldsymbol{w}']$) is given by an equivalent set of parameters, i.e., $\boldsymbol{\theta}' \sim \boldsymbol{\theta}$ (resp. $\boldsymbol{w}' \sim \boldsymbol{w}$) in the sense of Lemma 4 (i.e., they can be obtained from the permutations and elementwise scalings in Lemma 4).*

*Similarly, a PNN $\boldsymbol{p} = \mathrm{PNN}_{\boldsymbol{d},\boldsymbol{r}}[\boldsymbol{\theta}]$ (resp. hPNN $\boldsymbol{p} = \mathrm{hPNN}_{\boldsymbol{d},\boldsymbol{r}}[\boldsymbol{w}]$) is called **finite-to-one** if it admits only finitely many non-equivalent representations, that is, the set $\{\boldsymbol{\theta}' : \mathrm{PNN}_{\boldsymbol{d},\boldsymbol{r}}[\boldsymbol{\theta}'] = \boldsymbol{p}\}$ (resp. $\{\boldsymbol{w}' : \mathrm{hPNN}_{\boldsymbol{d},\boldsymbol{r}}[\boldsymbol{w}'] = \boldsymbol{p}\}$) contains finitely many non-equivalent parameters.*

**Example 7** (Example 5, continued)**.** *Thanks to links with tensor decompositions and their uniqueness, it is known that the hPNN in Example 3 has unique representation if $\boldsymbol{W}_2$ is invertible and $\boldsymbol{W}_1$ full row rank (rank 2), see Proposition 35 in Section 4.2.*

## 2.3 Identifiability and link to neurovarieties

An immediate question is *which PNN/hPNN architectures are expected to admit only a single (or finitely many) non-equivalent representations?* This question can be formalized using the notions of **global** and **finite identifiability**, which considers a general set of parameters.

**Definition 8** (Global and finite identifiability)**.** *The PNN (resp. hPNN) with architecture $(\boldsymbol{d}, \boldsymbol{r})$ is said to be **globally identifiable** if for a general choice of $\boldsymbol{\theta} = (\boldsymbol{w}, \boldsymbol{b}) \in \mathbb{R}^{\sum d_\ell(d_{\ell-1}+1)}$, (resp. $\boldsymbol{w} \in \mathbb{R}^{\sum d_\ell d_{\ell-1}}$) (i.e., for all choices of parameters except for a set of Lebesgue measure zero), the network $\mathrm{PNN}_{\boldsymbol{d},\boldsymbol{r}}[\boldsymbol{\theta}]$ (resp. $\mathrm{hPNN}_{\boldsymbol{d},\boldsymbol{r}}[\boldsymbol{w}]$) has a unique representation.*

*Similarly, the PNN (resp. hPNN) with architecture $(\boldsymbol{d}, \boldsymbol{r})$ is said to be **finitely identifiable** if for a general choice of $\boldsymbol{\theta}$, (resp. $\boldsymbol{w}$) the network $\mathrm{PNN}_{\boldsymbol{d},\boldsymbol{r}}[\boldsymbol{\theta}]$ (resp. $\mathrm{hPNN}_{\boldsymbol{d},\boldsymbol{r}}[\boldsymbol{w}]$) is finite-to-one (i.e., it admits only finitely many non-equivalent representations).*

In the following, we use the term "identifiable" to refer to finite identifiability unless stated otherwise. Note also that the notion of finite identifiability is much stronger than the related notion of local identifiability (i.e., a model being identifiable only in a neighborhood of a parameterization).

**Example 9** (Example 7, continued). *From Example 7, we see that the hPNN architecture with $\boldsymbol{d} = (3, 2, 2)$, $\boldsymbol{r} = (2)$ is identifiable due to the fact that generic matrices $\boldsymbol{W}_1$ and $\boldsymbol{W}_2$ are full rank.*

Note that Definition 8 excludes a set of parameters of Lebesgue measure zero. Thus, for an identifiable architecture such as the one mentioned in Example 9, there exists rare sets of pathological parameters for which the hPNN is non-unique (e.g., weight matrices containing collinear rows).

With some abuse of notation, let $\mathrm{hPNN}_{\boldsymbol{d,r}}[\cdot]$ be the map taking $\boldsymbol{w}$ to $\mathrm{hPNN}_{\boldsymbol{d,r}}[\boldsymbol{w}]$. Then the image of $\mathrm{hPNN}_{\boldsymbol{d,r}}[\cdot]$ is called a *neuromanifold*, and the *neurovariety* $\mathscr{V}_{\boldsymbol{d,r}}$ is defined as its closure in the Zariski topology[3]. The study of neurovarieties and their properties is a topic of recent interest [7, 41, 11, 10]. More details are given in Appendix A. An important property for our case is the link between identifiability of an hPNN, the dimension of its neurovariety, and the rank of its Jacobian.

**Proposition 10.** *The architecture $\mathrm{hPNN}_{\boldsymbol{d,r}}[\cdot]$ is finitely identifiable if and only if the dimension of $\mathscr{V}_{\boldsymbol{d,r}}$ is equal to the effective number of parameters, i.e., $\dim \mathscr{V}_{\boldsymbol{d,r}} = \sum_{\ell=1}^{L} d_\ell d_{\ell-1} - \sum_{\ell=1}^{L-1} d_\ell$. In such case, $\mathscr{V}_{\boldsymbol{d,r}}$ is said to be **nondefective**. Equivalently, the rank of the Jacobian of the map $\mathrm{hPNN}_{\boldsymbol{d,r}}[\cdot]$ is maximal and equal to $\sum_{\ell=1}^{L} d_\ell d_{\ell-1} - \sum_{\ell=1}^{L-1} d_\ell$ at a general parameter $\boldsymbol{w}$.*

## 3 Main results

### 3.1 Main results on the identifiability of deep hPNNs

Although several works have studied the identifiability of 2-layer NNs, tackling the case of deep networks is significantly harder. However, when we consider the opposite statement, i.e., the *non-identifiability* of a network, it is much easier to show such connection: in a deep network with $L > 2$ layers, the lack of identifiability of any 2-layer subnetwork (formed by two consecutive layers) clearly implies that the full network is not identifiable. What our main result shows is that, surprisingly, under mild additional conditions the converse is also true for hPNNs: if the every 2-layer subnetwork is identifiable for some subset of their inputs, then the full network is identifiable as well. This is formalized in the following theorem.

**Theorem 11** (Localization theorem). *Let $((d_0, \ldots, d_L), (r_1, \ldots, r_{L-1}))$ be the hPNN format. For $\ell = 0, \ldots, L - 2$ denote $\widetilde{d}_\ell := \min\{d_0, \ldots, d_\ell\}$. Then the following holds true: if for all $\ell = 1, \ldots, L - 1$ the two-layer architecture $\mathrm{hPNN}_{(\widetilde{d}_{\ell-1}, d_\ell, d_{\ell+1}), r_\ell}[\cdot]$ is finitely identifiable, then the $L$-layer architecture $\mathrm{hPNN}_{\boldsymbol{d,r}}[\cdot]$ is finitely identifiable as well.*

The technical proofs are relegated to the appendices. This key result shows a strong relation between the finite identifiability of shallow and deep hPNNs. However, as we move into the deeper layers, the identifiability conditions required by Theorem 11 are stricter than in the shallow case, since the number of inputs is reduced to $\widetilde{d}_\ell$. This can lead to a requirement of larger activation degrees to guarantee identifiability compared to the shallow case.

Theorem 11 allows us to derive identifiability conditions for hPNNs using the link between 2-layer hPNNs and partially symmetric tensor decompositions and their generic uniqueness based on classical Kruskal-type conditions. We use the following sufficient condition for the identifiability of shallow networks.

**Proposition 12** (Sufficient condition for identifiability of 2-layer hPNN). *Let $d_0, d_1 \geq 2$, $d_2 \geq 1$ be the layer widths and $r \geq 2$ such that*

$$r \geq \frac{2d_1 - \min(d_2, d_1)}{\min(d_1, d_0) - 1}. \tag{4}$$

*Then the 2-layer hPNN with architecture $((d_0, d_1, d_2), r)$ is globally identifiable.*

**Remark 13.** *If the above condition is satisfied for every 2-layer architecture $((\widetilde{d}_{\ell-1}, d_\ell, d_{\ell+1}), r_\ell)$, $\ell = 1, \ldots, L - 1$, then Theorem 11 implies that the $L$-layer hPNN is finitely identifiable for the $L$-layer architecture $(\boldsymbol{d}, \boldsymbol{r})$.*

---

[3]i.e., the smallest algebraic variety that contains the image of the map $\mathrm{hPNN}_{\boldsymbol{d,r}}[\cdot]$.

**Remark 14.** *Note that for the single output case $d_L = 1$, Equation (4) means the activation degree in the last layer must satisfy $r_{L-1} \geq 3$, in contrast to $r_\ell \geq 2$ for $\ell < L - 1$.*

**Remark 15** (Our bounds are constructive). *We note that the condition (4) for identifiability is not the best possible (and can be further improved using much stronger results on generic uniqueness of decompositions, see e.g., [75, Corollary 37]). However, the bound (4) is constructive, and we can use standard polynomial-time tensor algorithms to recover the parameters of the 2-layer hPNN.*

### 3.2 Implications for specific architectures

Proposition 12 has direct implications for the finite identifiability of several architectures of practical interest, including pyramidal and bottleneck networks, and for the activation thresholds of hPNNs, as shown in the following corollaries.

**Corollary 16** (Pyramidal hPNNs are always identifiable). *The hPNNs with architectures containing non-increasing layer widths (except possibly the last layer), i.e., $d_0 \geq d_1 \geq \cdots d_{L-1} \geq 2$ and $d_L \geq 1$, are finitely identifiable for any degrees satisfying*

$$\text{(i)} \ \ r_1, \ldots, r_{L-1} \geq 2 \ \ \text{if} \ \ d_L \geq 2; \quad \text{or} \quad \text{(ii)} \ \ r_1, \ldots, r_{L-2} \geq 2, \ r_{L-1} \geq 3 \ \ \text{if} \ \ d_L = 1.$$

Note that, due to the connection between the identifiability of hPNNs and the neurovarieties presented in Proposition 10, a direct consequence of Corollary 16 is that the neurovariety $\mathscr{V}_{\boldsymbol{d}, \boldsymbol{r}}$ has expected dimension. This settles a recent conjecture presented in [11, Section 4]. This implication is explained in detail in Appendix A.

Instead of seeking conditions on the layer widths for a fixed (or minimal) degree, a complementary perspective is to determine what are the smallest degrees $r_\ell$ such that a given architecture $\boldsymbol{d}$ is finitely identifiable. Following the terminology introduced in [11], we refer to those values as the *activation degree thresholds for identifiability* of an hPNN. An upper bound is given in the following corollary:

**Corollary 17** (Activation degree thresholds for identifiability). *For fixed layer widths $\boldsymbol{d} = (d_0, \ldots, d_L)$ with $d_\ell \geq 2$, $\ell = 0, \ldots, L - 1$, the hPNNs with architectures $(\boldsymbol{d}, (r_1, \ldots, r_{L-1}))$ are finitely identifiable for any degrees satisfying*

$$r_\ell \geq 2d_\ell - 1 \,.$$

Note that due to Proposition 10, the result in this corollary implies that the neurovariety $\mathscr{V}_{\boldsymbol{d}, \boldsymbol{r}}$ has expected dimension. This means that $(2d_\ell - 1)$ is also a universal upper bound to the so-called *activation thresholds* for hPNN expressiveness introduced in [11]. The existence of such activation degree thresholds was conjectured in [7] and recently proved in [11, Theorem 18], but the for a *quadratic* in $d_\ell$ bound (the bound in Corollary 17 is *linear*).

**Remark 18** (Admissible layer sizes). *The possible layer sizes in a deep network are tightly linked with the degree of the activation. For example, for $r_\ell = 2$, identifiability is impossible if $d_\ell > \frac{d_{\ell-1}(d_{\ell-1}+1)}{2}$ (for general $r_\ell$, a similar bound $O(d_{\ell-1}^{r_\ell})$ follows from a link with tensor decompositions [76]). Therefore, to allow for larger layer widths, we need to have higher-degree activations.*

It is enlightening to consider the admissible layer widths when taking into account the joint effect of layer widths and degrees. By doing this, Proposition 12 can be leveraged to yield identifiability conditions for the case of bottleneck networks, as illustrated in the following corollary.

**Corollary 19** (Identifiability of bottleneck hPNNs). *Consider the "bottleneck" architecture with*

$$d_0 \geq d_1 \geq \cdots \geq \ d_b \ \leq d_{b+1} \leq \ldots \leq d_L$$

*and $d_b \geq 2$. Suppose that $r_1, \ldots, r_b \geq 2$ and that the decoder part satisfies $\frac{d_\ell}{r_\ell} \leq d_b - 1$ for $\ell \in \{b + 1, \ldots, L - 1\}$. Then the bottleneck hPNN is finitely identifiable.*

This shows that encoder-decoder hPNNs architectures are identifiable under mild conditions on the layer widths and decoder degrees, providing a polynomial networks-based counterpart to previous studies that analyzed linear autoencoders [77, 78].

Note that the width of the bottleneck layer $d_b$ constrains the entire decoder part of the architecture: the degrees $r_\ell$, $\ell \geq b$ are constrained according to the width $d_b$. The presence of bottlenecks has also been shown to affect the expressivity of hPNNs in [7, Theorem 19]: for $d_b = 2d_0 - 2$ there exists a number of layers $L$ such that for $r_\ell \geq 2$ and $d_0 \geq 2$, the hPNN neurovariety is *non-filling* (i.e., its dimension never reaches that of the ambient space) for any choice of widths $d_1, \ldots, d_{b-1}, d_{b+1}, \ldots, d_L$.

### 3.3 PNNs with biases

The identifiability of general PNNs (with biases) can be studied via the properties of hPNNs. The simplest idea is *truncation* (i.e., taking only higher-order terms of the polynomials), which eliminates biases from PNNs. Such an approach was already taken in [44] for shallow PNNs with general polynomial activation, and is described in Appendix D.3. We will follow a different approach based on the well-known idea of **homogenization**: we transform a PNN to an equivalent hPNN with structured parameters keeping the information about biases at the expense of increasing the layer widths. Our key result is to show how this can be used to study the identifiability of PNNs with bias terms. The following correspondence is well-known.

**Definition 20** (Homogenization). *There is a one-to-one mapping between polynomials in $d$ variables of degree $r$ and homogeneous polynomials of the same degree in $d + 1$ variables. We denote this mapping $\mathscr{P}_{d,r} \to \mathscr{H}_{d+1,r}$ by $\mathrm{homog}(\cdot)$, and it acts as follows: for every polynomial $p \in \mathscr{P}_{d,r}$, $\widetilde{p} = \mathrm{homog}(p) \in \mathscr{H}_{d+1,r}$ (that is $\widetilde{p}(x_1, \ldots, x_d, x_{d+1})$) is the unique homogeneous polynomial in $d + 1$ variables such that*

$$\widetilde{p}(x_1, \ldots, x_d, 1) = p(x_1, \ldots, x_d).$$

**Example 21.** *For the polynomial $p \in \mathscr{P}_{2,2}$ in variables $(x_1, x_2)$ given by*

$$p(x_1, x_2) = ax_1^2 + bx_1x_2 + cx_2^2 + ex_1 + fx_2 + g,$$

*its homogenization $\widetilde{p} = \mathrm{homog}(p) \in \mathscr{H}_{3,2}$ in 3 variables $(x_1, x_2, x_3)$ is*

$$\widetilde{p}(x_1, x_2, x_3) = ax_1^2 + bx_1x_2 + cx_2^2 + ex_1x_3 + fx_2x_3 + gx_3^2,$$

*and we can verify that $\widetilde{p}(1, x_1, x_2) = p(x_1, x_2)$.*

Similarly, we extend homogenization to polynomial vectors, which gives the following.

**Example 22.** *Let $f(\boldsymbol{x}) = \boldsymbol{W}_2\rho_{r_1}(\boldsymbol{W}_1\boldsymbol{x} + \boldsymbol{b}_1) + \boldsymbol{b}_2$, and define extended matrices as*

$$\widetilde{\boldsymbol{W}}_1 = \begin{bmatrix} \boldsymbol{W}_1 & \boldsymbol{b}_1 \\ 0 & 1 \end{bmatrix} \in \mathbb{R}^{(d_1+1)\times(d_0+1)}, \quad \widetilde{\boldsymbol{W}}_2 = [\boldsymbol{W}_2 \quad \boldsymbol{b}_2] \in \mathbb{R}^{d_2\times(d_1+1)}$$

*Then its homogenization $\widetilde{f} = \mathrm{homog}(f)$ is an hPNN of format $(d_0 + 1, d_1 + 1, d_2)$*

$$\widetilde{f}(\widetilde{\boldsymbol{x}}) = \widetilde{\boldsymbol{W}}_2\rho_{r_1}\left(\widetilde{\boldsymbol{W}}_1\widetilde{\boldsymbol{x}}\right)$$

*where $\widetilde{\boldsymbol{x}} = [x_0, x_1, \ldots, x_{d_0}, x_{d_0+1}]^\mathsf{T}$, so that $\widetilde{f}(x_1, \ldots, x_{d_0}, 1) = f(x_1, \ldots, x_{d_0})$.*

The construction in Example 22 similar to the well-known idea of augmenting the network with an artificial (constant) input. The following proposition generalizes this example to the case of multiple layers, by "propagating" the constant input.

**Proposition 23.** *Fix the architecture $\boldsymbol{r} = (r_1, \ldots, r_{L-1})$ and $\boldsymbol{d} = (d_0, \ldots, d_L)$. Then a polynomial vector $\boldsymbol{p} \in (\mathscr{P}_{d_0, r_{total}})^{\times d_L}$ admits a PNN representation $\boldsymbol{p} = \mathrm{PNN}_{\boldsymbol{d},\boldsymbol{r}}[(\boldsymbol{w}, \boldsymbol{b})]$ with $(\boldsymbol{w}, \boldsymbol{b})$ as in (2) if and only if its homogenization $\widetilde{\boldsymbol{p}} = \mathrm{homog}(\boldsymbol{p})$ admits an hPNN decomposition for the same activation degrees $\boldsymbol{r}$ and extended $\widetilde{\boldsymbol{d}} = (d_0 + 1, \ldots, d_{L-1} + 1, d_L)$, $\widetilde{\boldsymbol{p}} = \mathrm{hPNN}_{\widetilde{\boldsymbol{d}},\boldsymbol{r}}[\widetilde{\boldsymbol{w}}]$, $\widetilde{\boldsymbol{w}} = (\widetilde{\boldsymbol{W}}_1, \ldots, \widetilde{\boldsymbol{W}}_L)$, with matrices given as*

$$\widetilde{\boldsymbol{W}}_\ell = \begin{cases} \begin{bmatrix} \boldsymbol{W}_\ell & \boldsymbol{b}_\ell \\ 0 & 1 \end{bmatrix} \in \mathbb{R}^{(d_\ell+1)\times(d_{\ell-1}+1)}, & \ell < L, \\ \begin{bmatrix} \boldsymbol{W}_L & \boldsymbol{b}_L \end{bmatrix} \in \mathbb{R}^{(d_L)\times(d_{L-1}+1)}, & \ell = L. \end{cases}$$

That is, PNNs are in one-to-one correspondence to hPNNs with increased number of inputs and structured weight matrices.

**Uniqueness of PNNs from homogenization:** An important consequence of homogenization is that the uniqueness of the homogenized hPNN implies the uniqueness of the original PNN with bias terms, which is a key result to support the application of our identifiability results to general PNNs.

**Proposition 24.** *If $\mathrm{hPNN}_{\boldsymbol{r}}[\widetilde{\boldsymbol{w}}]$ from Proposition 23 is unique (resp. finite-to-one) as an hPNN (without taking into account the structure), then the original PNN representation $\mathrm{PNN}_{\boldsymbol{r}}[(\boldsymbol{w}, \boldsymbol{b})]$ is unique (resp. finite-to-one).*

The proposition follows from the fact that we can always fix the permutation ambiguity for the "artificial" input.

**Remark 25.** *Despite the one-to-one correspondence, for generic properties (e.g., finite identifiability) we cannot immediately apply the results from the homogeneous case, because the matrices $\widetilde{W}_\ell$ are structured (they form a set of measure zero inside $\mathbb{R}^{(d_\ell+1)\times(d_{\ell-1}+1)}$).*

However, we can prove that the identifiability of the hPNN implies the identifiability of the PNN.

**Lemma 26.** *Let the $2$-layer hPNN architecture be finitely (resp. globally) identifiable for $((d_0 + 1, d_1 + 1, d_2), r_1)$. Then the PNN architecture with widths $(d_0, d_1, d_2)$ and degree $r_1$ is also finitely (resp. globally) identifiable.*

Using Lemma 26 and specializing the proof of Theorem 11, we obtain the following result:

**Proposition 27.** *Let $((d_0,\ldots,d_L),(r_1,\ldots,r_{L-1}))$ be the PNN format. For $\ell = 0,\ldots,L-2$ denote $\widetilde{d}_\ell = \min\{d_0,\ldots,d_\ell\}$. Then the following holds true: If for all $\ell = 1,\ldots,L-1$ each two-layer architecture $\mathrm{hPNN}_{(\widetilde{d}_{\ell-1}+1,d_\ell+1,d_{\ell+1}),r_\ell}[\cdot]$ is finitely identifiable, then the $L$-layer PNN with architecture $(\boldsymbol{d},\boldsymbol{r})$ is finitely identifiable as well.*

In particular, we have the following bounds for generic uniqueness.

**Corollary 28.** *Let $((d_0,\ldots,d_L),(r_1,\ldots,r_{L-1}))$ be such that $d_\ell \geq 1$, and $r_\ell \geq 2$ satisfy*

$$r_\ell \geq \frac{2(d_\ell + 1) - \min(d_\ell + 1, d_{\ell+1})}{\min(d_\ell, \widetilde{d}_{\ell-1})},$$

*then the $L$-layer PNN with architecture $(\boldsymbol{d},\boldsymbol{r})$ is finitely identifiable (and globally identifiable if $L = 2$).*

**Remark 29.** *For general PNNs with bias, similar conclusions hold to the ones in the hPNN case. In particular, for fixed layer widths $d_\ell \geq 1$, the activation threshold for a PNN architecture $(\boldsymbol{d},\boldsymbol{r})$ becomes $r_\ell \geq 2d_\ell + 1$. Also, pyramidal PNNs are identifiable in degree $2$.*

*A distinctive feature of PNNs with bias is that they can be identifiable even for architectures with layers containing a single hidden neuron: for $d_\ell = 1$ and $d_{\ell+1} \geq 2$ and/or $\widetilde{d}_{\ell-1} = 1$, the condition in Corollary 28 is still satisfied when $r_\ell \geq 2$.*

## 4 Proofs and main tools

Our main results in Theorem 11 translates the identifiability conditions of deep hPNNs into those of shallow hPNNs. Our results are strongly related to the decomposition of partially symmetric tensors (we review basic facts about tensors and tensors decompositions and recall their connection between to hPNNs in later subsections). More details are provided in the appendices, and we list key components of the proof below.

### 4.1 Identifiability of deep PNNs: necessary conditions

**Increasing hidden layers breaks uniqueness.** The key insight is that if we add to any architecture a neuron in any hidden layer, then the uniqueness of the hPNN is not possible, which is formalized as following lemma (whose proof is based, in its turn, on tensor decompositions).

**Lemma 30.** *Let $\boldsymbol{p} = \mathrm{hPNN}_{\boldsymbol{r}}[\boldsymbol{w}]$ be an hPNN of format $(d_0,\ldots,d_\ell,\ldots,d_L)$. Then for any $\ell$ there exists an infinite number of representations of hPNNs $\boldsymbol{p} = \mathrm{hPNN}_{\boldsymbol{r}}[\boldsymbol{w}]$ with architecture $(d_0,\ldots,d_\ell + 1,\ldots,d_L)$. In particular, the augmented hPNN is not unique (and is not finite-to-one).*

**Internal features of a unique hPNN are linearly independent.** This is an easy consequence of Lemma 30 (as linear dependence would allow for pruning neurons).

**Lemma 31.** *For $\boldsymbol{d} = (d_0,\ldots,d_L)$, let $\boldsymbol{p} = \mathrm{hPNN}_{\boldsymbol{r}}[\boldsymbol{w}]$ have a unique (or finite-to-one) $L$-layers decomposition. Consider the output at any $\ell$-th internal level $\ell < L$ after the activations*

$$\boldsymbol{q}_\ell(\boldsymbol{x}) = \rho_{r_\ell} \circ \boldsymbol{W}_\ell \circ \cdots \circ \rho_{r_1} \circ \boldsymbol{W}_1(\boldsymbol{x}). \tag{5}$$

*Then the elements of $\boldsymbol{q}_\ell(\boldsymbol{x}) = [q_{\ell,1}(\boldsymbol{x}) \quad \cdots \quad q_{\ell,d_\ell}(\boldsymbol{x})]^{\mathsf{T}}$ are linearly independent polynomials.*

**Identifiability for hPNNs and Kruskal rank.** Identifiability of 2-layer hPNNs, or equivalently uniqueness of CPD is strongly related to the concept of Kruskal rank of a matrix that we define below.

**Definition 32.** *The Kruskal rank of a matrix $\boldsymbol{A}$ (denoted $\mathrm{krank}\{\boldsymbol{A}\}$) is the maximal number $k$ such that any $k$ columns of $\boldsymbol{A}$ are linearly independent.*

This is in contrast with the usual rank, which is the maximal $k$ *such that there exist* $k$ linearly independent columns. Therefore $\mathrm{krank}\{\boldsymbol{A}\} \leq \mathrm{rank}\{\boldsymbol{A}\}$. Note that $\mathrm{krank}\{\boldsymbol{A}\} \geq 2$ means that none of the pairs of columns of $\boldsymbol{A}$ are linearly dependent (no columns are pairwise collinear). Using the notion of Kruskal rank, we can state a necessary condition on weight matrices for identifiability of hPNNs, which is a generalization of the well-known necessary condition for the uniqueness of CPD tensor decompositions (6) (i.e., shallow networks), and is a corollary of Lemma 30 and Lemma 31.

**Proposition 33.** *As in Lemma 31, let the widths be $\boldsymbol{d} = (d_0, \ldots, d_L)$, and $\boldsymbol{p} = \mathrm{hPNN}_{\boldsymbol{r}}[\boldsymbol{w}]$ have a unique (or finite-to-one) L-layers decomposition. Then we have that for all $\ell = 1, \ldots, L-1$*

$$\mathrm{krank}\{\boldsymbol{W}_\ell^\mathsf{T}\} \geq 2, \quad \mathrm{krank}\{\boldsymbol{W}_{\ell+1}\} \geq 1,$$

*where $\mathrm{krank}\{\boldsymbol{W}_{\ell+1}\} \geq 1$ simply means that $\boldsymbol{W}_{\ell+1}$ does not have zero columns.*

## 4.2 Shallow hPNNs and tensor decompositions

An order-$s$ tensor $\boldsymbol{\mathcal{T}} \in \mathbb{R}^{m_1 \times \cdots \times m_s}$ is an $s$-way multidimensional array (more details are provided in Appendix B.2 and more background on tensors can be found in [14–16]). It is said to have a $d$-term CPD (canonical polyadic decomposition) if it admits a decomposition into $d$ rank-1 terms $\boldsymbol{\mathcal{T}} = \sum_{j=1}^d \boldsymbol{a}_{1,j} \otimes \cdots \otimes \boldsymbol{a}_{s,j}$ for $\boldsymbol{a}_{i,j} \in \mathbb{R}^{m_i}$, with $\otimes$ being the tensor (outer) product. The CPD is also written compactly as $\boldsymbol{\mathcal{T}} = [\![\boldsymbol{A}_1, \boldsymbol{A}_2, \cdots, \boldsymbol{A}_s]\!]$ for matrices $\boldsymbol{A}_i = [\boldsymbol{a}_{i,1}, \cdots, \boldsymbol{a}_{i,d}] \in \mathbb{R}^{m_i \times d}$. $\boldsymbol{\mathcal{T}}$ is said to be (partially) *symmetric* if it is invariant to any permutation of (a subset of) its indices [79]. Concretely, we will consider tensors $\boldsymbol{\mathcal{T}}$ partially symmetric on dimensions $i \in \{2, \ldots, s\}$, with CPD that is also partially symmetric, i.e., with $\boldsymbol{A}_i$, $i \geq 2$ satisfying $\boldsymbol{A}_2 = \boldsymbol{A}_3 = \cdots = \boldsymbol{A}_s$. Our main proofs strongly rely on results of [7] on the connection between hPNN and tensors decomposition in the shallow (i.e., 2-layer) case (see also [79]).

**Proposition 34.** *There is a one-to-one mapping between partially symmetric tensors $\boldsymbol{\mathcal{F}} \in \mathbb{R}^{d_2 \times d_0 \times \cdots \times d_0}$ and polynomial vectors $\boldsymbol{f} \in (\mathscr{H}_{d_0,r})^{\times d_2}$, which can be written as*

$$\boldsymbol{\mathcal{F}} \mapsto \boldsymbol{f}(\boldsymbol{x}) = \boldsymbol{F}^{(1)} \boldsymbol{x}^{\otimes r},$$

*with $\boldsymbol{F}^{(1)} \in \mathbb{R}^{d_2 \times d_0^r}$ the first unfolding of $\boldsymbol{\mathcal{F}}$. Under this mapping, the partially symmetric CPD*

$$\boldsymbol{\mathcal{F}} = [\![\boldsymbol{W}_2, \boldsymbol{W}_1^\mathsf{T}, \cdots, \boldsymbol{W}_1^\mathsf{T}]\!] \tag{6}$$

*is mapped to hPNN $\boldsymbol{W}_2 \rho_r(\boldsymbol{W}_1 \boldsymbol{x})$. Thus, uniqueness of $\mathrm{hPNN}_{(d_0,d_1,d_2),r}[(\boldsymbol{W}_1, \boldsymbol{W}_2)]$ is equivalent to uniqueness of the partially symmetric CPD of $\boldsymbol{\mathcal{F}}$.*

Thanks to the link with the partially symmetric CPD, we prove the following Kruskal-based sufficient condition for uniqueness (which is a counterpart of Proposition 33).

**Proposition 35.** *Let $\boldsymbol{p}_w(\boldsymbol{x}) = \boldsymbol{W}_2 \rho_{r_1}(\boldsymbol{W}_1 \boldsymbol{x})$ be a 2-layer hPNN with layer sizes $(d_0, d_1, d_2)$ satisfying $d_0, d_1 \geq 2$, $d_2 \geq 1$. Assume that $r \geq 2$, $\mathrm{krank}\{\boldsymbol{W}_2\} \geq 1$, $\mathrm{krank}\{\boldsymbol{W}_1^\mathsf{T}\} \geq 2$ and that:*

$$r \geq \frac{2d_1 - \mathrm{krank}\{\boldsymbol{W}_2\}}{\mathrm{krank}\{\boldsymbol{W}_1^\mathsf{T}\} - 1},$$

*then the 2-layer hPNN $\boldsymbol{p}_w(\boldsymbol{x})$ is unique (or equivalently, the CPD of $\boldsymbol{\mathcal{F}}$ in (6) is unique).*

**Remark 36.** *For 2-layer hPNNs ($L = 2$), when the activation degree $r$ is high enough Proposition 33 gives both necessary and sufficient conditions for uniqueness due to Proposition 35.*

**Remark 37.** *Proposition 35 forms the basis of the proof of Proposition 12, which comes from the fact that the Kruskal rank of a generic matrix is equal to its smallest dimension.*

**Remark 38.** *Proposition 35 is based on basic (Kruskal) uniqueness conditions [80–82]. As mentioned in Remark 15, by using more powerful results on generic uniqueness [83, 84], we can obtain better bounds for identifiability of 2-layer PNNs. For example, for "bottleneck" architectures (as in Corollary 19), the results of [83, Thm 1.11-12] imply that for degrees $r_\ell = 2$, identifiability holds for decoder layer sizes satisfying a weaker condition $d_\ell \leq \frac{(d_b-1)d_b}{2}$ (instead of $\frac{d_\ell}{r_\ell} \leq d_b - 1$).*

### 4.3 Proof of the main result

The proof of Theorem 11 proceeds by induction over the layers $\ell = 1, \ldots, L$. The key idea is based on a procedure that allows us to prove finite identifiability of the $L$-th layer given the assumption that the previous layers are identifiable. For this, we introduce a map (*last layer map*)

$$\psi[\boldsymbol{q}, \boldsymbol{W}_L] := \boldsymbol{W}_L \rho_{r_{L-1}}(\boldsymbol{q}(x_1, \ldots, x_{d_0})), \tag{7}$$

where $\boldsymbol{q}$ is the vector polynomial of degree $R = r_1 \cdots r_{L-2}$, representing the output of the $(L-1)$-th linear layer. Then the $L$-layer hPNN is a composition:

$$\text{hPNN}_{\boldsymbol{r}}[\boldsymbol{\theta}, \boldsymbol{W}_L] = \psi[\text{hPNN}_{(r_1, \ldots, r_{L-2})}[\boldsymbol{\theta}], \boldsymbol{W}_L], \quad \text{for } \boldsymbol{\theta} = (\boldsymbol{W}_1, \ldots, \boldsymbol{W}_{L-1}).$$

To obtain finite identifiability, we look at the Jacobian of the composite map. The key to this recursion is to show that the Jacobian $J_\psi(\boldsymbol{q}, \boldsymbol{W}_L)$ (Jacobian of $\psi$ with respect to the input polynomial vector and $\boldsymbol{W}_L$) is of maximal possible rank. For this, we construct a "certificate" of finite identifiability $\widehat{\boldsymbol{q}}$ realized by $\text{hPNN}_{(r_1, \ldots, r_{L-2})}[\widehat{\boldsymbol{\theta}}]$, but of simpler structure which inherits identifiability of a shallow hPNN.

**Remark 39.** *For $d_L = 1$, maximality of the rank for $J_\psi(\boldsymbol{q}, \boldsymbol{W}_L)$ is closely related to nondefectivity of the variety of sums of powers of forms, which is often proved by establishing Hilbert genericity of an ideal generated by the elements of $\boldsymbol{q}$ (a question raised in Fröberg conjecture, see e.g., [85]).*

A key limitation of our techniques is that they only allow for establishing finite identifiability for deep PNNs. There exist recent results linking finite and global identifiability, [75, 86] but only for additive decompositions (shallow case). We state, however, the following conjecture.

**Conjecture 40.** *Under the assumptions of Theorem 11, the L-layer hPNN is globally identifiable.*

Note that the conjecture may be valid only for global identifiability (i.e., for a generic choice of parameters) and not for uniqueness, since it is not true that the composition of unique shallow hPNNs yield a unique deep hPNNs, as shown by the following example.

**Example 41.** *Consider two polynomials: $\boldsymbol{p}(x_1, x_2) = \left[(x_1^2 + x_2^2)^2 \quad (x_1^2 - x_2^2)^2\right]^\mathsf{T}$. We see that this polynomial vector admits two different representations*

$$\boldsymbol{p}(\boldsymbol{x}) = \boldsymbol{I}\rho_2(\boldsymbol{W}_2\rho_2(\boldsymbol{I}\boldsymbol{x})) = \boldsymbol{W}_3\rho_2\left(\frac{1}{2}\boldsymbol{W}_2\rho_2(\boldsymbol{W}_2\boldsymbol{x})\right),$$

*with*

$$\boldsymbol{W}_2 = \begin{bmatrix} 1 & 1 \\ 1 & -1 \end{bmatrix}, \quad \boldsymbol{W}_3 = \begin{bmatrix} 1 & 0 \\ 1 & -1 \end{bmatrix},$$

*which are not equivalent. However, each 2-layer subnetwork is unique (see Example 7).*

## 5 Discussion

In this paper, we presented a comprehensive analysis of the identifiability of deep feedforward PNNs by using their connections to tensor decompositions. Our main result is the *localization of identifiability*, showing that deep PNNs are finitely identifiable if every 2-layer subnetwork is also finitely identifiable for a subset of their inputs. Our results can be also useful for compression (pruning) neural networks as they give an indication about the architectures that are not reducible. An important perspective is also to understand when two different identifiable PNN architectures can represent the same function, as the identifiable representations can potentially occur for different non-compatible formats (e.g., a PNN in format $\boldsymbol{d} = (2, 4, 4, 2)$ could be potentially pruned to two *different* identifiable representations, say, $\boldsymbol{d} = (2, 3, 4, 2)$ and $\boldsymbol{d} = (2, 4, 3, 2)$).

While our results focus on the case of monomial activations, we believe that this approach can be extended for establishing theoretical guarantees for other types of architectures and activation functions. In fact, the monomial case constitutes as a key first step in addressing general polynomial activations (see, e.g., [45]) which, in turn, can approximate most commonly used activations on compact sets. Moreover, the close connection between PNNs and partially symmetric tensor decompositions (which benefit from efficient computational algorithms based on linear algebra [87]) can also serve as support for the development of computational algorithms based on tensor decompositions for training deep PNNs. In fact, tensor decompositions have been combined with the method of moments to learn small NN architectures (see, e.g., [52, 88]), extending such approaches for training deep PNNs with finite datasets is an important direction for future work.

## Acknowledgments

This work was supported in part by the French National Research Agency (ANR) under grants ANR-23-CE23-0024, ANR-23-CE94-0001, by the PEPR project CAUSALI-T-AI, and by the National Science Foundation, under grant NSF 2316420.

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

# A roadmap to the appendices[4]

The appendices of the paper contain background on tensor decompositions and neurovarieties, the proofs of the technical results, as well as a discussion on the changes between the originally submitted and final version of the paper. They are organized as follows:

- Appendix A presents background on neurovarieties for homogeneous PNNs. This is a crucial part for understanding the link between finite identifiability of an hPNN, the dimension of its neurovariety and the rank of the Jacobian of its parametrization map.

- Appendix B contains the main technical tools used in the proof the localization theorem and follows the structure of Section 4. In particular, it presents the proofs of necessary conditions for uniqueness (Section 4.1), background on tensor decompositions and Kruskal-based sufficient conditions for the identifiability of 2-layer hPNNs (Section 4.2).

- Appendix C presents the proof of the localization theorem (Theorem 11) and its consequences for several hPNN architectures, as well as some supporting technical results.

- Appendix D presents the proofs for the case of PNNs with biases. Appendix D.3 discusses the idea of *truncation*, an alternative approach to tackle the PNNs with biases.

- Appendix E discusses necessary and sufficient conditions for the identifiability of hPNNs, as well as changes between the originally submitted and the final version of the paper which were done to correct a mistake in the proof of one of the main results.

## A  Homogeneous PNNs and neurovarieties

hPNNs are often studied through the prism of neurovarieties, using their algebraic structure. Our results have direct implications on the expected dimension of the neurovarieties, as explained in this appendix.

### A.1  Neurovarieties and dimension

An hPNN architecture $(\boldsymbol{d}, \boldsymbol{r})$ defines a map $\mathrm{hPNN}_{\boldsymbol{d},\boldsymbol{r}}[\cdot]$ from the weight tuple $\boldsymbol{w} = (\boldsymbol{W}_1, \ldots, \boldsymbol{W}_L)$ to a (polynomial) function space $\mathscr{H}$:

$$\mathrm{hPNN}_{\boldsymbol{d},\boldsymbol{r}}[\cdot]: \quad \boldsymbol{w} \mapsto \mathrm{hPNN}_{\boldsymbol{d},\boldsymbol{r}}[\boldsymbol{w}]$$
$$\mathbb{R}^{\sum_\ell d_\ell d_{\ell-1}} \to \mathscr{H}.$$

The space $\mathscr{H}$ is the space of length-$d_L$ vectors of homogeneous polynomials of degree $r_{total} = r_1 r_2 \ldots r_{L-1}$ in $d_0$ variables:

$$\mathscr{H} := (\mathscr{H}_{d_0, r_{total}})^{\times d_L};$$

thus $\mathscr{H}$ is a finite-dimensional vector space of dimension

$$N = \dim(\mathscr{H}) = d_L \binom{d_0 + r_{total} - 1}{r_{total}},$$

which follows from the fact that $\dim(\mathscr{H}_{d,r}) = \binom{d+r-1}{r}$.

The key observation is that $\mathrm{hPNN}_{\boldsymbol{d},\boldsymbol{r}}[\cdot]$ is a *polynomial-in-the-parameters* map, which has important implication on the space of networks with a given architecture. The image $\mathrm{Im}(\mathrm{hPNN}_{\boldsymbol{d},\boldsymbol{r}}[\cdot])$, called a *neuromanifold*, is a semi-algebraic set[5]. The properties of $\mathrm{Im}(\mathrm{hPNN}_{\boldsymbol{d},\boldsymbol{r}}[\cdot])$ are tightly linked to the properties of the *neurovariety* $\mathscr{V}_{\boldsymbol{d},\boldsymbol{r}}$ defined as the closure of $\mathrm{Im}(\mathrm{hPNN}_{\boldsymbol{d},\boldsymbol{r}}[\cdot])$ in the Zariski topology, i.e., the smallest algebraic variety ( algebraic set[6]) containing $\mathrm{Im}(\mathrm{hPNN}_{\boldsymbol{d},\boldsymbol{r}}[\cdot])$. The key property is the dimension of the neurovariety[7] which is equal to the dimension of the neurovariety [89, Prop. 2.8.2].

---

[4]The appendices have been reorganized and reworked for better readability.
[5][89, Def. 2.1.1]: a set cut out by polynomial equations and inequalities.
[6][89, Def. 2.1.4]: a set cut out by polynomial equations.
[7]roughly defined as the dimension of the tangent space at general point, see [89, §2.8] for more details.

The properties of neurovarieties depend on the field (i.e., results can differ between $\mathbb{R}$ or $\mathbb{C}$), and we focus on the real case. However, most of the results can be translated to the complex case as well. We mostly follow [90, Section 4], and an overview on semialgebraic sets can be also found in [91] (see [89] for a detailed account).

The following upper bound on $\dim \mathscr{V}_{\boldsymbol{d},\boldsymbol{r}}$ the bound was presented in [7]:

$$\dim \mathscr{V}_{\boldsymbol{d},\boldsymbol{r}} \leq \min \bigg( \underbrace{\sum_{\ell=1}^{L} d_\ell d_{\ell-1} - \sum_{\ell=1}^{L-1} d_\ell}_{\text{degrees of freedom}}, \quad \underbrace{\dim \mathscr{H}}_{\text{output space dimension}} \bigg). \tag{8}$$

If there is an equality in the bound (8), we say that the neurovariety has *expected dimension*. There are two fundamentally different cases when the expected dimension is reached.

**Expressive case.** If the right bound is reached, i.e., the neurovariety:

$$\dim \mathscr{V}_{\boldsymbol{d},\boldsymbol{r}} = \dim \big( \mathscr{H} \big) = d_L \binom{d_0 + r_{total} - 1}{r_{total}},$$

the hPNN is *expressive*, and the neurovariety $\mathscr{V}_{\boldsymbol{d},\boldsymbol{r}}$ is said to be *thick* [7], as it fills the whole function space $\mathscr{H}$ (and thus the neuromanifold is of positive Lebesgue measure). In particular, this implies that (see [7, Proposition 5]) any homogeneous polynomial vector from $\mathscr{H}$ (i.e., of degree $r_{total}$ with $d_0$ inputs and $d_L$ outputs, with degrees fixed as $r_1 = r_2 = \cdots = r_{L-1}$) can be represented as an hPNN with layer widths $(d_0, 2d_1, \ldots, 2d_{L-1}, d_L)$ and the same activation degrees.

**Identifiable case.** The left bound $(\sum_{\ell=1}^{L} d_\ell d_{\ell-1} - \sum_{\ell=1}^{L-1} d_\ell)$ follows from the presence of equivalences defined in Lemma 4 (i.e., the size of the vector $\boldsymbol{w}$ minus the number of independent rescalings) and defines the number of effective parameters of the representation (this is explained in the following subsections). Moreover, the left bound is reached if and only if the hPNN architecture is finitely identifiable:

**Proposition 10** *The architecture* $\mathrm{hPNN}_{\boldsymbol{d},\boldsymbol{r}}[\cdot]$ *is finitely identifiable if and only if the dimension of* $\mathscr{V}_{\boldsymbol{d},\boldsymbol{r}}$ *is equal to the effective number of parameters, i.e.,* $\dim \mathscr{V}_{\boldsymbol{d},\boldsymbol{r}} = \sum_{\ell=1}^{L} d_\ell d_{\ell-1} - \sum_{\ell=1}^{L-1} d_\ell$. *In such case,* $\mathscr{V}_{\boldsymbol{d},\boldsymbol{r}}$ *is said to be* **nondefective**. *Equivalently, the rank of the Jacobian of the map* $\mathrm{hPNN}_{\boldsymbol{d},\boldsymbol{r}}[\cdot]$ *is maximal and equal to* $\sum_{\ell=1}^{L} d_\ell d_{\ell-1} - \sum_{\ell=1}^{L-1} d_\ell$ *at a general parameter* $\boldsymbol{w}$.

Proposition 10 is central to the proof of the main results of paper. The proof of Proposition 10 relies on properties of fibers of polynomial maps and is reviewed in the next subsection, together with the Jacobian of the parameterization.

## A.2 Polynomial maps and fiber dimension

We recall some key facts on the polynomial maps and their images. We begin by highlighting the link between dimensions of semialgebraic sets and the Jacobian of the polynomial maps.

**Lemma A.1.** *Let* $\varphi : \mathbb{R}^m \to \mathbb{R}^n$ *be a polynomial map, and denote by* $J_\varphi(\boldsymbol{\theta})$ *the* $n \times m$ *Jacobian matrix. Let*

$$r_0 := \max_{\boldsymbol{\theta}} \mathrm{rank}\{J_\varphi(\boldsymbol{\theta})\}.$$

*Then we have that:*

1. $\mathrm{rank}\{J_\varphi(\boldsymbol{\theta})\} = r_0$ *for generic* $\boldsymbol{\theta}$ *(i.e., for all* $\boldsymbol{\theta} \in \mathbb{R}^m$ *except a set of Lebesgue measure zero, where the rank of the Jacobian is strictly less than* $r_0$*).*

2. $r_0$ *is equal to the dimension of* $\mathrm{Im}(\varphi)$ *and its (Zariski) closure:*

$$r_0 = \dim \big( \mathrm{Im}(\varphi) \big) = \dim \big( \overline{\mathrm{Im}(\varphi)} \big).$$

The proof of Lemma A.1 is given in [90, Theorem 4.7] and the preceding paragraph (in [90], the number $r_0$ is called *generic rank* of the parameterization $\varphi$). It mainly follows from semicontinuity of the rank of a matrix.

**Remark A.2** (On genericity). *Due to the algebraic structure of $\varphi$, the genericity statement in Lemma A.1 is much stronger: in fact, the set of points $\boldsymbol{\theta}$ where $\text{rank}\{J_\varphi(\boldsymbol{\theta})\} \neq r_0$ is a semialgebraic subset of $\mathbb{R}^m$ of dimension strictly less than $m$. The same holds for all generic statements and definitions in the paper (such as finite identifiability, global identifiability, etc.), see the definition of genericity in [90, Definition 4.1].*

**Remark A.3.** *The right bound for neurovariety dimension in* (8) *follows essentially from Lemma A.1: indeed, in the case $\varphi(\cdot) = \text{hPNN}_{\boldsymbol{d},\boldsymbol{r}}[\cdot]$, $\text{rank}\{J_\varphi\}$ does not exceed the dimension of the ambient space of $\varphi$ (equal to $\dim(\mathscr{H})$).*

The following lemma is key for linking finite identifiability to the dimension of the neurovariety.

**Lemma A.4** (Fiber dimension). *Let $\varphi : \mathbb{R}^m \to \mathbb{R}^n$ be a polynomial map, so that $r_0 = \dim(\text{Im}(\varphi))$. Then the dimension of its generic fiber is equal to $m - r_0$, that is, for generic $\boldsymbol{\theta} \in \mathbb{R}^m$, the preimage $\varphi^{-1}(\varphi(\boldsymbol{\theta}))$ is a semialgebraic set with*

$$\dim \varphi^{-1}\big(\varphi(\boldsymbol{\theta})\big) = m - r_0.$$

Lemma A.4 is well known to specialists, but in the literature it is mostly formulated for the complex case (see [90, Theorem 4.7]). For the real field it is a special case of [90, Theorem 4.9].

A particular case is when $r_0 = m$, in which case Lemma A.4 implies finiteness of the fiber:

**Corollary A.5.** *The following two statements are equivalent:*

- *For general $\boldsymbol{\theta} \in \mathbb{R}^m$, $\text{rank}\{J_\varphi(\boldsymbol{\theta})\} = m$;*

- *For general $\boldsymbol{\theta} \in \mathbb{R}^m$, the fiber (i.e., the preimage $\varphi^{-1}(\varphi(\boldsymbol{\theta}))$) consists of a finite number of points.*

*Proof.* The statement follows from Lemma A.4 specialized to $(r_0 = m)$ and from the fact that 0-dimensional semialgebraic sets are collections of a finite number of points. $\square$

Finally we make the following remark that is very commonly used.

**Corollary A.6.** *If $\text{rank}\{J_\varphi(\boldsymbol{\theta}_0)\} = m$ for some $\boldsymbol{\theta}_0 \in \mathbb{R}^m$, then $\text{rank}\{J_\varphi(\boldsymbol{\theta})\} = m$ for generic $\boldsymbol{\theta}$.*

*Proof.* This directly follows from Lemma A.1, since $r_0$ in Lemma A.1 is equal to $m$. $\square$

**Remark A.7.** *Corollary A.6 implies that finding a single point with full column rank Jacobian implies finitieness of the generic fiber.*

### A.2.1 The case of neurovarieties

The first implication of Lemma A.4 is the left upper bound in (8). It is based on the following lemma from [7], for which we provide a short proof for completeness.

**Lemma A.8** ([7, Lemma 13]). *For a general parameter $\boldsymbol{w} = (\boldsymbol{W}_1, \dots, \boldsymbol{W}_L)$, the set of equivalent hPNN representations in Lemma 4 is semialgebraic and of dimension $\sum_{\ell=1}^{L-1} d_\ell$.*

*Proof.* First, note that the set of equivalent representations is of dimension at most $\sum_{\ell=1}^{L-1} d_\ell$ (by the number of parameters). Consider a general $\boldsymbol{w} = (\boldsymbol{W}_1, \dots, \boldsymbol{W}_L)$, so that the first column of each $\boldsymbol{W}_\ell$, for $\ell = 1, \dots, L-1$, equal to $\boldsymbol{v}_\ell \in \mathbb{R}^{d_\ell}$, does not have zero elements. Now take any collection of vectors $\widetilde{\boldsymbol{v}}_1 \in \mathbb{R}^{d_1}, \dots, \widetilde{\boldsymbol{v}}_{L-1} \in \mathbb{R}^{d_{L-1}}$ having elementwise the same signs as $\boldsymbol{v}_\ell$. Then there exist matrices $\boldsymbol{D}_\ell$ so that the equivalent weight matrices $\widetilde{\boldsymbol{W}}_\ell = \widetilde{\boldsymbol{D}}_\ell \boldsymbol{W}_\ell \widetilde{\boldsymbol{D}}_{\ell-1}^{-r_{\ell-1}}$ have $\widetilde{\boldsymbol{v}}_\ell$ exactly as their first columns. Thus the set of equivalent representations is exactly of dimension $\sum_{\ell=1}^{L-1} d_\ell$. $\square$

**Remark A.9.** *The left upper bound in* (8) *simply follows from Lemma A.8 (as written in [7, Lemma 13]): indeed, the dimension of the fiber of $\text{hPNN}_{\boldsymbol{d},\boldsymbol{r}}[\cdot]$ must be at least $\sum_{\ell=1}^{L-1} d_\ell$. This implies, by Lemma A.4,*

$$\text{rank}\{J_\varphi(\boldsymbol{\theta})\} \leq \sum_{\ell=1}^{L} d_{\ell-1} d_\ell - \sum_{\ell=1}^{L-1} d_\ell, \tag{9}$$

*which is exactly the right dimension bound in* (8) *by Lemma A.1.*

Note that Proposition 10 will exactly consider the case when the equality is reached in (9) for generic $\boldsymbol{\theta}$. Similarly to Corollary A.6, the following corollary of Lemma A.1 implies that for the case of neurovarieties it suffices to find a single set of parameters $\boldsymbol{w}$ where the Jacobian of the parameterization is of maximal rank to guarantee finite identifiability of hPNN architecture. This will be used in the proofs to give a *certificate* of finite idenitifiability.

**Corollary A.10.** *If there exists a particular point $\boldsymbol{\theta}_0$ such that equality is achieved in* (9)*, then the equality in* (9) *is achieved for generic $\boldsymbol{\theta}$.*

*Proof.* Since there exists such a $\boldsymbol{\theta}_0$, then the $r_0$ defined in Lemma A.1 satisfies

$$r_0 \geq \sum_{\ell=1}^{L} d_{\ell-1}d_\ell - \sum_{\ell=1}^{L-1} d_\ell. \tag{10}$$

But from (9), $r_0$ must be bounded from above by the same number. Therefore the equality for $r_0$ is achieved in (10). □

### A.3 Proof of the proposition

*Proof of Proposition 10.* We denote $\varphi(\cdot) = \text{hPNN}_{\boldsymbol{d},\boldsymbol{r}}[\cdot]$ for simplicity (so that $m = \sum_{\ell=1}^{L} d_\ell d_{\ell-1}$ and $n = \dim \mathcal{H}$) and consider separately the "only if" ($\Rightarrow$) and "if" ($\Leftarrow$) parts.

$\boxed{\Rightarrow}$ Assume that for a generic $\boldsymbol{w}$ the fiber $\varphi^{-1}(\varphi(\boldsymbol{w}))$ consists of finite number of equivalence classes, thus it is a finite union of non-intersecting semialgebraic subsets of dimension $\sum_{\ell=1}^{L-1} d_\ell$. Therefore, by [89, Theorem 2.8.5] the whole fiber $\varphi^{-1}(\varphi(\boldsymbol{w}))$ has the dimension equal to $\sum_{\ell=1}^{L-1} d_\ell$ as well, hence $\dim \mathcal{V}_{\boldsymbol{d},\boldsymbol{r}} = \sum_{\ell=1}^{L} d_\ell d_{\ell-1} - \sum_{\ell=1}^{L-1} d_\ell$.

$\boxed{\Leftarrow}$ The proof follows a similar argument as in the proof of [90, Theorem 4.9]. We consider a (Zariski open) subset of parameters without zero values $\mathcal{U} = (\mathbb{R} \setminus \{0\})^m$. It can be shown that the preimage of the image of its complement $\mathcal{Z} := \varphi^{-1}(\varphi(\mathbb{R}^m \setminus \mathcal{U}))$ is a (semialgebraic) set of measure zero. Therefore for the set $\mathcal{U}' := \mathcal{U} \setminus \mathcal{Z}$ the preimage of the image is contained in $\mathcal{U}$:

$$\varphi^{-1}(\varphi(\mathcal{U}')) \subset \mathcal{U}.$$

Note that any $\boldsymbol{w} \in \mathcal{U}$ can be brought (by diagonal scaling and permutation) to an equivalent form:

$$\boldsymbol{W}_\ell = \begin{bmatrix} 1 \cdots 1 \\ \hline \overline{\boldsymbol{W}}_\ell \end{bmatrix}, \quad \overline{\boldsymbol{W}}_\ell \in \mathbb{R}^{(d_\ell-1)\times d_{\ell-1}} \tag{11}$$

for all $\ell = 2, \ldots, L$ where the reduced $\overline{\boldsymbol{W}}_\ell$ parameterize the classes of equivalent parameters in $\mathcal{U}$ up to permutation. Now denote $\overline{\boldsymbol{w}} = (\boldsymbol{W}_1, \overline{\boldsymbol{W}}_2, \ldots, \overline{\boldsymbol{W}}_L)$ and define $\boldsymbol{w}(\overline{\boldsymbol{w}}) = (\boldsymbol{W}_1, \ldots, \boldsymbol{W}_L)$ with $\boldsymbol{W}_\ell$ as in (11). Consider the following map

$$\psi : \overline{\boldsymbol{w}} \mapsto \text{hPNN}_{\boldsymbol{d},\boldsymbol{r}}[\boldsymbol{w}(\overline{\boldsymbol{w}})].$$

Then if the generic fiber of $\psi$ is finite, this will imply that on $\mathcal{U}'$, the fiber of the map $\varphi$ contains finitely many equivalence classes. For this, note that the Jacobian of $\psi$ is just a submatrix of the Jacobian of $\varphi$ with exactly $m - \sum_{\ell=1}^{L-1} d_\ell$ columns. We will show that it is full rank at a generic point $\overline{\boldsymbol{w}}$.

Consider the following map

$$\xi : (\boldsymbol{W}_1, \overline{\boldsymbol{W}}_2, \ldots, \overline{\boldsymbol{W}}_L, \boldsymbol{D}_1, \ldots, \boldsymbol{D}_2) \mapsto (\boldsymbol{W}_1, \widetilde{\boldsymbol{W}}_2, \ldots, \widetilde{\boldsymbol{W}}_L)$$

defined as

$$\widetilde{\boldsymbol{W}}_\ell = \boldsymbol{D}_\ell \begin{bmatrix} 1 \cdots 1 \\ \hline \overline{\boldsymbol{W}}_\ell \end{bmatrix} \boldsymbol{D}_\ell^{-r_{\ell-1}}$$

for $\ell = 2, \ldots, L$ (with the convention that $\boldsymbol{D}_L = \boldsymbol{I}_{d_L}$.

Consider a particular $\overline{\boldsymbol{w}}_0$ constructed as above (by normalization of a $\boldsymbol{w} \in \overline{\mathcal{U}}$). Then for a neighborhood $\overline{\mathcal{U}}$ of $\overline{\boldsymbol{w}}_0$ and a neighbourhood $\mathcal{V}$ of $(\boldsymbol{I}_{d_1}, \ldots, \boldsymbol{I}_{d_{L-1}})$, the map $\xi$ is a diffeomorphism from $\overline{\mathcal{U}} \times \mathcal{V}$ to an open neigbourhood of the corresponding $\boldsymbol{w}_0 = \boldsymbol{w}(\overline{\boldsymbol{w}}_0)$.

Consider the composition $\varphi \circ \xi$. Then at the point $(\boldsymbol{W}_1, \overline{\boldsymbol{W}}_2, \ldots, \overline{\boldsymbol{W}}_L, \boldsymbol{I}_{d_1}, \ldots, \boldsymbol{I}_{d_{L-1}})$, we have that (i) the derivatives with respect to $\boldsymbol{D}_\ell$ at identity matrices are zero and (ii) the Jacobian of $\varphi \circ \xi$ with respect to $\overline{\boldsymbol{w}}$ coincides with the Jacobian of $\psi$, hence it must have full column rank $(m - \sum_{\ell=1}^{L-1} d_\ell)$ which is equal to the dimension of the neurovariety. Hence, the fiber of $\psi$ is finite, which implies finite identifiability of $\varphi$. □

# B Main tools for the proof

This appendix contains the main technical tools used in the proof the localization theorem. It is organized in three subsections, following the same structure as in Section 4:

- Appendix B.1 presents the proofs of necessary conditions for uniqueness corresponding to Section 4.1 in the main body of the paper;

- Appendix B.2 presents background on tensor decompositions and the proof of Proposition 34 from the main body of the paper, which shows the link between 2-layer hPNNs and partially symmetric tensors;

- Appendix B.3 presents Kruskal-based sufficient conditions for the identifiability of 2-layer hPNNs (Propositions 35 and 12 in the main paper).

## B.1 Necessary conditions for uniqueness

In this subsection we prove the key lemmas stated in Section 4.1 (Lemma 30 and Lemma 31). These results give necessary conditions for the uniqueness of an hPNN in terms of the minimality of an unique architectures and the independence (non-redundancy) of its internal representations.

***Lemma 30.*** *Let $\boldsymbol{p} = \mathrm{hPNN}_{\boldsymbol{r}}[\boldsymbol{w}]$ be an hPNN of format $(d_0, \ldots, d_\ell, \ldots, d_L)$. Then for any $\ell$ there exists an infinite number of representations of hPNNs $\boldsymbol{p} = \mathrm{hPNN}_{\boldsymbol{r}}[\boldsymbol{w}]$ with architecture $(d_0, \ldots, d_\ell + 1, \ldots, d_L)$. In particular, the augmented hPNN is not unique (and is not finite-to-one).*

*Proof of Lemma 30.* Let $(\boldsymbol{W}_0, \cdots, \boldsymbol{W}_L)$ the weight matrices associated with the representation of format $(d_0, \ldots, d_\ell, \ldots, d_L)$ of the hPNN $\boldsymbol{p} = \mathrm{hPNN}_{\boldsymbol{r}}[\boldsymbol{w}]$. By assumptions on the dimensions, the two matrices $\boldsymbol{W}_\ell \in \mathbb{R}^{d_\ell \times d_{\ell-1}}$ and $\boldsymbol{W}_{\ell+1} \in \mathbb{R}^{d_{\ell+1} \times d_\ell}$ read

$$\boldsymbol{W}_{\ell+1} = [\boldsymbol{w}_1 \quad \cdots \quad \boldsymbol{w}_{d_\ell}], \text{ where, for each } i, \ \boldsymbol{w}_i \in \mathbb{R}^{d_{\ell+1}},$$

$$\boldsymbol{W}_\ell = [\boldsymbol{v}_1 \quad \cdots \quad \boldsymbol{v}_{d_\ell}]^\mathsf{T}, \text{ where, for each } i, \ \boldsymbol{v}_i \in \mathbb{R}^{d_{\ell-1}}.$$

Without loss of generality, let us assume none of the $\boldsymbol{w}_i$ is a zero vector[8], and set

$$\widetilde{\boldsymbol{W}}_\ell = [\boldsymbol{0} \quad \boldsymbol{v}_1 \quad \cdots \quad \boldsymbol{v}_{d_\ell}]^\mathsf{T} \in \mathbb{R}^{(d_\ell+1) \times d_{\ell-1}},$$

in which we add a row of zeroes to $\boldsymbol{W}_\ell$. In this case, we can take the following family of matrices defined for any $\boldsymbol{u} \in \mathbb{R}^{d_{\ell+1}}$:

$$\widetilde{\boldsymbol{W}}_{\ell+1}^{(\boldsymbol{u})} = [\boldsymbol{u} \quad \boldsymbol{w}_1 \quad \cdots \quad \boldsymbol{w}_{d_\ell}] \in \mathbb{R}^{d_{\ell+1} \times (d_\ell+1)}.$$

Then, we have that for any choice of $\boldsymbol{u}$ and for any $\boldsymbol{z}$,

$$\widetilde{\boldsymbol{W}}_{\ell+1}^{(\boldsymbol{u})} \rho_{r_\ell}(\widetilde{\boldsymbol{W}}_\ell \boldsymbol{z}) = \boldsymbol{W}_{\ell+1} \rho_{r_\ell}(\boldsymbol{W}_\ell \boldsymbol{z}).$$

The matrices $\widetilde{\boldsymbol{W}}_{\ell+1}^{(\boldsymbol{0})}$ and $\widetilde{\boldsymbol{W}}_{\ell+1}^{(\boldsymbol{u})}$ for $\boldsymbol{u} \neq \boldsymbol{0}$ have a different number of zero columns and cannot be a permutation/rescaling of each other, constituting different representations of the same hPNN $\boldsymbol{p}$. In fact, every choice of $\boldsymbol{u}'$ that is not collinear to $\boldsymbol{u}$ and $\boldsymbol{w}_i$, $i = 1, \ldots, d_\ell$ leads to a different non-equivalent representation of $\boldsymbol{p}$. Thus, we have an infinite number of non-equivalent representations

$$(\boldsymbol{W}_0, \ldots, \boldsymbol{W}_{\ell-1}, \widetilde{\boldsymbol{W}}_\ell, \widetilde{\boldsymbol{W}}_{\ell+1}^{(\boldsymbol{u})}, \ldots, \boldsymbol{W}_L)$$

of format $(d_0, \ldots, d_\ell + 1, \ldots, d_L)$ for the hPNN $\boldsymbol{p} = \mathrm{hPNN}_{\boldsymbol{r}}[\boldsymbol{w}]$. □

---

[8]otherwise, we can replace a zero vector $\boldsymbol{w}_i$ with a randomly chosen non-zero vector and set the corresponding $\boldsymbol{v}_i = 0$

Lemma 30 can be seen as a form of minimality or irreducibility of unique hPNNs, as it shows that a unique hPNN does not admit a smaller (i.e., with a lower number of neurons) representation.

**Lemma 31.** *For the widths $\boldsymbol{d} = (d_0, \ldots, d_L)$, let $\boldsymbol{p} = \mathrm{hPNN}_{\boldsymbol{r}}[\boldsymbol{w}]$ be a unique $L$-layers decomposition. Consider the vector output at any $\ell$-th internal level $\ell < L$ after the activations*

$$\boldsymbol{q}_\ell(\boldsymbol{x}) = \rho_{r_\ell} \circ \boldsymbol{W}_\ell \circ \cdots \circ \rho_{r_1} \circ \boldsymbol{W}_1(\boldsymbol{x}).$$

*Then the elements $\boldsymbol{q}_\ell(\boldsymbol{x}) = [q_{\ell,1}(\boldsymbol{x}) \quad \cdots \quad q_{\ell,d_\ell}(\boldsymbol{x})]^\mathsf{T}$ are linearly independent polynomials.*

*Proof of Lemma 31.* By contradiction, suppose that the polynomials $q_{\ell,1}(\boldsymbol{x}), \ldots, q_{\ell,d_\ell}(\boldsymbol{x})$ are linearly dependent. Assume without loss of generality that, e.g., the last polynomial $q_{\ell,d_\ell}(\boldsymbol{x})$ can expressed as a linear combination of the others. Then, there exists a matrix $\boldsymbol{B} \in \mathbb{R}^{d_\ell \times (d_\ell - 1)}$ so that

$$\boldsymbol{p} = \boldsymbol{W}_L \circ \rho_{r_{L-1}} \circ \cdots \circ \rho_{r_{\ell+1}} \circ \boldsymbol{W}_{\ell+1} \boldsymbol{B} \begin{bmatrix} q_{\ell,1}(\boldsymbol{x}) \\ \vdots \\ q_{\ell,d_\ell-1}(\boldsymbol{x}) \end{bmatrix},$$

i.e., the hPNN $\boldsymbol{p}$ admits a representation of size $\boldsymbol{d} = (d_0, \ldots, d_\ell - 1, \ldots, d_L)$ with parameters $(\boldsymbol{W}_1, \ldots, \boldsymbol{W}_{\ell+1} \boldsymbol{B}, \ldots, \boldsymbol{W}_L)$. Therefore, by Lemma 30 its original representation is not unique, which is a contradiction. $\qquad\square$

Using Lemma 30 and Lemma 31, we can prove the conditions on the Kruskal ranks of weight matrices that are necessary for uniqueness. These conditions are based on the notion of Kruskal rank which we recall from [15].

**Definition 32.** *The Kruskal rank of a matrix $\boldsymbol{A}$ (denoted $\mathrm{krank}\{\boldsymbol{A}\}$) is the maximal number $k$ such that any $k$ columns of $\boldsymbol{A}$ are linearly independent.*

Note that the following two cases of particular interest also have simple equivalent interpretations:

- $\mathrm{krank}\{\boldsymbol{A}\} \geq 1$ is equivalent to saying that matrix $\boldsymbol{A}$ has no zero columns;
- $\mathrm{krank}\{\boldsymbol{A}\} \geq 2$ is equivalent to saying that no pair of the columns of matrix $\boldsymbol{A}$ are collinear.

**Proposition 33.** *As in Lemma 31, let the widths be $\boldsymbol{d} = (d_0, \ldots, d_L)$, and $\boldsymbol{p} = \mathrm{hPNN}_{\boldsymbol{r}}[\boldsymbol{w}]$ have a unique (or finite-to-one) $L$-layers decomposition. Then we have that for all $\ell = 1, \ldots, L-1$*

$$\mathrm{krank}\{\boldsymbol{W}_\ell^\mathsf{T}\} \geq 2, \quad \mathrm{krank}\{\boldsymbol{W}_{\ell+1}\} \geq 1,$$

*where $\mathrm{krank}\{\boldsymbol{W}_{\ell+1}\} \geq 1$ simply means that $\boldsymbol{W}_{\ell+1}$ does not have zero columns.*

*Proof of Proposition 33.* Suppose that $\mathrm{krank}\{\boldsymbol{W}_\ell^\mathsf{T}\} < 2$. Then we have that at level $\ell$, the vector $\boldsymbol{q}_\ell(\boldsymbol{x})$ of internal features defined in (5) contains linearly dependent or zero polynomials, which violates Lemma 31.

Similarly if $\mathrm{krank}\{\boldsymbol{W}_{\ell+1}\} = 0$, then the neuron corresponding to the zero column can be pruned to obtain a representation with $(d_\ell - 1)$ neurons at the $\ell$-th level, which implies loss of uniqueness by Lemma 30 and thus leads to a contradiction. $\qquad\square$

## B.2 Background on tensors

In this appendix, we first present a background on tensors and the CP tensor decomposition, and demonstrate the link between hPNNs and the partially symmetric CPD (Proposition 34 in the main paper).

### B.2.1 Basics on tensors and tensor decompositions

**Notation.** The order of a tensor is the number of dimensions, also known as ways or modes. Vectors (tensors of order one) are denoted by boldface lowercase letters, e.g., $\boldsymbol{a}$. Matrices (tensors of order two) are denoted by boldface capital letters, e.g., $\boldsymbol{A}$. Higher-order tensors (order three or higher) are denoted by boldface Euler script letters, e.g., $\boldsymbol{\mathcal{X}}$.

**Unfolding of tensors.** The $p$-th unfolding (also called mode-$p$ unfolding) of a tensor of order $s$, $\boldsymbol{\mathcal{T}} \in \mathbb{R}^{m_1 \times \cdots \times m_s}$ is the matrix $\boldsymbol{T}^{(p)}$ of size $m_p \times (m_1 m_2 \cdots m_{p-1} m_{p+1} \cdots m_s)$ defined as

$$\left[\boldsymbol{T}^{(p)}\right]_{i_p, j} = \boldsymbol{\mathcal{T}}_{i_1, \ldots, i_p, \ldots, i_s}, \text{ where } j = 1 + \sum_{\substack{n=1 \\ n \neq p}}^{s} (i_n - 1) \prod_{\substack{\ell=1 \\ \ell \neq p}}^{n-1} m_\ell \,.$$

We give an example of unfolding extracted from [14]. Let the frontal slices of $\boldsymbol{\mathcal{X}} \in \mathbb{R}^{3 \times 4 \times 2}$ be

$$\boldsymbol{X}_1 = \begin{pmatrix} 1 & 4 & 7 & 10 \\ 2 & 5 & 8 & 11 \\ 3 & 6 & 9 & 12 \end{pmatrix}, \ \boldsymbol{X}_2 = \begin{pmatrix} 13 & 16 & 19 & 22 \\ 14 & 17 & 20 & 23 \\ 15 & 18 & 21 & 24 \end{pmatrix} .$$

Then the three mode-$n$ unfoldings of $\boldsymbol{\mathcal{X}}$ are

$$\boldsymbol{X}^{(1)} = \begin{pmatrix} 1 & 4 & 7 & 10 & 13 & 16 & 19 & 22 \\ 2 & 5 & 8 & 11 & 14 & 17 & 20 & 23 \\ 3 & 6 & 9 & 12 & 15 & 18 & 21 & 24 \end{pmatrix}$$

$$\boldsymbol{X}^{(2)} = \begin{pmatrix} 1 & 2 & 3 & 13 & 14 & 15 \\ 4 & 5 & 6 & 16 & 17 & 18 \\ 7 & 8 & 9 & 19 & 20 & 21 \\ 10 & 11 & 12 & 22 & 23 & 24 \end{pmatrix}$$

$$\boldsymbol{X}^{(3)} = \begin{pmatrix} 1 & 2 & 3 & 4 & 5 & 6 & \cdots & 10 & 11 & 12 \\ 13 & 14 & 15 & 16 & 17 & 18 & \cdots & 22 & 23 & 24 \end{pmatrix}$$

**Symmetric and partially symmetric tensors.** A tensor of order $s$, $\boldsymbol{\mathcal{T}} \in \mathbb{R}^{m_1 \times \cdots \times m_s}$ is said to be *symmetric* if $m_1 = \cdots = m_s$ and for every permutation $\sigma$ of $\{1, \ldots, s\}$:

$$\boldsymbol{\mathcal{T}}_{i_1, i_2, \cdots, i_s} = \boldsymbol{\mathcal{T}}_{i_{\sigma(1)}, i_{\sigma(2)}, \ldots, i_{\sigma(s)}}.$$

The tensor $\boldsymbol{\mathcal{T}} \in \mathbb{R}^{m_1 \times \cdots \times m_s}$ is said to be *partially symmetric* along the modes $(p+1, \ldots, s)$ for $p < s$ if $m_{p+1} = \cdots = m_s$ and for every permutation $\sigma$ of $\{p+1, \ldots, s\}$

$$\boldsymbol{\mathcal{T}}_{i_1, i_2, \ldots, i_p, i_{p+1}, \cdots, i_s} = \boldsymbol{\mathcal{T}}_{i_1, \ldots, i_p, i_{\sigma(p+1)}, \ldots, i_{\sigma(s)}}.$$

**Mode products.** The $p$-mode (matrix) product of $\boldsymbol{\mathcal{T}} \in \mathbb{R}^{m_1 \times m_2 \times \cdots \times m_s}$ with a matrix $\boldsymbol{A} \in \mathbb{R}^{J \times m_p}$ is denoted by $\boldsymbol{\mathcal{T}} \bullet_p \boldsymbol{A}$ and is of size $m_1 \times \cdots \times m_{p-1} \times J \times m_{p+1} \times \cdots \times m_s$. It is defined as

$$\left[\boldsymbol{\mathcal{T}} \bullet_p \boldsymbol{A}\right]_{i_1, \ldots, i_{p-1}, j, i_{p+1}, \ldots, i_s} = \sum_{i_p=1}^{m_p} \boldsymbol{\mathcal{T}}_{i_1, \ldots, i_s} \boldsymbol{A}_{j, i_p} \,.$$

$R$**-term decomposition.** An $R$-term[9] canonical polyadic decomposition (CPD) of a tensor $\boldsymbol{\mathcal{T}}$ is a decomposition of a tensor as a sum of $R$ rank-1 tensors [14, 15], that is

$$\boldsymbol{\mathcal{T}} = \sum_{i=1}^{R} \boldsymbol{a}_i^{(1)} \otimes \cdots \otimes \boldsymbol{a}_i^{(s)},$$

where, for each $p \in \{1, \ldots, s\}$, $\boldsymbol{a}_i^{(p)} \in \mathbb{R}^{m_p}$, and $\otimes$ denotes the tensor (outer) product operation. Alternatively, we denote $\boldsymbol{A}^{(p)} = \left[\boldsymbol{a}_1^{(p)} \cdots \boldsymbol{a}_R^{(p)}\right] \in \mathbb{R}^{m_p \times R}$ and the corresponding CPD as

$$\boldsymbol{\mathcal{T}} = [\![\boldsymbol{A}^{(1)}, \boldsymbol{A}^{(2)}, \ldots, \boldsymbol{A}^{(s)}]\!].$$

When $\boldsymbol{\mathcal{T}}$ is partially symmetric along the modes $(p+1, \ldots, s)$, for $p < s$, its CPD satisfying $\boldsymbol{A}^{(p+1)} = \boldsymbol{A}^{(p+2)} = \cdots = \boldsymbol{A}^{(s)}$ is called *partially symmetric CPD*. The case of fully symmetric tensors (i.e., tensors which are symmetric along all their dimensions) deserves special attention [79]. The symmetric CPD of a fully symmetric tensor $\boldsymbol{\mathcal{T}} \in \mathbb{R}^{m \times m \times \cdots \times m}$ is defined as

$$\boldsymbol{\mathcal{T}} = \sum_{i=1}^{R} u_i \, \boldsymbol{a}_i \otimes \cdots \otimes \boldsymbol{a}_i \,,$$

where $u_i \in \mathbb{R}$ are real-valued coefficients. With a slight abuse of notation, we represent it compactly using the same notation as an order-$(n+1)$ tensor of size $1 \times m \times \cdots \times m$, as

$$\boldsymbol{\mathcal{T}} = [\![\boldsymbol{u}, \boldsymbol{A}, \cdots, \boldsymbol{A}]\!],$$

where $\boldsymbol{u} \in \mathbb{R}^{1 \times m}$ is a $1 \times m$ matrix (i.e., a row vector) containing the coefficients $u_i$, that is, $\boldsymbol{u}_i = u_i$, $i = 1, \ldots, R$.

---

[9]In the definition of CPD, we do not require $R$ to be minimal (thus $R$ is not necessarily equal to tensor rank).

### B.2.2 Link between hPNNs and partially symmetric tensors

Recall that $\mathscr{H}_{d_0,r}$ denotes the space of $d_0$-variate homogeneous polynomials of degree $\leq r$. The following proposition, originally presented in Section 4 of the main body of the paper, formalizes the link between polynomial vectors and partially symmetric tensors.

**Proposition 34.** *There is a one-to-one mapping between partially symmetric tensors $\mathcal{F} \in \mathbb{R}^{d_2 \times d_0 \times \cdots \times d_0}$ and polynomial vectors $\boldsymbol{f} \in (\mathscr{H}_{d_0,r})^{\times d_2}$, which can be written as[10]*

$$\mathcal{F} \mapsto \boldsymbol{f}(\boldsymbol{x}) = \boldsymbol{F}^{(1)}\boldsymbol{x}^{\otimes r},$$

*with $\boldsymbol{F}^{(1)} \in \mathbb{R}^{d_2 \times d_0^r}$ the first unfolding of $\mathcal{F}$. Under this mapping, the partially symmetric CPD*

$$\mathcal{F} = [\![\boldsymbol{W}_2, \boldsymbol{W}_1^\mathsf{T}, \cdots, \boldsymbol{W}_1^\mathsf{T}]\!]$$

*is mapped to hPNN $\boldsymbol{W}_2\rho_r(\boldsymbol{W}_1\boldsymbol{x})$. Thus, uniqueness of $\mathrm{hPNN}_{(d_0,d_1,d_2),r}[(\boldsymbol{W}_1, \boldsymbol{W}_2)]$ is equivalent to uniqueness of the partially symmetric CPD of $\mathcal{F}$.*

*Proof.* We distinguish the two cases, $d_2 = 1$ and $d_2 \geq 2$. We begin the proof by the more general case $d_2 \geq 2$.

**Case $d_2 \geq 2$.** Denoting by $\boldsymbol{u}_i \in \mathbb{R}^{d_2}$ the $i$-th column of $\boldsymbol{W}_2$ and $\boldsymbol{v}_i \in \mathbb{R}^{d_0}$ the $i$-th row of $\boldsymbol{W}_1$, the relationship between the 2-layer hPNN and tensor $\mathcal{F}$ can be written explicitly as

$$\begin{aligned}
\boldsymbol{f}(\boldsymbol{x}) &= \boldsymbol{W}_2\rho_r(\boldsymbol{W}_1\boldsymbol{x}) \\
&= \sum_{i=1}^{d_1} \boldsymbol{u}_i(\boldsymbol{v}_i^\mathsf{T}\boldsymbol{x})^r \\
&= \sum_{i=1}^{d_1} \boldsymbol{u}_i(\boldsymbol{v}_i^{\otimes r})^\mathsf{T}\boldsymbol{x}^{\otimes r} \\
&= \underbrace{\boldsymbol{W}_2\big(\boldsymbol{W}_1^\mathsf{T} \odot \cdots \odot \boldsymbol{W}_1^\mathsf{T}\big)^\mathsf{T}}_{=\boldsymbol{F}^{(1)}}\boldsymbol{x}^{\otimes r},
\end{aligned}$$

where $\odot$ denotes the Khatri-Rao product. The equivalence of the last expression and the first unfolding of the order-$(r+1)$ tensor $\mathcal{F}$ can be found in [14].

**The special case $d_2 = 1$.** When $d_2 = 1$, the columns of $\boldsymbol{W}_2 \in \mathbb{R}^{1 \times d_1}$ are scalars values $u_i \in \mathbb{R}$, $i = 1, \ldots, d_1$. In this case, $\big(\boldsymbol{W}_1^\mathsf{T} \odot \cdots \odot \boldsymbol{W}_1^\mathsf{T}\big)\boldsymbol{W}_2^\mathsf{T}$ becomes equivalent to the vectorization of $\mathcal{F}$, which is a fully symmetric tensor of order $r$ with factors $\boldsymbol{W}_1^\mathsf{T}$ and coefficients $[\boldsymbol{W}_2]_{1,i}$, $i = 1, \ldots, d_1$. $\qquad\square$

### B.3 Kruskal-based conditions for the uniqueness and identifiability of 2-layer networks

#### B.3.1 Sufficient conditions for uniqueness

The direct links between 2-layer ($L = 2$) hPNNs and partially symmetric CPDs in Proposition 34 allows us to obtain sufficient conditions for their uniqueness by means of Kruskal-based uniqueness results for the CPD, which we recall in the following lemma.

**Lemma B.1** (Kruskal's theorem, $s$-way version [82], Thm. 3). *Let $\mathcal{T} = [\![\boldsymbol{A}^{(1)}, \boldsymbol{A}^{(2)}, \cdots, \boldsymbol{A}^{(s)}]\!]$ the $R$-term CPD with $\boldsymbol{A}^{(i)} \in \mathbb{R}^{m_i \times R}$, such that*

$$\sum_{i=1}^{s} \mathrm{krank}\{\boldsymbol{A}^{(i)}\} \geq 2R + (s-1). \tag{12}$$

*Then the CP decomposition of $\mathcal{T}$ is unique up to permutation and scaling ambiguities, that is, for any alternative CPD $\mathcal{T} = [\![\widetilde{\boldsymbol{A}}^{(1)}, \widetilde{\boldsymbol{A}}^{(2)}, \cdots, \widetilde{\boldsymbol{A}}^{(s)}]\!]$, there exist a permutation matrix $\boldsymbol{\Pi}$ and invertible diagonal matrices $\boldsymbol{\Lambda}_1, \boldsymbol{\Lambda}_2, \ldots, \boldsymbol{\Lambda}_s$ such that*

$$\widetilde{\boldsymbol{A}}^{(i)} = \boldsymbol{A}^{(i)}\boldsymbol{\Pi}\boldsymbol{\Lambda}_i,$$

*for $i = 1, \ldots, s$.*

---

[10]In the definition of $\boldsymbol{f}$, the tensor $\boldsymbol{x}^{\otimes r}$ is viewed as a $d_0^r$ vector when multiplied by $\boldsymbol{F}^{(1)}$.

Now we prove Proposition 35 giving sufficient conditions for uniqueness in the case $L = 2$.

**Proposition 35.** *Let $p_w(x) = W_2 \rho_{r_1}(W_1 x)$ be a 2-layer hPNN with $W_1 \in \mathbb{R}^{d_1 \times d_0}$ and $W_2 \in \mathbb{R}^{d_2 \times d_1}$ and layer sizes $(d_0, d_1, d_2)$ satisfying $d_0, d_1 \geq 2$, $d_2 \geq 1$. Assume that $r \geq 2$, $\mathrm{krank}\{W_2\} \geq 1$, $\mathrm{krank}\{W_1^\mathsf{T}\} \geq 2$ and that:*

$$r \geq \frac{2d_1 - \mathrm{krank}\{W_2\}}{\mathrm{krank}\{W_1^\mathsf{T}\} - 1},$$

*then the 2-layer hPNN $p_w(x)$ is unique (or equivalently, the CPD of $\mathcal{F}$ in (6) is unique).*

*Proof of Proposition 35.* One can apply Proposition 34 to show that the 2-layer hPNN $p_w(x)$ is in one-to-one correspondence with the order $r + 1$ partially symmetric tensor

$$\mathcal{F} = [\![W_2, W_1^\mathsf{T}, \cdots, W_1^\mathsf{T}]\!], \tag{13}$$

thus, the uniqueness of $\mathrm{hPNN}_r[W_1, W_2]$ is equivalent to that of the CP-decomposition of $\mathcal{F}$ in (13). By Lemma B.1, the $d_1$-tem CP decomposition of $\mathcal{T}$ is unique provided that

$$\mathrm{krank}\{W_2\} + r\,\mathrm{krank}\{W_1^\mathsf{T}\} \geq 2d_1 + r.$$

By noting that $\mathrm{krank}\{W_1^\mathsf{T}\} > 1$ and rearranging the terms, we obtain the desired result. $\qquad\square$

Note that for the case of $d_0 \geq 2$ (i.e., hPNNs with at least two outputs), Proposition 35 gives conditions that may hold for quadratic activation degrees $r \geq 2$. On the other hand, for networks with a single output (i.e., $d_2 = 1$), it requires $r \geq 3$.

### B.3.2 Sufficient conditions for identifiability

Equipped with the sufficient conditions for the uniqueness of 2-layer hPNNs obtained in Proposition 35, we can now prove the generic identifiability result stated in Proposition 12.

**Proposition 12.** *Let $d_0, d_1 \geq 2$, $d_2 \geq 1$ be the layer widths and $r \geq 2$ such that*

$$r \geq \frac{2d_1 - \min(d_1, d_2)}{\min(d_1, d_0) - 1}.$$

*Then the 2-layer hPNN with architecture $((d_0, d_1, d_2), (r))$ is globally identifiable.*

*Proof of Proposition 12.* For general matrices $W_1 \in \mathbb{R}^{d_1 \times d_0}$ and $W_2 \in \mathbb{R}^{d_2 \times d_1}$, we have

$$\begin{aligned}
\mathrm{krank}\{W_1^\mathsf{T}\} &= \min(d_0, d_1), \\
\mathrm{krank}\{W_2\} &= \min(d_2, d_1).
\end{aligned}$$

Moreover, $d_0, d_1 \geq 2$, $d_2 \geq 1$ implies that generically $\mathrm{krank}\{W_1^\mathsf{T}\} \geq 2$ and $\mathrm{krank}\{W_2\} \geq 1$. This along with (4) means that the assumptions in Proposition 35 are satisfied generically (for all parameters except for a set of Lebesgue measure zero). Thus, the hPNN with architecture $((d_0, d_1, d_2), (r))$ is globally identifiable. $\qquad\square$

## C  Proof of the localization theorem

This appendix contains the main proofs of the localization theorem (Theorem 11) for deep hPNNs, as well as supporting lemmas and auxiliary technical results. We also provide proofs of the corollaries that specialize this result for several choices of architectures (e.g., pyramidal, bottleneck) and to the activation thresholds, discussed in Section 3.2 of the main paper.

**Results from the main paper**: Theorem 11, Corollaries 16, 19, and 17.

**Roadmap of the proof:** The proof of the localization theorem requires some setup. The main idea, as briefly sketched in Section 4.3 of the main paper, is to construct a recursion for Jacobian of the parameter map, and to certify that it has maximal rank (generically). This relies crucially on the properties of the neurovarieties associated to an hPNN as explained in Appendix A, in particular on Proposition 10 and Lemma A.4, which link the the finite identifiability of the hPNN to the rank of its Jacobian. The proof of the main result is presented towards the end of this appendix, in Appendix C.7, and proceeds by induction. However, it requires several technical tools which are build in the subsections that precede it.

- Appendix C.1 starts with some preparatory results on the rank of the Jacobian of a 2-layer hPNN, setting the base case.
- Appendix C.2 defines the so-called *last layer map* (i.e., the map that composes a $d_0$-variate polynomial with one hPNN layer) and illustrates the structure of its Jacobian by means of a detailed example.
- Appendix C.3 presents a key proposition which establishes a *certificate* to show that the Jacobian of the last layer map has maximal rank, and before proceeding to the proof, illustrates the result with an example.
- Appendix C.4 introduces some additional notation and setup which will be used in the proof of the key proposition.
- Appendix C.5 presents the proof of the key proposition for the special case when the number of input variables $d_0$ is equal to the number of variables used in the certificate (equal to the smallest bottleneck in the network).
- Appendix C.6 gives the proof of the key proposition in the general case when the number of input neurons $d_0$ can be larger than the number of variables the certificate.
- Appendix C.7 contains the proof of the localization theorem.
- Finally, Appendix C.8 presents the proofs for the results concerning the implications of the localization theorem to different hPNN architectures.

**Simplifying the notation:** In Appendices C.1 to C.4, we denote the number of input neurons by $m$, the number of hidden neurons in the second-to-last layer by $d$, and the number of output neurons as $n$. For two-layer networks, we denote the first- and second-layer weight matrices by $\boldsymbol{V}$ and $\boldsymbol{W}$, respectively.

## C.1 Preparatory lemmas - rank of Jacobian of a 2-layer PNN

**Lemma C.1.** *Let $(m, d, n)$ and $r$ be such that the 2-layer hPNN with architecture $((m, d, n), r)$ is finitely identifiable (resp. the partially symmetric $d$-term CPD is generically unique). Then for general matrices $\boldsymbol{V}, \boldsymbol{W}$ the Jacobian of the map $\varphi(\boldsymbol{V}, \boldsymbol{W}) = \mathrm{hPNN}_r[(\boldsymbol{V}, \boldsymbol{W})]$, given by*

$$J_\varphi = J_\varphi(\boldsymbol{V}, \boldsymbol{W}) = \begin{bmatrix} J_\varphi^{(\boldsymbol{V})} & J_\varphi^{(\boldsymbol{W})} \end{bmatrix},$$

*has maximal possible rank:*

$$\mathrm{rank}\{J_\varphi\} = (m + n - 1)d, \tag{14}$$

*and also its submatrix containing the derivatives with respect to elements of $\boldsymbol{V}$ is full column rank:*

$$\mathrm{rank}\{J_\varphi^{(\boldsymbol{V})}\} = md. \tag{15}$$

*Proof.* The first statement follows from dimension of the neurovariety (that is, $(m + n - 1)d$), and the second statement follows from the fact that the subset of pairs $(\boldsymbol{V}, \boldsymbol{W})$ with $\boldsymbol{W}$ given as

$$\boldsymbol{W} = \begin{bmatrix} 1 \cdots 1 \\ \overline{\boldsymbol{W}} \end{bmatrix}, \quad \overline{\boldsymbol{W}} \in \mathbb{R}^{(n-1) \times d}$$

parameterizes an open subset of the neurovariety (i.e., due to the scaling ambiguity, almost any pair of $\boldsymbol{V}$ and $\boldsymbol{W}$ can be reduced to such a form). As shown in the proof of Proposition 10 (specialized to $(\boldsymbol{W}_1, \boldsymbol{W}_2) = (\boldsymbol{V}, \boldsymbol{W})$), the reduced Jacobian is full column rank:

$$\mathrm{rank}\{\begin{bmatrix} J_\varphi^{(\boldsymbol{V})} & J_\varphi^{(\overline{\boldsymbol{W}})} \end{bmatrix}\} = md + (n-1)d,$$

where $J_\varphi^{(\overline{W})}$ denotes the Jacobian with respect to $\overline{W}$. Note tnis implies that all the submatrices are full column rank and, as therefore $J_\varphi^{(V)}$ is full column rank.

$\square$

**Remark C.2.** *The conditions in Lemma C.1 are satisfied, for example, if the Kruskal-based generic uniqueness conditions are satisfied (see Proposition 12).*

Before giving the elements of the main proof, we provide an example of explicit Jacobian computation for the map $\mathrm{hPNN}_{d,r}[\cdot]$ which will be the guiding example for the proof of identifiability.

**Example C.3** (Simplest architecture)**.** *Consider example $(m, d, n) = (2, 2, 2)$, $r = 2$, and denote the elements of $V$ and $W$ as*

$$V = \begin{bmatrix} \alpha_1 & \beta_1 \\ \alpha_2 & \beta_2 \end{bmatrix}, \quad W = \begin{bmatrix} W_{1,1} & W_{1,2} \\ W_{2,1} & W_{2,2} \end{bmatrix}.$$

*so the hPNN map $\varphi(V, W) = \mathrm{hPNN}_{d,r}[V, W]$ is given by*

$$\varphi(V, W) = w_1(\alpha_1 x_1 + \beta_1 x_2)^2 + w_2(\alpha_2 x_1 + \beta_2 x_2)^2,$$

*where $w_1, w_2$ denote the columns of the matrix $W$:*

$$w_1 = \begin{bmatrix} W_{1,1} \\ W_{2,1} \end{bmatrix}, \quad w_2 = \begin{bmatrix} W_{1,2} \\ W_{2,2} \end{bmatrix}.$$

*The image of $\varphi$ lives in the space of vector polynomials $(\mathscr{H}_{2,2})^{\times 2}$ (of dimension 6), therefore, the blocks of the Jacobian $J_\varphi^{(V)}$ and $J_\varphi^{(W)}$ are of sizes $6 \times 4$. The matrix $J_\varphi^{(V)}$ has as its columns derivatives with respect to $\alpha_j$ and $\beta_j$, for $j \in \{1, 2\}$ which are, respectively:*

$$\frac{\partial \varphi}{\partial \alpha_j} = 2w_j x_1(\alpha_j x_1 + \beta_1 x_2), \quad \frac{\partial \varphi}{\partial \beta_j} = 2w_j x_2(\alpha_j x_1 + \beta_1 x_2). \tag{16}$$

*Let us choose the canonical basis of $(\mathscr{H}_{2,2})^{\times 2}$ as $e_i x_1^{2-\ell} x_2^\ell$, $i \in \{1, 2\}$, $\ell \in \{0, 1, 2\}$, where $e_i$ are unit vectors. Then the block $J_\varphi^{(V)}$ is represented in the matrix form as:*

$$J_\varphi^{(V)} = (2) \cdot \begin{array}{c} e_1 x_1^2 \\ e_1 x_1 x_2 \\ e_1 x_2^2 \\ e_2 x_1^2 \\ e_2 x_1 x_2 \\ e_2 x_2^2 \end{array} \begin{bmatrix} \overset{\frac{\partial \varphi}{\partial \alpha_1}}{W_{1,1}\alpha_1} & \overset{\frac{\partial \varphi}{\partial \beta_1}}{0} & \overset{\frac{\partial \varphi}{\partial \alpha_2}}{W_{1,2}\alpha_2} & \overset{\frac{\partial \varphi}{\partial \beta_2}}{0} \\ W_{1,1}\beta_1 & W_{1,1}\alpha_1 & W_{1,2}\beta_2 & W_{1,2}\alpha_2 \\ 0 & W_{1,1}\beta_1 & 0 & W_{1,2}\beta_2 \\ W_{2,1}\alpha_1 & 0 & W_{2,2}\alpha_2 & 0 \\ W_{2,1}\beta_1 & W_{2,1}\alpha_1 & W_{2,2}\beta_2 & W_{2,2}\alpha_2 \\ 0 & W_{2,1}\beta_1 & 0 & W_{2,2}\beta_2 \end{bmatrix}$$

*The block $J_\varphi^{(W)}$ contains the derivatives with respect to $W_{i,j}$, for $i, j \in \{1, 2\}$, which are:*

$$\frac{\partial \varphi}{\partial W_{i,j}} = e_i(\alpha_j x_1 + \beta_j x_2)^2. \tag{17}$$

*In the same monomial basis, the matrix can be expressed as*

$$J_\varphi^{(W)} = \begin{array}{c} e_1 x_1^2 \\ e_1 x_1 x_2 \\ e_1 x_2^2 \\ e_2 x_1^2 \\ e_2 x_1 x_2 \\ e_2 x_2^2 \end{array} \begin{bmatrix} \overset{\frac{\partial \varphi}{\partial W_{1,1}}}{\alpha_1^2} & \overset{\frac{\partial \varphi}{\partial W_{2,1}}}{0} & \overset{\frac{\partial \varphi}{\partial \alpha_2}}{\alpha_2^2} & \overset{\frac{\partial \varphi}{\partial \beta_2}}{0} \\ 2\alpha_1\beta_1 & 0 & 2\alpha_2\beta_2 & 0 \\ \beta_1^2 & 0 & \beta_2^2 & 0 \\ 0 & \alpha_1^2 & 0 & \alpha_2^2 \\ 0 & 2\alpha_1\beta_1 & 0 & 2\alpha_1\beta_2 \\ 0 & \beta_1^2 & 0 & \beta_2^2 \end{bmatrix}$$

**Remark C.4.** *It is easy to show why (14) and (15) are satistfied for the architecture in Example C.3. For this example, we choose particular $V$ and $W$ to be identity matrices, which gives us*

$$\begin{bmatrix} J_\varphi^{(V)} & J_\varphi^{(W)} \end{bmatrix} = \begin{bmatrix} 2 & 0 & 0 & 0 & 1 & 0 & 0 & 0 \\ 0 & 2 & 0 & 0 & 0 & 0 & 0 & 0 \\ 0 & 0 & 0 & 0 & 0 & 0 & 1 & 0 \\ 0 & 0 & 0 & 0 & 0 & 1 & 0 & 0 \\ 0 & 0 & 2 & 0 & 0 & 0 & 0 & 0 \\ 0 & 0 & 0 & 2 & 0 & 0 & 0 & 1 \end{bmatrix}.$$

*It is easy to see that the left block (matrix $J_\varphi^{(V)}$) has rank 4, and the total matrix has rank $6 = (2 + 2 - 1)2$. Therefore, by Corollaries A.6 and A.10, (14) and (15) are satisfied generically.*

We will also need an explicit form of the Jacobian in the general case, which is a generalization of the expression in Example C.3.

**Remark C.5.** *Let $(m, d, n)$, $r$, $V$ and $W$ be as in Lemma C.1. With some abuse of notation we denote $\boldsymbol{v}_j \in \mathbb{R}^m$ and $\boldsymbol{w}_j \in \mathbb{R}^n$*

$$\boldsymbol{V}^\mathsf{T} = [\boldsymbol{v}_1 \quad \cdots \quad \boldsymbol{v}_d], \quad \boldsymbol{W} = [\boldsymbol{w}_1 \quad \cdots \quad \boldsymbol{w}_d],$$

*and let $\boldsymbol{z} = [z_1 \quad \cdots \quad z_m]^\mathsf{T}$ be the input variables. Then the hPNN $\varphi[\cdot] = \mathrm{hPNN}_{\boldsymbol{d},\boldsymbol{r}}[\cdot]$ has the form*

$$\varphi[\boldsymbol{V}, \boldsymbol{W}](\boldsymbol{z}) = \sum_{j=1}^d \boldsymbol{w}_j (\boldsymbol{v}_j^\mathsf{T} \boldsymbol{z})^r. \tag{18}$$

*Therefore, we have that derivatives with respect to the elements of the matrix $\boldsymbol{W}$ can be expressed as*

$$\frac{\partial \varphi}{\partial W_{i,j}} = \frac{\partial \varphi}{\partial (\boldsymbol{w}_j)_i} = \boldsymbol{e}_i (\boldsymbol{v}_j^\mathsf{T} \boldsymbol{z})^r, \tag{19}$$

*where $\boldsymbol{e}_i$ is the $i$-th unit vector in $\mathbb{R}^n$, and, with respect to elements of $\boldsymbol{V}$, we have*

$$\frac{\partial \varphi}{\partial V_{j,\ell}} = \frac{\partial \varphi}{\partial (\boldsymbol{v}_j)_\ell} = r z_\ell \cdot \boldsymbol{w}_j (\boldsymbol{v}_j^\mathsf{T} \boldsymbol{z})^{r-1}. \tag{20}$$

*Note that Lemma C.1 concerns the dimensions of linear spaces spanned by the sets of polynomials in (19)–(20). Also, (20) and (19) are generalizations of (16) and (17), respectively.*

## C.2 Jacobian of composition of polynomial maps

The goal of this subsection, is to exhibit the structure of the composition of polynomial NN-like maps and their Jacobians. Consider an outer layer of an hPNN, which is denoted as

$$\boldsymbol{W} \rho_r(q_1, ..., q_d).$$

In order to see what happens when we substitute variables $q_1, \ldots, q_d$ by $d_0$-variate polynomials $\boldsymbol{q}(x_1, \ldots, x_{d_0}) \in (\mathscr{H}_{d_0,R})^{\times d}$, we introduce the following definition (which corresponds to (7)):

**Definition C.6** (Last layer map). *Let $\boldsymbol{W} \in \mathbb{R}^{n \times d}$ be $n \times d$ matrix $r \in \mathbb{N}$. We define the map $\psi$ that transforms a vector of $R$-degree $d_0$-variate polynomial as follows:*

$$\psi : (\mathscr{H}_{d_0,R})^{\times d} \times \mathbb{R}^{n \times d} \to (\mathscr{H}_{d_0,Rr})^{\times n}$$
$$(\boldsymbol{q}(x_1, \ldots, x_{d_0}), \boldsymbol{W}) \mapsto \psi[\boldsymbol{q}, \boldsymbol{W}] := \boldsymbol{W} \rho_r(\boldsymbol{q}(x_1, \ldots, x_{d_0})),$$

*and denote the Jacobian with respect to the parameters (and its blocks) as*

$$J_\psi(\boldsymbol{q}, \boldsymbol{W}) = \begin{bmatrix} J_\psi^{(\boldsymbol{q})} & J_\psi^{(\boldsymbol{W})} \end{bmatrix},$$

*where $J_\psi^{(\boldsymbol{q})}$ has $d\binom{R+d_0-1}{R}$ columns and $J_\psi^{(\boldsymbol{W})}$ has $nd$ columns.*

**Example C.7.** *Similarly to Example C.3, we take the case $n = 2$, $d = 2$, $r = 2$, and denote $\boldsymbol{W} = [\boldsymbol{w}_1 \quad \boldsymbol{w}_2]$. Then the last layer map becomes*

$$\psi(q_1, q_2) = \boldsymbol{w}_1 q_1^2 + \boldsymbol{w}_2 q_2^2.$$

*Consider a special case $d_0 = 2$, $R = 2$ so that $\psi$ maps $(q_1, q_2) \in (\mathscr{H}_{2,2})^{\times 2}$ to a vector polynomial in $(\mathscr{H}_{2,4})^{\times 2}$, and let the input polynomials be parameterized as*

$$q_j(x_1, x_2) = q_j^{(2,0)} x_1^2 + 2 q_j^{(1,1)} x_1 x_2 + q_j^{(0,2)} x_2^2, \quad j \in \{1, 2\},$$

*where $q_j^{(i_1, i_2)}$, $(i_1, i_2) \in \{(2,0), (1,1), (0,2)\}$ are the coefficient of $q_j$ next to monomials $x_1^{i_1} x_2^{i_2}$. Then the Jacobian $J_\psi(\boldsymbol{q}, \boldsymbol{W})$ is a $10 \times 10$ matrix[11], whose blocks are described below.*

---

[11] since $\dim(\mathscr{H}_{2,4}) = 5$.

The block $J_\psi^{(q)}$ is a $10 \times 6$ matrix, whose columns are the 6 polynomials (similarly to (16)):

$$\frac{\partial \psi}{\partial q_j^{(i_1,i_2)}} = 2\boldsymbol{w}_j x_1^{i_1} x_2^{i_2}\big(q_j(x_1,x_2)\big), \quad j \in \{1,2\}, (i_1,i_2) \in \{(2,0),(1,1),(0,2)\}. \quad (21)$$

In the canonical basis is given as $J_\psi^{(q)} =$

$$(2) \cdot
\begin{array}{c}
\boldsymbol{e}_1 x_1^4 \\
\boldsymbol{e}_1 x_1^3 x_2 \\
\boldsymbol{e}_1 x_1^2 x_2^2 \\
\boldsymbol{e}_1 x_1 x_2^3 \\
\boldsymbol{e}_1 x_2^4 \\
\boldsymbol{e}_2 x_1^4 \\
\boldsymbol{e}_2 x_1^3 x_2 \\
\boldsymbol{e}_2 x_1^2 x_2^2 \\
\boldsymbol{e}_2 x_1 x_2^3 \\
\boldsymbol{e}_2 x_2^4
\end{array}
\begin{bmatrix}
W_{1,1}q_1^{(2,0)} & 0 & 0 & W_{1,2}q_2^{(2,0)} & 0 & 0 \\
W_{1,1}q_1^{(1,1)} & W_{1,1}q_1^{(2,0)} & 0 & W_{1,2}q_2^{(1,1)} & W_{1,2}q_2^{(2,0)} & 0 \\
W_{1,1}q_1^{(0,2)} & W_{1,1}q_1^{(1,1)} & W_{1,1}q_1^{(2,0)} & W_{1,2}q_2^{(0,2)} & W_{1,2}q_2^{(1,1)} & W_{1,2}q_2^{(2,0)} \\
0 & W_{1,1}q_1^{(0,2)} & W_{1,1}q_1^{(1,1)} & 0 & W_{1,2}q_2^{(0,2)} & W_{1,2}q_2^{(1,1)} \\
0 & 0 & W_{1,1}q_1^{(0,2)} & 0 & 0 & W_{1,2}q_2^{(0,2)} \\
W_{2,1}q_1^{(2,0)} & 0 & 0 & W_{2,2}q_2^{(2,0)} & 0 & 0 \\
W_{2,1}q_1^{(1,1)} & W_{2,1}q_1^{(2,0)} & 0 & W_{2,2}q_2^{(1,1)} & W_{2,2}q_2^{(2,0)} & 0 \\
W_{2,1}q_1^{(0,2)} & W_{2,1}q_1^{(1,1)} & W_{2,1}q_1^{(2,0)} & W_{2,2}q_2^{(0,2)} & W_{2,2}q_2^{(1,1)} & W_{2,2}q_2^{(2,0)} \\
0 & W_{2,1}q_1^{(0,2)} & W_{2,1}q_1^{(1,1)} & 0 & W_{2,2}q_2^{(0,2)} & W_{2,2}q_2^{(1,1)} \\
0 & 0 & W_{2,1}q_1^{(0,2)} & 0 & 0 & W_{2,2}q_2^{(0,2)}
\end{bmatrix}$$

with column headers $\frac{\partial \psi}{\partial q_1^{(2,0)}}, \frac{\partial \psi}{\partial q_1^{(1,1)}}, \frac{\partial \psi}{\partial q_1^{(0,2)}}, \frac{\partial \psi}{\partial q_2^{(2,0)}}, \frac{\partial \psi}{\partial q_2^{(1,1)}}, \frac{\partial \psi}{\partial q_2^{(0,2)}}$.

The second block, similarly to (17), is a $10 \times 4$ matrix whose columns are

$$\frac{\partial \psi}{\partial W_{i,j}} = \boldsymbol{e}_i\big(q_j(x_1,x_2)\big)^2, \quad i,j \in \{1,2\}, \quad (22)$$

and has similar structure to that $J_\varphi^{(W)}$ in Example C.3 (it will be explicitly shown in the next example).

### C.3   A certificate of maximal rank for the Jacobian of the last layer

The following proposition gives a condition for when the Jacobian of the last layer map has maximal rank, based on constructing a certificate.

**Proposition C.8** (Certificate of last layer map). *Let $m, d, n$ and $r \geq 2$ be fixed, and the matrices $\boldsymbol{V} \in \mathbb{R}^{d \times m}$ and $\boldsymbol{W} \in \mathbb{R}^{n \times d}$ be such that the equalities (14)–(15) are satisfied. Fix $d_0 \geq m$, $R \geq 2$, and consider the polynomial vector $\widehat{\boldsymbol{q}}(x_1, \ldots, x_m) \in (\mathscr{H}_{m,R})^{\times d} \subseteq (\mathscr{H}_{d_0,R})^{\times d}$ defined as*

$$\widehat{\boldsymbol{q}}(\boldsymbol{x}) := \boldsymbol{V} \begin{bmatrix} x_1^R \\ x_2^R \\ \vdots \\ x_m^R \end{bmatrix}. \quad (23)$$

*Then we have that the evaluation of the Jacobian of the last layer map $\psi$ (see Definition C.6) at the point $(\widehat{\boldsymbol{q}}, \boldsymbol{W})$ is of maximal possible rank:*

$$\mathrm{rank}\{J_\psi(\widehat{\boldsymbol{q}}, \boldsymbol{W})\} = d(n-1) + d\binom{d_0 + R - 1}{R} \quad (24)$$

*and its submatrix containing derivatives with respect to $\boldsymbol{q}$ is full column rank*

$$\mathrm{rank}\{J_\psi^{(q)}(\widehat{\boldsymbol{q}}, \boldsymbol{W})\} = d\binom{d_0 + R - 1}{R}. \quad (25)$$

Before proving Proposition C.8, we give an illustrative example of the Jacobian of the last layer map evaluated at the certificate $\widehat{\boldsymbol{q}}$.

**Example C.9** (Example C.3, continued). *We continue Examples C.3 and C.7. In this case, the vector polynomial $\widehat{\boldsymbol{q}}$ from Proposition C.8 reads*

$$\widehat{\boldsymbol{q}}(x_1, x_2) = \begin{bmatrix} \widehat{q}_1(x_1, x_2) \\ \widehat{q}_2(x_1, x_2) \end{bmatrix} = \begin{bmatrix} (\alpha_1 x_1^2 + \beta_1 x_2^2) \\ (\alpha_2 x_1^2 + \beta_2 x_2^2) \end{bmatrix},$$

*i.e., using the notation of Example C.3, the coefficients of the polynomials are*

$$(\widehat{q}_1^{(2,0)}, \widehat{q}_1^{(1,1)}, \widehat{q}_1^{(0,2)}) = (\alpha_1, 0, \beta_1),$$
$$(\widehat{q}_2^{(2,0)}, \widehat{q}_2^{(1,1)}, \widehat{q}_2^{(0,2)}) = (\alpha_2, 0, \beta_2).$$

*Specializing Example C.7 (and removing factor $2$ for simlicity), we get*

$$\frac{1}{2} J_\psi^{(q)}(\widehat{q}, W) = \begin{array}{c} \\ e_1 x_1^4 \\ e_1 x_1^3 x_2 \\ e_1 x_1^2 x_2^2 \\ e_1 x_1 x_2^3 \\ e_1 x_2^4 \\ e_2 x_1^4 \\ e_2 x_1^3 x_2 \\ e_2 x_1^2 x_2^2 \\ e_2 x_1 x_2^3 \\ e_2 x_2^4 \end{array} \begin{bmatrix} \frac{\partial \psi}{\partial q_1^{(2,0)}} & \frac{\partial \psi}{\partial q_1^{(1,1)}} & \frac{\partial \psi}{\partial q_1^{(0,2)}} & \frac{\partial \psi}{\partial q_2^{(2,0)}} & \frac{\partial \psi}{\partial q_2^{(1,1)}} & \frac{\partial \psi}{\partial q_2^{(0,2)}} \\ W_{1,1}\alpha_1 & 0 & 0 & W_{1,2}\alpha_2 & 0 & 0 \\ 0 & W_{1,1}\alpha_1 & 0 & 0 & W_{1,2}\alpha_2 & 0 \\ W_{1,1}\beta 1_1 & 0 & W_{1,1}\alpha_1 & W_{1,2}\beta_2 & 0 & W_{1,2}\alpha_2 \\ 0 & W_{1,1}\beta_1 & 0 & 0 & W_{1,2}\beta_2 & 0 \\ 0 & 0 & W_{1,1}\beta_1 & 0 & 0 & W_{1,2}\beta_2 \\ W_{2,1}\alpha_1 & 0 & 0 & W_{2,2}\alpha_2 & 0 & 0 \\ 0 & W_{2,1}\alpha_1 & 0 & 0 & W_{2,2}\alpha_2 & 0 \\ W_{2,1}\beta_1 & 0 & W_{2,1}\alpha_1 & W_{2,2}\beta_2 & 0 & W_{2,2}\alpha_2 \\ 0 & W_{2,1}\beta_1 & 0 & 0 & W_{2,2}\beta_2 & 0 \\ 0 & 0 & W_{2,1}\beta_1 & 0 & 0 & W_{2,2}\beta_2 \end{bmatrix}.$$

*The matrix $J_\psi^{(W)}$ then, according to (22), becomes*

$$J_\psi^{(W)}(\widehat{q}, W) = \begin{array}{c} \\ e_1 x_1^4 \\ e_1 x_1^3 x_2 \\ e_1 x_1^2 x_2^2 \\ e_1 x_1 x_2^3 \\ e_1 x_2^4 \\ e_1 x_1^4 \\ e_1 x_1^3 x_2 \\ e_1 x_1^2 x_2^2 \\ e_1 x_1 x_2^3 \\ e_1 x_2^4 \end{array} \begin{bmatrix} \frac{\partial \psi}{\partial W_{1,1}} & \frac{\partial \psi}{\partial W_{2,1}} & \frac{\partial \psi}{\partial W_{1,2}} & \frac{\partial \psi}{\partial W_{1,2}} \\ \alpha_1^2 & 0 & \alpha_2^2 & 0 \\ 0 & 0 & 0 & 0 \\ 2\alpha_1\beta_1 & 0 & 2\alpha_2\beta_2 & 0 \\ 0 & 0 & 0 & 0 \\ \beta_1^2 & 0 & \beta_2^2 & 0 \\ 0 & \alpha_1^2 & 0 & \alpha_2^2 \\ 0 & 0 & 0 & 0 \\ 0 & 2\alpha_1\beta_1 & 0 & 2\alpha_2\beta_2 \\ 0 & 0 & 0 & 0 \\ 0 & \beta_1^2 & 0 & \beta_2^2 \end{bmatrix}.$$

*The crux of the proof of Proposition C.8 is the following observation. If we stack together matrices $J = \begin{bmatrix} \frac{1}{2} J_\psi^{(q)} & J_\psi^{(W)} \end{bmatrix}$ and permute the rows and columns as follows, we get the block-diagonal matrix*

$$J = \begin{array}{c} \\ e_1 x_1^4 \\ e_1 x_1^2 x_2^2 \\ e_1 x_2^4 \\ e_2 x_1^4 \\ e_2 x_1^2 x_2^2 \\ e_2 x_2^4 \\ \hline e_1 x_1^3 x_2 \\ e_1 x_1 x_2^3 \\ e_2 x_1^3 x_2 \\ e_2 x_1 x_2^3 \end{array} \left[ \begin{array}{cccccccc|cc} \frac{\partial \psi}{\partial q_1^{(2,0)}} & \frac{\partial \psi}{\partial q_1^{(0,2)}} & \frac{\partial \psi}{\partial q_2^{(2,0)}} & \frac{\partial \psi}{\partial q_2^{(0,2)}} & \frac{\partial \psi}{\partial W_{1,1}} & \frac{\partial \psi}{\partial W_{2,1}} & \frac{\partial \psi}{\partial W_{1,2}} & \frac{\partial \psi}{\partial W_{1,2}} & \frac{\partial \psi}{\partial q_1^{(1,1)}} & \frac{\partial \psi}{\partial q_2^{(1,1)}} \\ W_{1,1}\alpha_1 & 0 & W_{1,2}\alpha_2 & 0 & \alpha_1^2 & 0 & \alpha_2^2 & 0 & & \\ W_{1,1}\beta_1 & W_{1,1}\alpha_1 & W_{1,2}\beta_2 & W_{1,2}\alpha_2 & 2\alpha_1\beta_1 & 0 & 2\alpha_2\beta_2 & 0 & & \\ 0 & W_{1,1}\beta_1 & 0 & W_{1,2}\beta_2 & \beta_1^2 & 0 & \beta_2^2 & 0 & & 0 \\ W_{2,1}\alpha_1 & 0 & W_{2,2}\alpha_2 & 0 & 0 & \alpha_1^2 & 0 & \alpha_2^2 & & \\ W_{2,1}\beta_1 & W_{2,1}\alpha_1 & W_{2,2}\beta_2 & W_{2,2}\alpha_2 & 0 & 2\alpha_1\beta_1 & 0 & 2\alpha_1\beta_2 & & \\ 0 & W_{2,1}\beta_1 & 0 & W_{2,2}\beta_2 & 0 & \beta_1^2 & 0 & \beta_2^2 & & \\ \hline & & & & & & & & W_{1,1}\alpha_1 & W_{1,2}\alpha_2 \\ & & & 0 & & & & & W_{1,1}\beta_1 & W_{1,2}\beta_2 \\ & & & & & & & & W_{2,1}\alpha_1 & W_{2,2}\alpha_2 \\ & & & & & & & & W_{2,1}\beta_1 & W_{2,2}\beta_2 \end{array} \right].$$

*We see that the top-left block of the matrix $J$ is nothing but the matrix*

$$\begin{bmatrix} \frac{1}{2} J_\varphi^{(V)} & J_\varphi^{(W)} \end{bmatrix},$$

*where $\varphi$ is as in Example C.3, thus it has rank $6$, and its left $4$ columns are linearly independent. Moreover, its bottom-right block can be viewed as submatrix $\frac{1}{2} J_\varphi^{(V)}$ (taking first and third columns, for instance), and therefore has full column rank $2$.*

*Thus matrix $J_\psi$ has rank $8 = 6 + 2$ and $J_\psi^{(q)}$ has rank $6 = 4 + 2$.*

## C.4 Extra notation for the proof of the proposition

In order to prove Proposition C.8 we introduce extra notation for the columns of $J_\psi$. We first let $\boldsymbol{W} = [\boldsymbol{w}_1 \quad \cdots \quad \boldsymbol{w}_d]$ as in Remark C.5, so we can express

$$\psi[\boldsymbol{q}, \boldsymbol{W}] = \sum_{j=1}^{d} \boldsymbol{w}_j (q_j)^r.$$

Already this, similarly to (19) gives us

$$\frac{\partial}{\partial W_{i,j}} \psi = \frac{\partial}{\partial (\boldsymbol{w}_j)_i} \psi = \boldsymbol{e}_i (q_j)^r,$$

and we denote the linear space spanned by these polynomials (i.e., the range of $J_\psi^{(\boldsymbol{W})}$) as

$$\mathcal{L}^{(\boldsymbol{W})} = \operatorname{span}\left\{ \frac{\partial}{\partial W_{i,j}} \psi \right\}_{i,j=1}^{n,d} = \operatorname{range}\{J_\psi^{(\boldsymbol{W})}\}.$$

Now we look into details of the structure of the matrix $J_\psi^{(\boldsymbol{q})}$. Let $\boldsymbol{i} = (i_1, \ldots, i_{d_0}) \in \mathcal{I}$ be a multi-index that runs over

$$\mathcal{I} = \{\boldsymbol{i} := (i_1, \ldots, i_{d_0}) : i_1, \ldots, i_{d_0} \geq 0 \text{ and } i_1 + \cdots + i_{d_0} = R\}$$

so that the coefficients of a polynomial $q \in \mathcal{H}_{d_0, r}$ can be numbered by the elements in $\mathcal{I}$ as

$$q(x_1, \ldots, x_{d_0}) = \sum_{\boldsymbol{i} \in \mathcal{I}} q^{(\boldsymbol{i})} x_1^{i_1} \ldots x_{d_0}^{i_{d_0}}.$$

Then, the columns of $J_\psi^{(\boldsymbol{q})}$ for $\boldsymbol{q}(\boldsymbol{x}) = [q_1(\boldsymbol{x}) \quad \cdots \quad q_d(\boldsymbol{x})]^\mathsf{T}$ are given by the polynomials

$$\boldsymbol{f}_{j,\boldsymbol{i}}(\boldsymbol{x}) := \frac{\partial \psi}{\partial q_j^{(\boldsymbol{i})}}(\boldsymbol{q}, \boldsymbol{W}) = (r x_1^{i_1} \cdots x_{d_0}^{i_{d_0}}) \boldsymbol{w}_j (q_j)^{r-1}, \quad j = 1, \ldots, d, \quad \boldsymbol{i} \in \mathcal{I}, \qquad (26)$$

which are precisely generalizations of (21). We denote the spaces spanned by such polynomials as

$$\mathcal{L}^{(\boldsymbol{q},\boldsymbol{i})} := \operatorname{span}\{\boldsymbol{f}_{j,\boldsymbol{i}}(\boldsymbol{x})\}_{j=1}^{d},$$

and their span (the range of $J_\psi^{(\boldsymbol{q})}$) as

$$\mathcal{L}^{(\boldsymbol{q})} = \operatorname{span}\{\mathcal{L}^{(\boldsymbol{q},\boldsymbol{i})}\}_{\boldsymbol{i} \in \mathcal{I}} = \operatorname{range}\{J_\psi^{(\boldsymbol{q})}\}.$$

**Example C.10.** *In notation of Example C.7,* $\mathcal{I} = \{(2,0),(1,1),(0,2)\}$. *In this case, we have*

$$\mathcal{L}^{(\boldsymbol{q},(2,0))} = \operatorname{span}\left\{ \frac{\partial \psi}{\partial q_1^{(2,0)}}, \frac{\partial \psi}{\partial q_2^{(2,0)}} \right\},$$

$$\mathcal{L}^{(\boldsymbol{q},(1,1))} = \operatorname{span}\left\{ \frac{\partial \psi}{\partial q_1^{(1,1)}}, \frac{\partial \psi}{\partial q_2^{(1,1)}} \right\},$$

$$\mathcal{L}^{(\boldsymbol{q},(0,2))} = \operatorname{span}\left\{ \frac{\partial \psi}{\partial q_1^{(0,2)}}, \frac{\partial \psi}{\partial q_2^{(0,2)}} \right\},$$

*which correspond to the columns* $\{1,4\}$, $\{2,5\}$, $\{3,6\}$, *respectively, of the matrix* $J_\psi^{(\boldsymbol{q})}$.

**Remark C.11.** *Proving Proposition C.8 (i.e., proving that (24)–(25) hold) is equivalent to showing that*

$$\dim \operatorname{span}\{\mathcal{L}^{(\boldsymbol{q})}, \mathcal{L}^{(\boldsymbol{W})}\} = d(n-1) + d\binom{d_0 + R - 1}{R}, \qquad (27)$$

$$\dim \mathcal{L}^{(\boldsymbol{q})} = d\binom{d_0 + R - 1}{R}, \qquad (28)$$

*respectively.*

*The strategy of proving that the dimensions of these subspaces are maximal is to show that the individual subspaces* $\mathcal{L}^{(\boldsymbol{q},\boldsymbol{i})}$ *are orthogonal under some conditions (which is similar to bringing* $\boldsymbol{J}$ *into the block-diagonal form in Example C.9).*

### C.5 Proof of the proposition on certificate: case $m = d_0$

We first prove the proposition for the case when the number of input variables $d_0$ is equal to the number of variables $m$ used in the certificate.

*Proof of Proposition C.8 (case $m = d_0$).* Recall that in the notation of the previous subsection we need to calculate
$$\dim \operatorname{span} \left( \mathcal{L}^{(W)}, \operatorname{span}\{\mathcal{L}^{(q,i)}\}_{i \in \mathcal{I}} \right).$$

Now let us consider these subspaces for a particular choice of $q = \widehat{q}$ of the form (23). We have that $f_{j,i}$ from (26) have the form
$$f_{j,(i_1,\ldots,i_m)}(x_1,\ldots,x_m) = \underbrace{( \quad \cdots \quad )}_{\text{polynomial in } x_1^R,\ldots,x_m^R} x_1^{i_1(\bmod R)} \ldots x_m^{i_m(\bmod R)}.$$

Therefore we get that $\mathcal{L}^{(q,i)} \perp \mathcal{L}^{(q,\ell)}$ unless one of the following conditions holds:
$$i = \ell \quad \text{or} \quad \{i,\ell\} \subset \mathcal{I}_0$$
with $\mathcal{I}_0 := \{(R,0,\ldots,0),(0,R,0,\ldots,0),\ldots,(0,0,\ldots,R)\}$. For the same reasons we get
$$\mathcal{L}^{(W)} \perp \mathcal{L}^{(q,i)} \text{ for all } i \in \mathcal{I} \setminus \mathcal{I}_0.$$

Therefore, we get
$$\operatorname{rank}\{J_\psi\} = \dim \operatorname{span} \left( \mathcal{L}^{(W)}, \operatorname{span}\{\mathcal{L}^{(q,i)}\}_{i \in \mathcal{I}_0} \right) + \sum_{i \in \mathcal{I} \setminus \mathcal{I}_0} \dim(\mathcal{L}^{(q,i)}).$$

Let us look at those dimensions separately. Denote $z = \begin{bmatrix} z_1 & \cdots & z_m \end{bmatrix}^\mathsf{T}$, with
$$z_1 = x_1^R, \quad \ldots, \quad z_m = x_m^R$$
so that for $\widehat{q}$ of the form (23) it holds
$$\widehat{q}_j = v_j^\mathsf{T} z.$$

Then, for $i \in \mathcal{I} \setminus \mathcal{I}_0$ it is easy to see that
$$\dim(\mathcal{L}^{(q,i)}) = \dim \operatorname{span} \left( \{w_j(\widehat{q}_j)^{r-1}\}_{j=1}^d \right) = d,$$

where the last equality follows from Lemma C.1 and (20).

By doing the same substitution, we obtain that
$$\operatorname{span} \left( \mathcal{L}^{(W)}, \operatorname{span}\{\mathcal{L}^{(q,i)}\}_{i \in \mathcal{I}_0} \right) = \operatorname{span} \left( \{e_i(v_j^\mathsf{T} z)^r\}_{i,j=1}^{n,d}, \{w_j z_\ell(v_j^\mathsf{T} z)^{r-1}\}_{j,\ell=1}^{d,m} \right),$$

which is exactly the set of vectors in (19)–(20). Therefore, by Lemma C.1, we have
$$\dim \operatorname{span} \left( \mathcal{L}^{(W)}, \{\mathcal{L}^{(q,\ell)}\}_{\ell \in \mathcal{I}_0} \right) = (n-1)d + md, \quad \text{and} \tag{29}$$
$$\dim \operatorname{span}\{\mathcal{L}^{(q,\ell)}\}_{\ell \in \mathcal{I}_0} = md. \tag{30}$$

Taking into account that
$$\#(\mathcal{I}_0) = m \quad \text{and} \quad \#(\mathcal{I}) = \binom{R+m-1}{R},$$

this proves (27) for $d_0 = m$. Equality (28) (for $d_0 = m$) can be proved similarly using the fact that
$$\operatorname{rank}\{J_\psi^{(q)}(\widehat{q},W)\} = \dim \operatorname{span}\{\mathcal{L}^{(q,\ell)}\}_{\ell \in \mathcal{I}_0} + \sum_{i \in \mathcal{I} \setminus \mathcal{I}_0} \dim(\mathcal{L}^{(q,i)})$$
$$= md + d(\#(\mathcal{I}) - \#(\mathcal{I}_0)) = d(\#(\mathcal{I})).$$

$\square$

## C.6 Proof of the proposition: extending to the case of more variables

*Proof of Proposition C.8 (case $m < d_0$).* We denote by $\mathcal{I}_m$ (with some abuse of notation) the multi-indices that correspond to the monomials that depend only on $x_1, \ldots, x_m$:

$$\mathcal{I}_m = \{i \in \mathcal{I} : i_{m+1}, \ldots, i_{d_0} = 0\}$$

and we define

$$\mathcal{L}_m^{(q)} := \operatorname{span}\{\mathcal{L}^{(q,i)}\}_{i \in \mathcal{I}_m}, \quad \mathcal{L}_{ext} := \operatorname{span}\{\mathcal{L}^{(q,i)}\}_{i \in \mathcal{I} \setminus \mathcal{I}_m}.$$

From the first part of the proof (case $m = d_0$), we have already proved that

$$\dim \operatorname{span}\left(\mathcal{L}_m^{(q)}, \mathcal{L}^{(W)}\right) = d(n-1) + d\binom{R+m-1}{R}. \tag{31}$$

and

$$\dim \mathcal{L}_m^{(q)} = d\binom{R+m-1}{R}. \tag{32}$$

What is left to show is that adding $\mathcal{L}_{ext}$ to these subspaces does not drop the rank.

Since the particular choice of $q = \widehat{q}(x_1, \ldots, x_m)$ depends only on the $m$ variables, thanks to (26) we have

$$f_{j,(i_1,\ldots,i_{d_0})}(x_1, \ldots, x_{d_0}) = \underbrace{(\quad \cdots \quad)}_{\text{polynomial in } x_1, \ldots, x_m} x_{m+1}^{i_{m+1}} \cdots x_{d_0}^{i_{d_0}}.$$

This immediately implies that $\mathcal{L}^{(q,i)} \perp \mathcal{L}^{(q,\ell)}$ if $(i_{m+1}, \ldots, i_{d_0}) \neq (\ell_{m+1}, \ldots, \ell_{d_0})$, as well as $\mathcal{L}^{(q,i)} \perp \mathcal{L}^{(W)}$ if $(i_{m+1}, \ldots, i_{d_0}) \neq 0$. Therefore, we get

$$\mathcal{L}^{(q)} = \mathcal{L}_m^{(q)} \oplus \mathcal{L}_{ext} \quad \text{and} \quad \operatorname{span}\left(\mathcal{L}^{(q)}, \mathcal{L}^{(W)}\right) = \operatorname{span}\left(\mathcal{L}_m^{(q)}, \mathcal{L}^{(W)}\right) \oplus \mathcal{L}_{ext},$$

and, consequently, we just need to show that $\mathcal{L}_{ext}$ is of maximal dimension. To show this, we split $\mathcal{I} \setminus \mathcal{I}_m$ into a direct sum according to the degrees of the last $d_0 - m$ variables:

$$\dim \mathcal{L}_{ext} = \sum_{\substack{i_{m+1},\ldots,i_{d_0} \geq 0 \\ 1 \leq i_{m+1}+\ldots+i_{d_0} \leq R}} \dim \mathcal{L}^{(q,(*,i_{m+1},\ldots,i_{d_0}))}$$

where

$$\mathcal{L}^{(q,(*,i_{m+1},\ldots,i_{d_0}))} := \operatorname{span}\{\mathcal{L}^{(q,(i_1,\ldots,i_m,i_{m+1},\ldots,i_{d_0}))}\}_{\substack{(i_1,\ldots,i_m):i_k \geq 0, \\ i_1+\ldots+i_{d_0}=R}}.$$

But then, for a fixed $(i_{m+1}, \ldots, i_{d_0})$ such that $i_{m+1} + \ldots + i_{d_0} = R_0 \leq R$, the dimension of this subspace is equal to

$$\begin{aligned}
\dim \mathcal{L}^{(q,(*,i_{m+1},\ldots,i_{d_0}))} &= \dim \operatorname{span}\{x_1^{i_1} \cdots x_{d_0}^{i_{d_0}} w_j(\widehat{q}_j)^{r-1}\}_{\substack{j=1,\ldots,d, \\ i_1,\ldots,i_m \geq 0 \\ i_1+\ldots+i_m=R-R_0}} \\
&= \dim \operatorname{span}\{x_1^{i_1} \cdots x_m^{i_m} w_j(\widehat{q}_j)^{r-1}\}_{\substack{j=1,\ldots,d, \\ i_1,\ldots,i_m \geq 0 \\ i_1+\ldots+i_m=R-R_0}} \\
&= \dim \operatorname{span}\{x_1^{R_0+i_1} \cdots x_m^{i_m} w_j(\widehat{q}_j)^{r-1}\}_{\substack{j=1,\ldots,d, \\ i_1,\ldots,i_m \geq 0 \\ i_1+\ldots+i_m=R-R_0}},
\end{aligned}$$

but the latter set of polynomials is linearly independent because it is a subset of the basis vectors of $\mathcal{L}_m^{(q)}$, which are linearly independent by (32). Therefore we get $\mathcal{L}_{ext}$ is of maximal possible dimension (the spanning columns are linearly independent). $\square$

## C.7 Localization theorem

**Theorem 11 (Localization theorem)** *Let $((d_0, \ldots, d_L), (r_1, \ldots, r_{L-1}))$ be the hPNN format. For $\ell = 0, \ldots, L-2$ denote $\widetilde{d}_\ell = \min\{d_0, \ldots, d_\ell\}$. Then the following holds true: if for all $\ell = 1, \ldots, L-1$ the two-layer architecture $\mathrm{hPNN}_{(\widetilde{d}_{\ell-1}, d_\ell, d_{\ell+1}), r_\ell}[\cdot]$ is finitely identifiable, then the $L$-layer architecture $\mathrm{hPNN}_{d,r}[\cdot]$ is finitely identifiable as well.*

*Proof.* (Proof of Theorem 11) We prove the theorem by induction.

- Base: $L = 2$ The base of the induction is trivial since the case $L = 2$ the full hPNN consists in a 2-layer network.

- Induction step: $(L = k - 1) \to (L = k)$ Assume that the statement holds for $L = k - 1$. Now consider the case $L = k$.

  With some abuse of notation, let $\boldsymbol{\theta} = (\boldsymbol{W}_1, \ldots, \boldsymbol{W}_{L-1})$, so that $\boldsymbol{w} = (\boldsymbol{\theta}, \boldsymbol{W}_L)$ and denote $R = r_1 \cdots r_{L-2}$.

  Let $\psi$ be as the one defined in Proposition C.8, but given for the last subnetwork, so that $n = d_L, d = d_{L-1}, r = r_{L-1}, \boldsymbol{W} = \boldsymbol{W}_L$. Then we have that

  $$\boldsymbol{p}[\boldsymbol{w}] := \text{hPNN}_{(r_1, \ldots, r_{L-1})}[(\boldsymbol{\theta}, \boldsymbol{W}_L)] = \psi[h(\boldsymbol{\theta}), \boldsymbol{W}_L]$$

  where $h(\boldsymbol{\theta}) = \text{hPNN}_{(r_1, \ldots, r_{L-2})}[\boldsymbol{\theta}]$.

  Therefore, by the chain rule

  $$\boldsymbol{J}_{\boldsymbol{p}}(\boldsymbol{w}) = \left[ \underbrace{\left( \boldsymbol{J}_\psi^{(q)} \Big|_{\boldsymbol{q}=h(\boldsymbol{\theta})} \right) \cdot \boldsymbol{J}_h(\boldsymbol{\theta})}_{=\boldsymbol{J}_1(\boldsymbol{w})} \quad \underbrace{\boldsymbol{J}_\psi^{(W)} \Big|_{\boldsymbol{q}=h(\boldsymbol{\theta})}}_{=\boldsymbol{J}_2(\boldsymbol{\theta})} \right],$$

  Now we are going to show that the matrices have necessary rank for generic $\boldsymbol{\theta}$. For this, note by the induction assumption, for generic $\boldsymbol{\theta}$, we have

  $$\text{rank}\{\boldsymbol{J}_h(\boldsymbol{\theta})\} = \sum_{\ell=0}^{L-2} d_\ell d_{\ell+1} - \sum_{\ell=1}^{L-2} d_\ell .$$

  Now we show the ranks for other matrices. Observe that

  $$\text{rank}\left\{ \left[ \left( \boldsymbol{J}_\psi^{(q)} \Big|_{\boldsymbol{q}=h(\boldsymbol{\theta})} \right) \quad \boldsymbol{J}_\psi^{(W)} \Big|_{\boldsymbol{q}=h(\boldsymbol{\theta})} \right] \right\} \leq d_{L-1} \binom{R + d_0 - 1}{R} + (d_L - 1)d_{L-1} \quad (33)$$

  due to the essential ambiguities. But then if we find a particular point $\widehat{\boldsymbol{\theta}}$, where rank is maximal for $\widehat{\boldsymbol{q}} = h(\widehat{\boldsymbol{\theta}})$, then the rank in (33) will be maximal for generic $\boldsymbol{\theta}$.

  But then, let $m = \tilde{d}_{L-1} = \min\{d_0, \ldots, d_{L-1}\}$ and consider the following matrices:

  $$\widehat{\boldsymbol{W}}_1 = \begin{bmatrix} \boldsymbol{I}_m & 0 \\ 0 & 0 \end{bmatrix}, \ldots, \widehat{\boldsymbol{W}}_{L-2} = \begin{bmatrix} \boldsymbol{I}_m & 0 \\ 0 & 0 \end{bmatrix},$$

  and

  $$\widehat{\boldsymbol{W}}_{L-1} = \begin{bmatrix} \boldsymbol{V} & 0 \end{bmatrix},$$

  for $\boldsymbol{V} \in \mathbb{R}^{d_{L-1} \times m}$ generic. Then we get that for $\widehat{\boldsymbol{\theta}} = (\widehat{\boldsymbol{W}}_1, \ldots, \widehat{\boldsymbol{W}}_{L-2})$

  $$h(\widehat{\boldsymbol{\theta}}) = \boldsymbol{V} \begin{bmatrix} x_1^R \\ \vdots \\ x_m^R \end{bmatrix},$$

  so exactly as in Proposition C.8 (whose conditions are satisfied by the assumption on finite identifiability of $\text{hPNN}_{(\tilde{d}_{L-2}, d_{L-1}, d_L), r_{L-1}}[\cdot]$). Therefore, rank in (33) will be maximal for generic $(\boldsymbol{\theta}, \boldsymbol{W}_L)$ and also

  $$\text{rank}\left\{ \left( \boldsymbol{J}_\psi^{(q)} \Big|_{\boldsymbol{q}=h(\boldsymbol{\theta})} \right) \right\} = d_{L-1} \binom{R + d_0 - 1}{R}$$

  for generic $\boldsymbol{\theta}$ (i.e., the matrix is full rank).

This leads to $\mathrm{rank}\{\boldsymbol{J}_1(\boldsymbol{w})\} = \boldsymbol{J}_h(\boldsymbol{\theta})$ for generic $\boldsymbol{\theta}$. Finally, we have that

$$\mathrm{rank}\{\boldsymbol{J_p}(\boldsymbol{w})\} = \mathrm{rank}\{\boldsymbol{J}_1(\boldsymbol{\theta})\} + \mathrm{rank}\{\Pi_{\mathrm{span}\,\boldsymbol{J}_1(\boldsymbol{\theta})_\perp}\boldsymbol{J}_2(\boldsymbol{\theta})\}$$

$$\geq \mathrm{rank}\{\boldsymbol{J}_1(\boldsymbol{\theta})\} + \mathrm{rank}\{\Pi_{\mathrm{span}\left(\boldsymbol{J}_\psi^{(\boldsymbol{q})}\big|_{\boldsymbol{q}=h(\boldsymbol{\theta})}\right)_\perp}\boldsymbol{J}_2(\boldsymbol{\theta})\}$$

$$= \sum_{\ell=0}^{L-2} d_\ell d_{\ell+1} - \sum_{\ell=1}^{L-2} d_\ell + (d_L-1)d_{L-1}$$

$$= \sum_{\ell=0}^{L-1} d_\ell d_{\ell+1} - \sum_{\ell=1}^{L-1} d_\ell\,,$$

where $\Pi_{\mathcal{U}}$ denotes the orthogonal projection onto a subspace $\mathcal{U}$. On the other hand,

$$\mathrm{rank}\{\boldsymbol{J_{p_w}}(\boldsymbol{w})\} \leq \sum_{\ell=0}^{L-1} d_\ell d_{\ell+1} - \sum_{\ell=1}^{L-1} d_\ell$$

due to presence of ambiguities (bound (8)). Hence, an equality holds and therefore the neurovariety has expected dimension.

$\square$

### C.8 Implications of the localization theorem

***Corollary 16 (Pyramidal hPNNs are always identifiable)*** *The hPNNs with architectures containing non-increasing layer widths (except possibly the last layer), i.e., $d_0 \geq d_1 \geq \cdots d_{L-1} \geq 2$ and $d_L \geq 1$, are finitely identifiable for any degrees satisfying (i) $r_1, \ldots, r_{L-1} \geq 2$ if $d_L \geq 2$; or (ii) $r_1, \ldots, r_{L-2} \geq 2$, $r_{L-1} \geq 3$ if $d_L = 1$.*

*Proof.* (Proof of Corollary 16) This follows from Theorem 11 and the following facts:

- For such a choice of $d_\ell$, $\widetilde{d}_\ell = d_\ell$ for all $\ell = 0, \ldots, L-1$;

- Network $(d_{\ell-1}, d_\ell, d_{\ell+1})$ with $d_{\ell-1} \geq d_\ell$ is identifiable (by Proposition 12) for:

    - $r_\ell \geq 2$, in case $d_{\ell+1} \geq 2$;
    - $r_\ell \geq 3$, in case $d_{\ell+1} = 1$.

$\square$

***Corollary 17 (Activation degree thresholds for identifiability)*** *For fixed layer widths $\boldsymbol{d} = (d_0, \ldots, d_L)$ with $d_\ell \geq 2$, $\ell = 0, \ldots, L-1$, the hPNNs with architectures $(\boldsymbol{d}, (r_1, \ldots, r_{L-1}))$ are identifiable for any degrees satisfying*

$$r_\ell \geq 2d_\ell - 1\,.$$

*Proof of Corollary 17.* Note that the assumptions guarantee that $\widetilde{d}_\ell \geq 2$. Then the Kruskal bound (in Proposition 12) for identifiability of $(\widetilde{d}_{\ell-1}, d_\ell, d_{\ell+1})$ can be bounded as

$$\frac{2d_\ell - \min(d_\ell, d_{\ell+1})}{\min(d_\ell, \widetilde{d}_{\ell-1}) - 1} \leq 2d_\ell - 1.$$

therefore, for $r_\ell \geq 2d_\ell - 1$ the hPNN $(\widetilde{d}_{\ell-1}, d_\ell, d_{\ell+1})$, $r_\ell$ is identifiable, which implies the identifiability of the $L$-layer architecture by Theorem 11. $\square$

***Corollary 19 (Identifiability of bottleneck hPNNs)*** *Consider the "bottleneck" architecture with*

$$d_0 \geq d_1 \geq \cdots \geq d_b \leq d_{b+1} \leq \ldots \leq d_L$$

*and $d_b \geq 2$. Suppose that $r_1, \ldots, r_b \geq 2$ and that the decoder part satisfies $\frac{d_\ell}{r_\ell} \leq d_b - 1$ for $\ell \in \{b+1, \ldots, L-1\}$. Then the bottleneck hPNN is finitely identifiable.*

*Proof of Corollary 19.* This follows from Theorem 11 and the following facts:

- For layers $\ell \in \{1, \ldots, b\}$ (the encoder part), we have $\widetilde{d}_\ell = d_\ell$ and thus identifiability of $(\widetilde{d}_{\ell-1}, d_\ell, d_{\ell+1})$ holds for $r_\ell \geq 2$ (the same argument as in the pyramidal case).

- For layers $\ell \in \{b+1, \ldots, L\}$ (the decoder part), we have $\widetilde{d}_\ell = d_b$ and thus identifiability of $(\widetilde{d}_{\ell-1}, d_\ell, d_{\ell+1})$ holds for

$$r_\ell \geq \frac{d_\ell}{d_b - 1},$$

  which, after rearranging, gives the desired result.

$\square$

# D  Analyzing case of PNNs with biases

This appendix contains the proofs and supporting technical results for the identifiability results of PNNs with bias terms presented in Section 3.3 of the main paper. We start by establishing the relationship between PNNs and hPNNs and their uniqueness by means of homogeneization. We then prove our main finite identifiability results showing that finite identifiability of 2-layer subnetworks of the homogeneized PNNs is sufficient to guarantee the finite identifiability of the original PNN.

> **Results from the main paper**: Definition 20, Propositions 23, 24, 27, Lemma 26, and Corollary 28.

## D.1  The homogeneization procedure: the hPNN associated to a PNN

Our homogeneization procedure is based on the following lemma:

**Definition 20.** *There is a one-to-one mapping between (possibly inhomogeneous) polynomials in $d$ variables of degree $r$ and homogeneous polynomials of the same degree in $d+1$ variables. We denote this mapping $\mathscr{P}_{d,r} \to \mathscr{H}_{d+1,r}$ by $\mathrm{homog}(\cdot)$, and it acts as follows: for every polynomial $p \in \mathscr{P}_{d,r}$, $\widetilde{p} = \mathrm{homog}(p) \in \mathscr{H}_{d+1,r}$ is the unique homogeneous polynomial in $d+1$ variables such that*

$$\widetilde{p}(x_1, \ldots, x_d, 1) = p(x_1, \ldots, x_d).$$

*Proof of Definition 20.* Let $p$ be a possibly inhomogeneous polynomial in $d$ variables, which reads

$$p(x_1, \ldots, x_d) = \sum_{\boldsymbol{\alpha}, |\boldsymbol{\alpha}| \leq r} b_{\boldsymbol{\alpha}} \, x_1^{\alpha_1} \cdots x_d^{\alpha_d},$$

for $\boldsymbol{\alpha} = (\alpha_1, \ldots, \alpha_d)$. One sets

$$\widetilde{p}(x_1, \ldots, x_d, z) = \sum_{\boldsymbol{\alpha}, |\boldsymbol{\alpha}| \leq r} b_{\boldsymbol{\alpha}} x_1^{\alpha_1} \cdots x_d^{\alpha_d} z^{r - \alpha_1 - \cdots - \alpha_d}$$

which satisfies the required properties. $\square$

**Associating an hPNN to a given PNN:** Now we prove that for each polynomial $\boldsymbol{p}$ admitting a PNN representation, its associated homogeneous polynomial admits an hPNN representation. This is formalized in the following result.

**Proposition 23.** *Fix the architecture $\boldsymbol{r} = (r_1, \ldots, r_{L-1})$ and $\boldsymbol{d} = (d_0, \ldots, d_L)$. Then a polynomial vector $\boldsymbol{p} \in (\mathscr{P}_{d_0, r_{total}})^{\times d_L}$ admits a PNN representation $\boldsymbol{p} = \mathrm{PNN}_{\boldsymbol{d}, \boldsymbol{r}}[(\boldsymbol{w}, \boldsymbol{b})]$ with $(\boldsymbol{w}, \boldsymbol{b})$ as in (2) if and only if its homogenization $\widetilde{\boldsymbol{p}} = \mathrm{homog}(\boldsymbol{p})$ admits an hPNN decomposition for the same activation degrees $\boldsymbol{r}$ and extended $\widetilde{\boldsymbol{d}} = (d_0 + 1, \ldots, d_{L-1} + 1, d_L)$, $\widetilde{\boldsymbol{p}} = \mathrm{hPNN}_{\widetilde{\boldsymbol{d}}, \boldsymbol{r}}[\widetilde{\boldsymbol{w}}]$, $\widetilde{\boldsymbol{w}} = (\widetilde{\boldsymbol{W}}_1, \ldots, \widetilde{\boldsymbol{W}}_L)$, with matrices given as*

$$\widetilde{\boldsymbol{W}}_\ell = \begin{cases} \begin{bmatrix} \boldsymbol{W}_\ell & \boldsymbol{b}_\ell \\ 0 & 1 \end{bmatrix} \in \mathbb{R}^{(d_\ell + 1) \times (d_{\ell-1} + 1)}, & \ell < L, \\ \begin{bmatrix} \boldsymbol{W}_L & \boldsymbol{b}_L \end{bmatrix} \in \mathbb{R}^{(d_L) \times (d_{L-1} + 1)}, & \ell = L. \end{cases}$$

*Proof of Proposition 23.* Denote $p_1(x) = \rho_{r_1}(W_1 x + b_1)$. Let $\widetilde{x} = \begin{bmatrix} x \\ z \end{bmatrix} \in \mathbb{R}^{d_0+1}$. Observe first that

$$\rho_{r_1}(\widetilde{W}_1 \widetilde{x}) = \begin{bmatrix} \widetilde{p}_1(\widetilde{x}) \\ z^{r_1} \end{bmatrix}.$$

We proceed then by induction on $L \geq 1$.

The case $L = 1$ is trivial. Assume that $L = 2$. Then

$$\widetilde{W}_2 \rho_{r_1}(\widetilde{W}_1 \widetilde{x}) = \widetilde{W}_2 \begin{bmatrix} \widetilde{p}_1(\widetilde{x}) \\ z^{r_1} \end{bmatrix} = W_2 \widetilde{p}_1(\widetilde{x}) + z^{r_1} b_2.$$

Specializing at $z = 1$, we recover

$$W_2 p_1(x) + b_2 = p(x) = \widetilde{p}(x, 1),$$

hence

$$\widetilde{W}_2 \rho_{r_1}(\widetilde{W}_1 \widetilde{x}) = \widetilde{p}(\widetilde{x}).$$

For the induction step, assume that $\tilde{q} = \mathrm{hPNN}_{(d_1+1,\ldots,d_{L-1}+1,d_L),r}[(\widetilde{W}_2,\ldots,\widetilde{W}_L)]$ is the homogeneization of $q = \mathrm{PNN}_{(d_1,\ldots,d_L),r}[((W_2,\ldots,W_L),(b_2,\ldots,b_L))]$. By assumption,

$$\widetilde{p}(x, 1) = \tilde{q}\left( \begin{bmatrix} \widetilde{p}_1(x, 1) \\ 1 \end{bmatrix} \right) = q(\widetilde{p}_1(x, 1)) = q(p_1(x)) = p(x),$$

which completes the proof. $\qquad\square$

**Proposition 24.** *If* $\mathrm{hPNN}_r[\widetilde{w}]$ *from Proposition 23 is unique as an hPNN (without taking into account the structure), then the original PNN representation* $\mathrm{PNN}_r[(w, b)]$ *is unique.*

*Proof of Proposition 24.* Suppose $\mathrm{hPNN}_r[\widetilde{w}]$ is unique (or finite-to-one), where $\widetilde{w}$ is structured as in Proposition 23. Note that any equivalent (in the sense of Lemma 4 specialized for $\mathrm{hPNN}_r[\widetilde{w}]$) parameter vector $\widetilde{w}' = (\widetilde{W}'_1, \ldots, \widetilde{W}'_L)$ realizing the same hPNN must satisfy

$$\widetilde{W}'_\ell = \begin{cases} \widetilde{P}_\ell \widetilde{D}_\ell \widetilde{W}_\ell \widetilde{D}_{\ell-1}^{-r_{\ell-1}} \widetilde{P}_{\ell-1}^\mathsf{T}, & \ell < L, \\ \widetilde{W}_L \widetilde{D}_{L-1}^{-r_{L-1}} \widetilde{P}_{L-1}^\mathsf{T}, & \ell = L. \end{cases} \tag{34}$$

for permutation matrices $\widetilde{P}_\ell$ and invertible diagonal matrices $\widetilde{D}_\ell$, with $\widetilde{P}_0 = \widetilde{D}_0 = I$. We are going to show that bringing $\widetilde{W}'_\ell$ to the form

$$\widetilde{W}'_\ell = \begin{cases} \begin{bmatrix} W'_\ell & b'_\ell \\ 0 & 1 \end{bmatrix}, & \ell < L, \\ \begin{bmatrix} W'_L & b'_L \end{bmatrix}, & \ell = L. \end{cases} \tag{35}$$

that does not introduce extra ambiguities besides the ones for PNN (given in Lemma 4).

By Proposition 33, for $\ell = 1, \ldots, L-1$ the extended matrices satisfy $\mathrm{krank}\{(\widetilde{W}_\ell)^\mathsf{T}\} \geq 2$ (as well as for any equivalent $\mathrm{krank}\{(\widetilde{W}'_\ell)^\mathsf{T}\} \geq 2$). This implies that the matrix $\widetilde{W}_\ell$ contains only a single row of the form $[0 \cdots 0\, \alpha]$ (which is its last row). Therefore in order for $\widetilde{W}'_1$ to be of the form (35), the matrices $\widetilde{P}_1, \widetilde{D}_1$ must be of the form

$$\widetilde{P}_1 = \begin{bmatrix} * & 0 \\ 0 & 1 \end{bmatrix}, \qquad \widetilde{D}_1 = \begin{bmatrix} * & 0 \\ 0 & 1 \end{bmatrix}.$$

Iterating this process for $\ell = 2, \ldots, L-1$, we impose constraints of the form

$$\widetilde{P}_\ell = \begin{bmatrix} * & 0 \\ 0 & 1 \end{bmatrix}, \qquad \widetilde{D}_\ell = \begin{bmatrix} * & 0 \\ 0 & 1 \end{bmatrix}.$$

This implies that $(W'_\ell, b'_\ell)$ and $(W_\ell, b_\ell)$ must be linked as in Lemma 4.

Now suppose that $\mathrm{hPNN}_r[\widetilde{w}]$ is finite-to-one. Then the same reasoning applies to all alternative (non-equivalent) parameters $\widetilde{w}$ that are realized by a PNN, because Proposition 23 holds for every solution. Since there are finitely many equivalence classes, the corresponding PNN representation is also finite-to-one. $\qquad\square$

## D.2 Generic identifiability conditions for PNNs with bias terms

**Lemma 26** *Let the* $2$*-layer hPNN architecture* $((d_0 + 1, d_1 + 1, d_2), (r_1))$ *be finitely (resp. globally) identifiable. Then the PNN architecture with widths* $(d_0, d_1, d_2)$ *and degree* $r_1$ *is also finitely (resp. globally) identifiable.*

*Proof of Lemma 26.* By Proposition 24 we just need to show that for general $(\boldsymbol{W}_2, \boldsymbol{b}_2, \boldsymbol{W}_1, \boldsymbol{b}_1)$, the following hPNN is unique (finite-to-one)

$$\boldsymbol{p}(\widetilde{\boldsymbol{x}}) = [\boldsymbol{W}_2 \quad \boldsymbol{b}_2] \, \rho_{r_1}(\widetilde{\boldsymbol{W}}_1 \widetilde{\boldsymbol{x}}) \tag{36}$$

with $\widetilde{\boldsymbol{W}}_1$ given as

$$\widetilde{\boldsymbol{W}}_1 = \begin{bmatrix} \boldsymbol{W}_1 & \boldsymbol{b}_1 \\ 0 & 1 \end{bmatrix}.$$

We see that $\widetilde{\boldsymbol{W}}_1$ lies in a subspace of $(d_1 + 1) \times (d_0 + 1)$ matrices.

We use the following fact: by multilinearity, both uniqueness and finite-to-one properties of an hPNN are invariant under multiplication of $\widetilde{\boldsymbol{W}}_1$ on the right by any nonsingular $(d_0 + 1) \times (d_0 + 1)$ matrix $\boldsymbol{Q}$. We note that the image of the polynomial map

$$\mathbb{R}^{(d_0+1) \times (d_0+1)} \times \mathbb{R}^{d_1 \times d_0} \times \mathbb{R}^{d_0} \to \mathbb{R}^{(d_1+1) \times (d_0+1)}$$

$$(\boldsymbol{Q}, \boldsymbol{W}_1, \boldsymbol{b}_1) \mapsto \widetilde{\boldsymbol{W}}_1 \boldsymbol{Q},$$

which is surjective, and its image is dense.

Therefore, identifiability (resp. finite identifiability) holds except some set of measure zero in $\mathbb{R}^{(d_1+1) \times (d_0+1)}$, then it also hold for $\widetilde{\boldsymbol{W}}_1$ constructed from almost all $(\boldsymbol{W}_1, \boldsymbol{b}_1)$ pairs. For example, for finite identifiability this is explained by the fact that there is a smooth point of the hPNN neurovariety corresponding to the parameters $([\boldsymbol{W}_2 \quad \boldsymbol{b}_2], \widetilde{\boldsymbol{W}}_1)$. $\qquad\square$

**Proposition 27** *Let* $((d_0, \ldots, d_L), (r_1, \ldots, r_{L-1}))$ *be the PNN format. For* $\ell = 0, \ldots, L - 2$ *denote* $\widetilde{d}_\ell = \min\{d_0, \ldots, d_\ell\}$. *Then the following holds true: If for all* $\ell = 1, \ldots, L - 1$ *each two-layer architecture* $\mathrm{hPNN}_{(\widetilde{d}_{\ell-1}+1, d_\ell+1, d_{\ell+1}), r_\ell}[\cdot]$ *is finitely identifiable, then the* $L$*-layer PNN with architecture* $(\boldsymbol{d}, \boldsymbol{r})$ *is finitely identifiable as well.*

For the proof of the main proposition, we need the following lemma.

**Lemma D.1.** *Global (resp. finite) identifiability of an hPNN of format* $((m, d, n), r)$ *implies (resp. finite) identifiability of the hPNN in format* $((m, d, n + k), r)$ *for any* $k > 0$.

*Proof.* Let the parameters of the larger hPNN be such that

$$\boldsymbol{W}_2 = \begin{bmatrix} \boldsymbol{A} \\ \boldsymbol{B} \end{bmatrix}, \; \boldsymbol{A} \in \mathbb{R}^{n \times d}, \; \boldsymbol{B} \in \mathbb{R}^{k \times d}, \; \boldsymbol{W}_1,$$

so that

$$\mathrm{hPNN}_{(m,d,n+k),\boldsymbol{r}}[\boldsymbol{W}_1, \boldsymbol{W}_2] = \begin{bmatrix} \mathrm{hPNN}_{(m,d,n),\boldsymbol{r}}[\boldsymbol{W}_1, \boldsymbol{A}] \\ \mathrm{hPNN}_{(m,d,k),\boldsymbol{r}}[\boldsymbol{W}_1, \boldsymbol{B}] \end{bmatrix} = \begin{bmatrix} \boldsymbol{A}\sigma_r(\boldsymbol{W}_1\boldsymbol{x}) \\ \boldsymbol{B}\sigma_r(\boldsymbol{W}_1\boldsymbol{x}) \end{bmatrix}.$$

But then assume that $\mathrm{hPNN}_{(m,d,n),\boldsymbol{r}}[\boldsymbol{W}_1, \boldsymbol{A}]$ is finite-to-one. Then by Lemma 31 we have that the elements of $\boldsymbol{q}_1(\boldsymbol{x}) = \sigma_r(\boldsymbol{W}_1\boldsymbol{x})$ are linearly independent, hence the linear system

$$\mathrm{hPNN}_{(m,d,k),\boldsymbol{r}}[\boldsymbol{W}_1, \boldsymbol{B}] = \boldsymbol{B}\boldsymbol{q}_1(\boldsymbol{x})$$

has the unique solution, equal to $\boldsymbol{B}$. Note that for $(\boldsymbol{W}_1, \boldsymbol{W}_2)$, the subset of parameters $(\boldsymbol{W}_1, \boldsymbol{A})$ is also generic, hence global (resp. finite) identifiability for widths $(m, d, n)$ implies global (resp. finite) identifiability for widths $(m, d, n + k)$. $\qquad\square$

*Proof of Proposition 27.* We are going to prove that under the condition of the theorem, two hPNN architectures for degrees $\boldsymbol{r}$ and widths

$$(d_0 + 1, \ldots, d_{L-1} + 1, d_L) \quad \text{and} \quad (d_0 + 1, \ldots, d_{L-1} + 1, d_L + 1)$$

are finitely identifiable.

We proceed by induction, similarly as in Theorem 11.

- **Base: $L = 2$** The base of the induction follows is trivial since it is the 2-layer network, and from Lemma D.1 for the architecture $(d_{\ell-1} + 1, d_\ell + 1, d_{\ell+1} + 1)$.

- **Induction step: $(L = k - 1) \to (L = k)$** Assume that the statement holds for $L = k - 1$. Now consider the case $L = k$. As in the proof of Theorem 11, we set $\widetilde{\boldsymbol{\theta}} = (\widetilde{\boldsymbol{W}}_1, \ldots, \widetilde{\boldsymbol{W}}_{L-1})$, so that $\widetilde{\boldsymbol{w}} = (\widetilde{\boldsymbol{\theta}}, \widetilde{\boldsymbol{W}}_L)$, where $\widetilde{\boldsymbol{W}}_\ell$ is as in Proposition 23, and denote $R = r_1 \cdots r_{L-2}$. The difference is that the parameters are now $\widetilde{\boldsymbol{\theta}} := \widetilde{\theta}(\boldsymbol{\theta})$, where

$$\boldsymbol{\theta} = (\boldsymbol{W}_1, \ldots, \boldsymbol{W}_{L-1}, \boldsymbol{b}_1, \ldots, \boldsymbol{b}_{L-1}).$$

Let $\psi$ be as the one defined in Proposition C.8, but given for the last subnetwork, so that $n = d_L, d = d_{L-1} + 1, r = r_{L-1}, \widetilde{\boldsymbol{W}} = \widetilde{\boldsymbol{W}}_L$. Then we have that

$$\boldsymbol{p}[\boldsymbol{\theta}, \widetilde{\boldsymbol{W}}] := \mathrm{hPNN}_{\boldsymbol{r}}[\widetilde{\boldsymbol{w}}] = \psi[h(\widetilde{\theta}(\boldsymbol{\theta})), \widetilde{\boldsymbol{W}}],$$

where $h(\widetilde{\boldsymbol{\theta}}) = \mathrm{hPNN}_{(r_1, \ldots, r_{L-2})}[\widetilde{\boldsymbol{\theta}}]$.

Again, by the chain rule

$$\boldsymbol{J_p}(\boldsymbol{\theta}, \widetilde{\boldsymbol{W}}) = \left[ \underbrace{\left( \boldsymbol{J}_\psi^{(q)} \Big|_{\boldsymbol{q} = h(\widetilde{\theta}(\boldsymbol{\theta}))} \right) \cdot \boldsymbol{J}_h(\widetilde{\theta}(\boldsymbol{\theta}))}_{= \boldsymbol{J}_1(\widetilde{\boldsymbol{w}})} \quad \underbrace{\boldsymbol{J}_\psi^{(\widetilde{\boldsymbol{W}})} \Big|_{\boldsymbol{q} = h(\widetilde{\theta}(\boldsymbol{\theta}))}}_{= \boldsymbol{J}_2(\boldsymbol{\theta})} \right] \begin{bmatrix} \boldsymbol{J}_{\widetilde{\theta}}(\boldsymbol{\theta}) & \\ & \boldsymbol{I} \end{bmatrix},$$

where the matrix in the right hand side is full column rank. Therefore, we just need to show that the left hand side matrix is full column rank for a particular $\widetilde{\boldsymbol{\theta}} = \widetilde{\theta}(\boldsymbol{\theta})$. But, for this, remark that we can use almost the same construction example Proposition C.8, but choosing slightly different matrices: $\widetilde{\boldsymbol{\theta}}' = (\widehat{\boldsymbol{W}}'_1, \ldots, \widehat{\boldsymbol{W}}'_{L-1})$ with

$$\widehat{\boldsymbol{W}}'_1 = \begin{bmatrix} \boldsymbol{0} & \boldsymbol{0} \\ \boldsymbol{0} & \boldsymbol{I}_m \end{bmatrix}, \ldots, \widehat{\boldsymbol{W}}_{L-2} = \begin{bmatrix} \boldsymbol{0} & \boldsymbol{0} \\ \boldsymbol{0} & \boldsymbol{I}_m \end{bmatrix}$$

and

$$\widehat{\boldsymbol{W}}'_{L-1} = \begin{bmatrix} \boldsymbol{0} & \boldsymbol{V}' \end{bmatrix},$$

where in Lemma C.1 we can choose generic $\boldsymbol{V}'$ structured as

$$\boldsymbol{V}' = \begin{bmatrix} \boldsymbol{W}^{(V')} & \boldsymbol{b}^{(V')} \\ 0 & 1 \end{bmatrix}.$$

Indeed, we need this to be a smooth point (i.e., full rank Jacobian of $\boldsymbol{W} \rho_{r_{L-1}}(\boldsymbol{V}'\boldsymbol{x})$), which is full rank for generic $\boldsymbol{W}^{(V')}, \boldsymbol{b}^{(V')}$, by the same argument as in the proof of Lemma 26.

But such $\widetilde{\boldsymbol{\theta}}'$ indeed belongs to the image of $\widetilde{\theta}(\boldsymbol{\theta})$ as they share the needed structure, which completes the proof.

$\square$

**Corollary 28.** *Let $((d_0, \ldots, d_L), (r_1, \ldots, r_{L-1}))$ be such that $d_\ell \geq 1$, and $r_\ell \geq 2$ satisfy*

$$r_\ell \geq \frac{2(d_\ell + 1) - \min(d_\ell + 1, d_{\ell+1})}{\min(d_\ell, \widetilde{d}_{\ell-1})},$$

*then the L-layer PNN with architecture $(\boldsymbol{d}, \boldsymbol{r})$ is finitely identifiable (and globally identifiable when $L = 2$).*

*Proof of Corollary 28.* This directly follows from combining Lemma 26, Proposition 12 and Proposition 27. $\square$

## D.3 Truncation of PNNs with bias terms

In this appendix, we describe an alternative (to homogenization) approach to prove the identifiability of the weights $\boldsymbol{W}_\ell$ of $\mathrm{PNN}_{\boldsymbol{d},\boldsymbol{r}}[(\boldsymbol{w},\boldsymbol{b})]$ based on *truncation*. The key idea is that the truncation of a PNN is an hPNN, which allow one to leverage the uniqueness results for hPNNs. However, we note that unlike homogeneization, truncation does not by itself guarantees the identifiability of the bias terms $\boldsymbol{b}_\ell$.

For truncation, we use leading terms of polynomials, i.e. for $p \in \mathscr{P}_{d,r}$ we define $\mathrm{lt}\{p\} \in \mathscr{H}_{d,r}$ the homogeneous polynomial consisting of degree-$r$ terms of $p$:

**Example D.2.** *For a bivariate polynomial $p \in \mathscr{P}_{2,2}$ given by*

$$p(x_1, x_2) = ax_1^2 + bx_1x_2 + cx_2^2 + ex_1 + fx_2 + g, \; .$$

*its truncation $q = \mathrm{lt}\{p\} \in \mathscr{H}_{2,2}$ becomes*

$$q(x_1, x_2) = ax_1^2 + bx_1x_2 + cx_2^2 \, .$$

In fact $\mathrm{lt}\{\cdot\}$ is an orthogonal projection $\mathscr{P}_{d,r} \to \mathscr{H}_{d,r}$; we also apply $\mathrm{lt}\{\cdot\}$ to vector polynomials coordinate-wise. Then, PNNs with biases can be treated using the following lemma.

**Lemma D.3.** *Let $\boldsymbol{p} = \mathrm{PNN}_{\boldsymbol{d},\boldsymbol{r}}[(\boldsymbol{w},\boldsymbol{b})]$ be a PNN with bias terms. Then its truncation is the hPNN with the same weight matrices*

$$\mathrm{lt}\{\boldsymbol{p}\} = \mathrm{hPNN}_{\boldsymbol{d},\boldsymbol{r}}[\boldsymbol{w}].$$

*Proof.* The statement follows from the fact that $\mathrm{lt}\{(q(\boldsymbol{x}))^r\} = \mathrm{lt}\{(q(\boldsymbol{x}))\}^r$. Indeed, this implies $\mathrm{lt}\{(\langle \boldsymbol{v}, \boldsymbol{x}\rangle + \boldsymbol{c})^r\} = (\langle \boldsymbol{v}, \boldsymbol{x}\rangle)^r$, which can be applied recursively to $\mathrm{PNN}_{\boldsymbol{d},\boldsymbol{r}}[(\boldsymbol{w},\boldsymbol{b})]$. □

**Example D.4.** *Consider a 2-layer PNN*

$$f(\boldsymbol{x}) = \boldsymbol{W}_2 \rho_{r_1}(\boldsymbol{W}_1 \boldsymbol{x} + \boldsymbol{b}_1) + \boldsymbol{b}_2. \tag{37}$$

*Then its truncation is given by*

$$\mathrm{lt}\{f\}(\boldsymbol{x}) = \boldsymbol{W}_2 \rho_{r_1}(\boldsymbol{W}_1 \boldsymbol{x}).$$

This idea is well-known and in fact was used in [44] to analyze identifiability of a 2-layer network with arbitrary polynomial activations.

**Remark D.5.** *Thanks to Lemma D.3, the identifiability results obtained for hPNNs can be directly applied. Indeed, we obtain identifiability of weights, under the same assumptions as listed for the hPNN case. However, this does not guarantee identifiability of biases, which was achieved using homogeneization.*

# E Localization theorem: necessary and sufficient conditions for identifiability

This appendix has been added to the camera ready version on the request of the program committee. It explains the changes between the original submission and the camera-ready version.

Our main technical result in Theorem 11 gives sufficient conditions for finite identifiability of deep hPNNs based on identifiability of two-layer subnetworks. In an earlier (submitted) version of the paper, the following results were claimed.

**Claim** (A, specific uniqueness)**.** *Let $\mathrm{hPNN}_{\boldsymbol{r}}[\boldsymbol{w}]$, $\boldsymbol{w} = (\boldsymbol{W}_1, \ldots, \boldsymbol{W}_L)$ be an L-layer hPNN with architecture $(\boldsymbol{d}, \boldsymbol{r})$ satisfying $d_0, ..., d_{L-1} \geq 2$, $d_L \geq 1$ and $r_1, \ldots, r_{L-1} \geq 2$. Then, $\mathrm{hPNN}_{\boldsymbol{r}}[\boldsymbol{w}]$ is unique according to Definition 6 if and only if for every $\ell = 1, \ldots, L-1$ the 2-layer subnetwork $\mathrm{hPNN}_{(r_\ell)}[(\boldsymbol{W}_\ell, \boldsymbol{W}_{\ell+1})]$ is unique as well.*

This strong claim implied another claim on identifiability of hPNN architectures, which can be seen as a counterpart of the current Theorem 11.

**Claim** (B, identifiability)**.** *The L-layer hPNN with architecture $(\boldsymbol{d}, \boldsymbol{r})$ satisfying $d_0, \ldots, d_{L-1} \geq 2$, $d_L \geq 1$ and $r_1, \ldots, r_L \geq 2$ is identifiable according to Definition 8 if and only if for every $\ell = 1, \ldots, L-1$ the 2-layer subnetwork with architecture $((d_{\ell-1}, d_\ell, d_{\ell+1}), (r_\ell))$ is identifiable as well.*

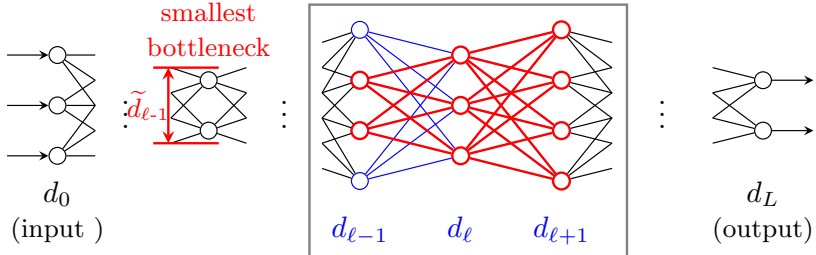

Figure 1: (from NeurIPS poster) Necessary and sufficient conditions for identifiability of an $L$-layer PNN. Blue: necessary conditions, i.e., "only if" part of claim B (identifiability of the $((d_{\ell-1}, d_\ell, d_{\ell+1}), (r_\ell))$ subnetwork). Red: sufficient condition as given by Theorem 11 (identifiability of the $((\widetilde{d}_{\ell-1}, d_\ell, d_{\ell+1}), (r_\ell))$ subnetwork).

We note that the "only if" part always holds (as argued in the beginning of Section 3.1), as non-uniqueness of any 2-layer subnetwork implies non-uniqueness of the overall network. The relation between necessary and sufficient conditions for identifiability is illustrated in Fig. 1.

Thus, both claims (A) and (B) were in fact aiming to answer the following questions:

(A) *Does uniqueness of all 2-layer subnetworks* $\mathrm{hPNN}_{(r_\ell)}[(\boldsymbol{W}_\ell, \boldsymbol{W}_{\ell+1})]$ *imply uniqueness of the overall network* $\mathrm{hPNN}_{(r_1,\ldots,r_{L-1})}[(\boldsymbol{W}_1, \ldots, \boldsymbol{W}_L)]$?

(B) *Does identifiability of all* $((d_{\ell-1}, d_\ell, d_{\ell+1}), (r_\ell))$ *2-layer architectures imply the identifiability of the overall architecture* $((d_0, \ldots, d_L), (r_1, \ldots, r_{L-1}))$?

We show below that the answer to these questions is negative, both for specific uniqueness (uniqueness of a particular choice of parameters) and generic uniqueness (identifiability of a given architecture), which motivated the update of the paper.

### E.1 Supporting examples

**Absence of specific uniqueness (counterexample to claim (A)).** Consider the simplest architecture with $\boldsymbol{d} = (2, 2, 2)$, $\boldsymbol{r} = (2, 2)$, for which the conditions of Theorem 11 are verified due to Proposition 12. Example 41 from the last section of the paper provides an example of specific network of the format $(\boldsymbol{d}, \boldsymbol{r})$ violating claim (A). We provide below an expanded version of this example.

**Example** 41 (No specific uniqueness). Consider two polynomials:

$$\boldsymbol{p}(x_1, x_2) = \begin{bmatrix} (x_1^2 + x_2^2)^2 \\ (x_1^2 - x_2^2)^2 \end{bmatrix}.$$

Note that $\begin{bmatrix} x_1^2 & x_2^2 \end{bmatrix}^\top = \rho_2(x_1, x_2)$, therefore this polynomial vector can be written as

$$\boldsymbol{p}(\boldsymbol{x}) = \rho_2\big(\boldsymbol{W}_2 \rho_2(\boldsymbol{x})\big)$$

for the following choice of weight matrix:

$$\boldsymbol{W}_2 = \begin{bmatrix} 1 & 1 \\ 1 & -1 \end{bmatrix},$$

so that we have

$$\begin{aligned} \boldsymbol{p}(\boldsymbol{x}) &= \boldsymbol{I}_2 \rho_2\big(\boldsymbol{W}_2 \boldsymbol{I}_2 \rho_2(\boldsymbol{x})\big) \\ &= \mathrm{hPNN}_{(2,2)}[(\boldsymbol{I}_2, \boldsymbol{W}_2, \boldsymbol{I}_2)], \end{aligned}$$

where $\boldsymbol{I}_2$ is the identity matrix. On the other hand, we can use the expansions

$$x_1^2 + x_2^2 = \frac{(x_1 + x_2)^2 + (x_1 - x_2)^2}{2}$$

$$2x_1 x_2 = \frac{(x_1 + x_2)^2 - (x_1 - x_2)^2}{2}$$

and the fact that
$$(x_1^2 - x_2^2) = (x_1^2 + x_2^2)^2 - (2x_1x_2)^2$$
to show that there exists an alternative hPNN expansion of $p(x)$, summarized as
$$p(x) = W_3\rho_2\left(\frac{1}{2}W_2\rho_2(W_2x)\right) = \text{hPNN}_{(2,2)}[(W_2, \frac{1}{2}W_2, W_3)],$$
where
$$W_3 = \begin{bmatrix} 1 & 0 \\ 1 & -1 \end{bmatrix}.$$

We see that the two representations are not equivalent: $(I_2, W_2, I_2) \not\sim (W_2, \frac{1}{2}W_2, W_3)$, as $W_2$ cannot be obtained from scaling and permutations of rows of $I_2$.

On the other hand, all the matrices in the expansions $(I_2, W_2, W_3)$ are $2 \times 2$ invertible and thus, for example, the networks $\text{hPNN}_{(2)}[(I_2, W_2)]$ and $\text{hPNN}_{(2)}[(W_2, I_2)]$ have unique representations (similarly to Example 7). More precisely, all the matrices $(I_2, W_2, W_3)$ as well as their transposes have their rank and Kruskal rank both equal to 2, and therefore the conditions of Lemma B.1 are satisfied.

**Absense of generic uniqueness (counterexample to claim (B)).** Example 41 is not just an isolated example that can be circumvented by looking at a generic parameter set, as shown in the following example.

**Example E.1** (No generic identifiability without further assumptions). *We provide a counterexample to the conjecture that localization holds in full generality in the generic sense based on the count of dimension. Consider the following architecture:*
$$d = (2, 3, 3, 1) \quad and \quad r = (3, 3).$$
*It is easy to see that the subnetworks $((d_0, d_1, d_2), r_1)$ and $((d_1, d_2, d_3), r_2)$ both satisfy the Kruskal-based criterion in Proposition 12 as*
$$3 \geq \frac{2d_1 - \min(d_2, d_1)}{\min(d_1, d_0) - 1} = 3, \qquad 3 \geq \frac{2d_2 - \min(d_3, d_2)}{\min(d_2, d_1) - 1} = \frac{5}{2},$$
*so both subnetworks are identifiable. However, due to Proposition 10, for the global network $(d, r)$ to be identifiable the dimension of its associated neurovariety must be equal to*
$$d_0d_1 + d_1d_2 + d_2d_3 - d_1 - d_2 = 12.$$
*However, the image of $\text{hPNN}_{(d,r)}[\cdot]$ is in the space of degree-9 homogeneous bivariate polynomials, and therefore the neuromanifold (and the neurovariety) lies in $\mathcal{H}_{2,9}$. But $\mathcal{H}_{2,9}$ has dimension 10, thus we arrive at a contradiction with the identifiability of the 3-layer network.*

## E.2 Statement of changes

In the camera-ready version, the claims (A) and (B) have been replaced with Theorem 11 which uses a stricter condition. This replacement preserves the main conclusions and contributions of the original paper, notably:

1. *The localization of identifiability*: identifiability of 2-layer subnetworks (composed by two consecutive layers) is sufficient to guarantee identifiability of a deep $L$-layer polynomial network;

2. As a consequence, *uniqueness theorems for tensors can be leveraged* to prove identifiability of deep PNNs; for example, well-known Kruskal theorems imply:

   a) that pyramidal networks (and their generalizations) are identifiable in degrees $\geq 2$;

   b) linear bounds on the so-called *activation degree thresholds* (i.e., identifiability holds for degrees linear in the layer widths);

3. *Identifiability of networks with biases is implied by identifiability of* (augmented) *bias-free PNN architectures.*

**Drawbacks:** Despite the fact that our main conclusions still hold, the amended version of the localization theorem lead to the following changes:

- The theorem and the corresponding corollaries for deep architectures concern generic properties (and not specific) and finite identifiability (instead of global identifiability).

- Theorem 11 requires a stronger assumption on 2-layer subnetworks: not only each 2-layer block needs to be identifiable, but also with a possibly smaller number of inputs.

- This stronger condition weakened the result for networks with a bottleneck layer, but keeps the same conclusion (that is, a decoder network needs to have higher degrees compared to the encoder in order to allow for increasing the layer widths).

In the following, we explain the mistake in the original proof of Theorem 11 and discuss the current challenges to extending the amended proof to the localization of global identifiability.

### E.3 Remark on the mistake in the original proof and related problems

The mistake in the original proof of Theorem 11 concerned equations (11)–(13) in the original paper (Section A.2.2 of the original supplementary materials), in the induction step of the theorem (going from $L-1$ to $L$ layers). The original argument is based on constructing the polynomial vector $\boldsymbol{p}'_w(\boldsymbol{z})$ using the flattening operation $\boldsymbol{x}^{\otimes r'} \mapsto \boldsymbol{z}$, $\mathscr{H}_{d_0, r'} \to \mathbb{R}^{(d_0)^{r'}}$. The issue is that the equation (12, original paper) is only valid on a subset of $\boldsymbol{z}$ ($\boldsymbol{z}$ structured as a tensor power) and thus does not imply (13, original paper) as we originally claimed. The absence of this implication broke the inductive argument, requiring the proof to be amended.

An interpretation of this issue is that the flattening destroys the structure from lower layers. In fact, the flattening mapping corresponds to a projection appearing in the computation of the decomposition of polynomials as sums of powers of forms (see the commutative diagram in [92, Section 4]), which makes such a computation (decomposition as a sum of powers of polynomials) very difficult and currently an open problem in general, unless additional knowledge can be used [93].

Our new proof still proceeds similarly by induction (going from $L-1$ to $L$ layers), where the induction step is related to showing non-defectivity (finite identifiability) of a subvariety of variety of powers of forms, thus connecting to subtle questions in algebraic geometry, such as Fröberg's conjecture [92] (the latter not solved in full generality, see [85] for an account of recent progress). Extending finite identifiability to identifiability seems challenging, at least with the techniques we are aware of; very recent work in algebraic geometry [75, 86] shows that this transition (i.e., *finite identifiability implies global identifiability*) is possible for the so-called X-rank decompositions, but this result is only applicable to shallow polynomial networks. We are not aware of any systematic progress in the direction of non-additive structures, of which deep PNNs is a special case. Thus, the transition from finite to global identifiability of deep PNNs was left as an open conjecture (Conjecture 40) in the camera-ready version of the paper. We hope that future progress in the field of algebraic geometry will provide the adequate tools to settle this challenging problem[12].

---

[12]While preparing the update of the camera-ready version of the paper, we became aware of a recent preprint [94] that claims to prove a much stronger (in many cases) result than Theorem 11 and claims global identifiability as well.

