# OpenReview forum: "Identifiability of Deep Polynomial Neural Networks"
_NeurIPS.cc/2025/Conference — NeurIPS 2025 oral_

### Official Review · Reviewer_Ux2R · 2025-06-13

**Clarity:** 3
**Significance:** 2
**Originality:** 3
**Rating:** 5
**Confidence:** 4

**Summary:**

This manuscript develops an algebraic-geometric theory of global identifiability for MLPs with monomial activation functions, specifically $x^r$. The arguments are based on the relationship between partially symmetric canonical polyadic tensor decomposition and monomial MLPs. The main contributions of the paper are:
 - An L-layer monomial MLP is globally identifiable if and only if every consecutive two-layer block is identifiable (Theorem 11).
 - Establishing a lower bound for the activation degree at which monomial MLPs become identifiable (Propositions 13 and 15). This bound is linear in terms of layer widths, improving the quadratic bound provided in [1].
 -  Providing separate analyses for pyramidal architectures and those featuring bottleneck layers.
- Extending the analysis to monomial MLPs with bias through a homogenization technique, which to my knowledge was not considered in previous literatures.

[1] Finkel et al., Activation thresholds and expressiveness of polynomial neural networks, 2024.

**Questions:**

-  In lines 37-38, the authors claim that results in [1, 2, 3] focus on local identifiability rather than global identifiability. However, these references appear to describe the generic fiber of the parameterization, such as the proof of Theorem 18 in [1] and the birationality result in [2]. Could the authors clarify why this is considered local rather than global identifiability?
- In Section 4.3, regarding shallow networks, the locus of functions admitting unique decomposition is described via the Kruskal rank condition. Does this locus correspond exactly to functions in monomial MLPs without any neurons being inactive? Is this statement true for deep case based on theorem 11?

 [2] Shahverdi et al., On the geometry and optimization of polynomial convolutional networks, 2024.

 [3] Henry et al., Geometry of lightning self-attention: Identifiability and dimension, 2024.

**Ethical Concerns:**

["NO or VERY MINOR ethics concerns only"]

**Final Justification:**

The paper tackles an important and long-standing problem, improving and extending existing results, and providing new techniques. It is mathematically rigorous and original. Therefore, I believe that the work deserves acceptance.

**Limitations:**

As mentioned above, the limitations regarding the choice of monomial activation are not discussed. I recommend the authors to include it.

**Paper Formatting Concerns:**

There are no formatting issues in the paper.

**Quality:**

3

**Strengths And Weaknesses:**

# Strengths
- The manuscript including the appendix is overall well-written, and the method used to reduce the analysis of deep monomial MLPs to shallow ones is very elegant.

- Although the results have limited implications for real-world applications, the approach has the potential to be extended to the analysis of MLPs with arbitrary polynomial activations. This is particularly important in real-world applications, as any continuous activation function can be approximated by polynomials over a compact set.

- Another notable aspect is the inclusion of bias in the analysis. Prior works on monomial MLPs such as [1] ignored bias, which posed a limitation for applying algebraic geometry techniques to general neural network analysis.

 # Weaknesses
- The authors refer to "polynomial" activation function in the abstract and introduction without specifying that they mean "monomial," which could lead to confusion. I suggest for clarity, the term monomial being mentioned as this is the primary setup for this paper.

- From an empirical perspective, the paper’s impact appears limited. This point is not sufficiently discussed, and the potential for generalizing the results to arbitrary polynomial activations—an extension that could substantially enhance the paper’s relevance—is not addressed.

- Lastly, there is a minor typo on line 271: "Example 51" should be corrected to "Example 5."

Overall, I highly recommend this work for its novel methodology and its potential impact in the context of the algebraic analysis of deep neural netwoks.

---

> ### Author Rebuttal · Authors · 2025-07-29
>
> We would like to thank the Reviewer for the careful evaluation of our work and for the valuable suggestions for improvements. In the following, we provide responses and clarifications to the weaknesses or questions raised by the Reviewer.
>
> Q1: **Tackling monomial vs. general polynomial activations**: This is a good point and we totally agree with the Reviewer, however, we also emphasize that, in all papers starting from the paper by Kileel, Trager, and Bruna (reference [7]), the term "polynomial neural networks" refers to neural networks with monomial activations, and it is a standard terminology used in the literature. Nevertheless, we will clarify that the results focus on monomial activations instead of general polynomial activations when describing the setup of the revised paper, which should better contextualize our contribution, and also in the discussion of the limitations of the paper. We would also like to emphasize that the study of the monomial case is a fundamental first step in tackling more general polynomial activations, as further clarified in the answer to the following question.
>
> [7] J. Kileel et al. On the expressive power of deep polynomial neural networks. Advances in neural information processing systems, 32, 2019.
>
>
> Q2: **Empirical impact and generalization to arbitrary polynomial activations**: We thank the Reviewer for the suggestion. We agree that an extension to consider general polynomial activations instead of monomial ones would greatly improve the practical impact of our results. We do believe such an extension is possible. In fact, the monomial case is the key to address the general polynomial case. For instance, the very recent paper [Shahverdi et al., 2025] (published May 17th, 2025, after the submission of our paper) considers more general polynomial activations and they first rely on results for the monomial case. This is an exciting subject which we plan to investigate in a followup work.
>
> [Shahverdi et al., 2025] V. Shahverdi et al. Learning on a Razor's Edge: the Singularity Bias of Polynomial Neural Networks. arXiv preprint arXiv:2505.11846. (2025).
>
>
> Q3: **Typo in line 271**: Thank you very much for pointing this out, we will fix it in the revised paper.
>
>
> Q4: **Local identifiability rather than global identifiability in [1, 2, 3]**: The reviewer is right, this was an incorrect wording from our side. Indeed, the proof of Theorem 18 in [11] contains a result on global identifiability (for a high degree though, so weaker than our results, as we mentioned in the introduction), but the statement of the theorem is about local identifiability (more precisely, about dimension of the neurovariety, which implies that the parameterization map is finite-to-one, subject to scaling ambiguities). References [8] and [9] indeed also contain results  about global identifiability, but for other/particular architectures (convolutional and attention-like layers) and thus are not applicable to general PNNs. What we meant is that identifiability is not the main focus of these papers, but we believe it is a central concept that  is key to understanding the geometry of PNNs. We do apologize for this incorrect wording and we will correct it in the revised paper.
>
>
> Q5: **Kruskal rank condition and MLPs without inactive neurons**: Thank you for the insightful question. The Kruskal rank conditions used in our results requires that, for a 2-layer network with weight matrices $W_1$ and $W_2$ for the first and second layer, respectively, $W_1$ has no collinear rows, and $W_2$ has no zero columns. This in turn corresponds to networks that are **irreducible** in the sense that the activations in the hidden layer are linearly independent, which is slightly stronger than being "active"/nonzero.
>
> In fact, we have that for high enough degrees, uniqueness is achieved **if and only if** the weight matrices satisfy these Kruskal rank constraints (these necessary conditions become sufficient). This is also true for the case of deep (L-layers) networks due to the localization theorem. We will improve the discussion of the implication of the Kruskal rank conditions in the revised version of the paper.

---

> > ### Comment · Reviewer_Ux2R · 2025-08-04
> >
> > I am grateful to the authors for their precise and thoughtful reply, which has addressed my concerns (which were minor). I am happy to keep my positive score.

---

### Official Review · Reviewer_f8tx · 2025-06-21

**Clarity:** 3
**Significance:** 2
**Originality:** 3
**Rating:** 4
**Confidence:** 2

**Summary:**

This paper studies the identifiability of the parameters of single-hidden-layer neural networks with polynomial activation functions. The main result shows that under mild conditions on the activation degree and input distribution, a polynomial network with kk hidden units is identifiable from the input-output function it computes. The authors characterize when two parameter settings define the same function and give conditions under which all parameters are uniquely recoverable up to symmetry. Their analysis builds on algebraic tools, including properties of polynomial ideals and ranks of function spaces.

**Questions:**

The main result is mathematically clean and addresses an important foundational question—identifiability—in a tractable setting. Although the class of networks studied (shallow, polynomial activations) is of limited practical use today, the result itself is of theoretical interest and may inform future work on understanding parameter-function mappings in broader network families.


- Include a short discussion of whether and how the results extend to real analytic or piecewise-linear activations.
- Consider a brief experiment or example showing what identifiability looks like in practice—e.g., a visualization of function-equivalent but parameter-distinct networks failing under certain assumptions.
- Clarify the implications for learning: does identifiability imply anything about convergence or optimization in practice for this class?

**Ethical Concerns:**

["NO or VERY MINOR ethics concerns only"]

**Final Justification:**

The paper is good and technically sound, though the results have limited applicability and it is difficult to say howto extend them. From my personal point of view, the main weakness is not a technical one, it is about the research topic. Let us assume that the goal of the community  is to understand fundamental principles of learning and time is short to achieve this goal. Then I feel that identifiability of polynomial networks is not a high priority topic. My grade follow from this belief not by problems with the mathematics of the paper.

**Limitations:**

yes

**Paper Formatting Concerns:**

no problem

**Quality:**

3

**Strengths And Weaknesses:**

Among the strengths are a clear, elegant identifiability result: The core result on the identifiability of shallow polynomial networks is cleanly stated and technically sound. It adds clarity to a fundamental question—when is the function computed by a neural network uniquely determined by its parameters?—in a setting amenable to analysis. Furthermore, the authors use appropriate tools from algebraic geometry and multilinear algebra to derive their results. The arguments are accessible while still mathematically rigorous. Finally, the paper is clearly written, especially for a mathematically inclined audience. The structure, notation, and logical flow make the main result easy to follow.
---
Among the weaknesses, I think the relevance is limited. The results concern networks with monomial activations, which are never used in modern deep learning. The authors do not discuss how the identifiability properties extend—or fail to extend—to more realistic activation functions (e.g., ReLU, tanh). As such, the results may have limited direct impact on practical neural network design or training. Furthermore while this is a theory paper, even a small numerical experiment (e.g., verifying identifiability or its failure in small examples) would make for better readability.  The identifiability of polynomial systems is closely related to classical results like Bézout’s Theorem, which concerns the number of solutions to systems of polynomial equations. Finally, the paper does not explore whether the results might extend to deeper networks, approximate identifiability (in the presence of noise), or probabilistic identifiability. These are directions of  interest in both learning theory and applied statistics.

---

> ### Author Rebuttal · Authors · 2025-07-29
>
> We would like to thank the Reviewer for the careful evaluation of our work and for the valuable suggestions for improvements. In the following, we provide responses and clarifications to the weaknesses or questions raised by the Reviewer.
>
> **Response to weaknesses**: First we would like to emphasize that the proposed techniques constitute a crucial first step in the study of the identifiability of more general architectures, such as general polynomial networks (which can provide good approximations to activations commonly used by practitioners), see the answer to Q2 by Reviewer Ux2R for more details. Moreover, our results also address networks of arbitrary depth. Furthermore, we also address the question of uniqueness of a specific set of NN parameters (Theorem 11), which is a much stronger property than identifiability (Corollary 12).
>
> We also agree that approximate or probabilistic identifiability are of great interest in ML theory and applied statistics. However, a first stepping stone to these analysis requires understanding the identifiability in the noiseless case, which is the aim of our paper. We do provide an a clear and easy-to-digest answer to this question and advance the understanding of their neurovarieties in a wide range of settings. Thus, while approximate or probabilistic identifiability are are exciting and challenging topics, these will be better suited to be addressed in a dedicated future work.
>
> Q1: **Role of Bézout's theorem**: We thank the Reviewer for this very interesting remark.
> Bézout's Theorem is indeed is very classical and fundamental result in algebraic geometry. Briefly, Bézout's theorem asserts that a **general** system of $n$ polynomial equations in $n$ variables has a finite number of solutions. However, for several reasons, it is not applicable in our situation.
> - The uniqueness property can indeed be written as a system of polynomial equations. Nevertheless to fulfill the requirements of Bézout's theorem, we would need to have more independent equations than the number of variables, that is, a "small" number of neurons in the hidden layers compared to the others parameters (degree, extremal widths). This is generally not the case in applications.
> - Bézout's theorem holds for **general** polynomials, i.e. chosen out of a set of zero Lebesgue measure. However, this set is not explicit: in other words, Bézout does not provide an explicit manner to check whether our uniqueness property (or even identifiability) holds for the equations arising from a given PNN.
> - It is also important to notice that for **monomial** activations, as we explained in the paper, we can only have uniqueness/finiteness **up to the trivial ambiguities**, and there always exist infinitely many equivalent decompositions. Results like Bézout's theorem do not apply directly in this context, and we would need more advanced algebro-geometric techniques to tackle this very important question.
>
> Q2: **Relevance and extension to more common (real analytic or piecewise-linear activations) activations**: On the one hand, our proof techniques are considerably different than those of previous works focused on special cases of analytic functions such as tanh or sigmoidal activations; thus, directly generalizing our proofs to general real analytic functions appears to be highly nontrivial. On the other hand, considering monomial activation functions is a necessary and crucial step to tackle the case of PNNs with general polynomial activations. In a future work, we plan to tackle this general polynomial case, which is of significant practical interest, by extending our proof techniques. We will include a discussion about possible research directions to extend our results to NNs with more general polynomial activation functions in the revised version of the paper.
>
> Q3: **Experiment or example showing what identifiability looks like in practice**: Thank you for the valuable suggestion. In order to better illustrate identifiable and non-identifiable networks in practice, we consider will include additional examples in the revised paper. In particular, we will include an example to compare PNNs that are 1) unique and thus identifiable, 2) identifiable but non-unique (i.e., with pathological weight matrices, such as collinear rows), and 3) non-identifiable (and thus, also not unique).
>
> Q4: **Implications of identifiability on learning or optimization**: We thank the Reviewer for this interesting question. In fact, identifiability has many important links to learning and optimization of PNNs. In particular, identifiability provides important information on the dimension of their associated neurovariety and neuromanifold, which, in turn, has deep implications on their expressivity, sample complexity, and on the geometry of their learning landscapes, as noted in previous works [7] and [9]. Moreover, a key result from singular learning theory (see [Watanabe, 2009] for background) is that non-identifiable networks are precisely singular points of the neurovariety, and in fact, learning algorithms are biased towards such solutions [Shahverdi et al., 2025]. Thus, it is very important to characterize which networks are non-identifiable [Shahverdi et al., 2025]. We will comment on the implications of identifiability to learning in the revised version of the paper.
>
> [Shahverdi et al., 2025] Shahverdi, V., Marchetti, G. L., and Kohn, K. (2025). Learning on a Razor's Edge: the Singularity Bias of Polynomial Neural Networks. arXiv preprint arXiv:2505.11846.
>
> [Watanabe, 2009] S. Watanabe, "Algebraic Geometry and Statistical Learning Theory", Cambridge University Press, 2009.
>
> [7] J. Kileel et al. On the expressive power of deep polynomial neural networks. Advances in neural information processing systems, 32, 2019.
>
> [9] N. W Henry et al. Geometry of lightning self-attention: Identifiability and dimension. ICLR 2025, 2025. arXiv preprint arXiv:2408.17221.

---

> > ### Comment · Reviewer_f8tx · 2025-08-02
> > **Most NNs are overparametrized**
> >
> > I want to clarify why I mentioned Bezout theorem. One of its implications - I think - is that in the overparametrized case -- when the number N of equations f(x_i)=y_i is smaller than the number of variables V (the weights) -- the interpolating solutions have degeneracy W-N. This is a situation that often happens in practice during training. I am curious about what can you say about identifiability of these interpolating solutions.

---

> > > ### Author Response · Authors · 2025-08-04
> > >
> > > We would like to thank the Reviewer for taking the time to read our responses and for the clarification. The overparametrized setting mentioned by the Reviewer is a very interesting problem. When it comes to identifiability, however, the overparameterized setting is in some sense contradictory to identifiability. If the model is over-parameterized, then identifiability does not happen, see for example a related discussion in [Breiding et al., 2023]. In essence, the overparameterized setting corresponds to expressive polynomial neural networks (able to represent any polynomial map) and was first treated in [7]. The optimization landscape of overparametrized shallow (2-layer) PNNs was recently studied in [13].
> > >
> > > The parameter count argument suggested by the Reviewer makes a lot of sense. However, a technical challenge is that to be able to apply it, we must first be able to find the dimension of the neurovariety---in particular, we must prove that the neurovariety is *non-defective* (e.g., its dimension is equal to the expected dimension, that is, the number of parameters *minus* the number of trivial ambiguities). A main contribution of our paper that it allows us to characterize all non-defective cases for deep PNNs (with a minor modification of the localization theorem), and reduce it to finding non-defective cases of 2-layer PNNs. The nondefectivity of 2-layer networks, due to their link with tensor decompositions and Segre-Veronese varieties, are nearly completely studied. Therefore, our paper allows us to nearly completely characterize non-defective deep PNN neurovarieties. We will include a comment to emphasize this consequence of our localization theorem in the revised paper.
> > >
> > > [Breiding et al., 2023] Breiding, P., Gesmundo, F., Michałek, M., and Vannieuwenhoven, N. Algebraic compressed sensing. Applied and Computational Harmonic Analysis, 65, 374-406, 2023.
> > >
> > > [7] J. Kileel et al. On the expressive power of deep polynomial neural networks. Advances in neural information processing systems, 32, 2019.
> > >
> > > [13] Yossi Arjevani, Joan Bruna, Joe Kileel, Elzbieta Polak, and Matthew Trager. Geometry and optimization of shallow polynomial networks. arXiv preprint arXiv:2501.06074, 2025.

---

### Official Review · Reviewer_gdGi · 2025-07-01

**Clarity:** 3
**Significance:** 4
**Originality:** 3
**Rating:** 4
**Confidence:** 3

**Summary:**

This paper develops a comprehensive theory of when deep polynomial neural networks (PNNs) are identifiable, i.e., when their weights can be recovered uniquely from their input-output map. The key insight is the localization theorem, which shows that a deep PNN is globally identifiable if and only if every consecutive two-layer block is identifiable. Building on this, the authors derive simple, constructive conditions (involving layer widths and activation degrees) guaranteeing identifiability for a wide range of architectures, including pyramidal nets and encoder-decoder nets. Their bounds on activation degrees improve prior quadratic requirements to linear in layer widths, and they settle several conjectures in recent literature.

**Questions:**

- Could the authors expand on the work’s relevance to practitioners?

**Ethical Concerns:**

["NO or VERY MINOR ethics concerns only"]

**Final Justification:**

The author's response has addressed my concerns, especially on relevance to practitioners.

**Limitations:**

yes

**Quality:**

3

**Strengths And Weaknesses:**

Strengths
- The theoretical contribution is strong. The paper establishes a clear and modular criterion for identifiability of deep PNNs and resolves prior open conjectures on identifiability and neurovariety dimensions. The activation threshold bounds for identifiability are shown to be linear rather than quadratic, improving prior results significantly.
- The proofs are constructive, offering parameter recovery via tensor decomposition algorithms.
- The results are general enough to cover architectures with and with bias, and of different types such as non-increasing width and encoder-decoder networks.

Weaknesses
- The paper does not present numerical experiments or simulations to illustrate the tightness of theoretical bounds or demonstrate recovery of network weights in practice.
- Computing Kruskal ranks and performing CPD uniqueness tests can be computationally intensive for large-scale networks. The paper does not discuss algorithmic costs, heuristics, or approximate checks for use on real-world PNNs.

---

> ### Author Rebuttal · Authors · 2025-07-29
>
> We would like to thank the Reviewer for the careful evaluation of our work and for the valuable suggestions for improvements. In the following, we provide responses and clarifications to the weaknesses or questions raised by the Reviewer.
>
> Q1: **Lack of numerical experiments**: We agree that the paper would benefit from numerical experiments. However, our main goal to clearly present the uniqueness/identifiability result and its proof in a digestible form, which we really hope will be very useful for community. We emphasize, however, that the constructive nature of our proofs make it feasible to develop algorithms based on linear algebra to recover the parameters of an (h)PNN. We will include a discussion on numerical algorithms and how they can be derived from the proposed proofs in the revised version of the manuscript.
>
> Q2: **Computationally intensive to compute Kruskal ranks**: Thank you for this important question. We note that we do not actually need to compute Kruskal ranks to verify several architectures of practical interest. For various architectures, for example, those with sufficiently high degrees, we just need to check whether the weight matrices have collinear rows/columns. Moreover, in many cases (for high degrees or numbers of neurons not being too large) the corresponding tensor representation of a PNN can be decomposed using efficient large-scale algorithms based on linear algebra, such as the one in [Batselier et al., 2016]. This is a topic we plan to study in a followup work. We will include a discussion about the development of computationally efficient algorithms in the revised version of the paper.
>
> [Batselier et al., 2016] K. Batselier et al. Symmetric tensor decomposition by an iterative eigendecomposition algorithm. Journal of Computational and Applied Mathematics, 308, 69-82. (2016)
>
> Q3: **Expand on the relevance to practitioners**: We thank the Reviewer for this interesting question. As we already mentioned in the answer to Question 1 of Reviewer Mfr2, interpretability of neural networks is tightly linked to uniqueness of its parameters.
> Our work could be also a guide for the design of pruning strategies, aiming at building identifiable architectures from a non-identifiable given one while preserving its original functional response. This work can be an useful to guide for compression.
>
> Furthermore, as also mentioned in the answer to Question 1 from Reviewer Mfr2, identifiability plays an important role in model comparison, "stitching" or averaging. This has been used in several recent works [34], [Ito et al., 2025], [Ainsworth et al., 2022], where one finds a permutation (and possibly rescaling) that best aligns the hidden representations of different pretrained models to perform model comparison, averaging, uncertainty quantification, among other tasks. For this procedure to yield meaningful results, identification is essential.
>
> Finally, identifiability/uniqueness also has direct implications on the geometry of PNNs, since it is directly linked to the dimension of its associated neurovariety as discussed in Appendix B of our paper.
>
> We will emphasize the role of identification in the revised manuscript, as well as the influence of the equivalence class of parameters.
>
> [34] C. Godfrey et al. On the symmetries of deep-learning models and their internal representations. Advances in Neural Information Processing Systems, 35:11893–11905, 2022
>
> [Ito et al., 2025] A. Ito et al. (2025). Linear Mode Connectivity between Multiple Models modulo Permutation Symmetries. In Forty-second International Conference on Machine Learning.
>
> [Ainsworth et al., 2022] S. K. Ainsworth et al. . Git re-basin: Merging models modulo permutation symmetries. The Eleventh International Conference on Learning Representations. (2022)

---

> > ### Comment · Reviewer_gdGi · 2025-08-05
> >
> > The author's response has addressed my concerns. I appreciate the detailed explanations. I would like to maintain my positive rating.

---

### Official Review · Reviewer_vryW · 2025-07-02

**Clarity:** 4
**Significance:** 3
**Originality:** 2
**Rating:** 5
**Confidence:** 4

**Summary:**

This paper investigates the identifiability of deep Polynomial Neural Networks (PNNs). Identifiability of a network means the ability of identify the network parameters from its functional input-outputs.
The central is a two-fold "localization theorem" which proves that the global identifiability of a deep PNN is equivalent to the identifiability of all its constituent 2-layer sub-networks. The proof is relatively elementary and easy to follow, nevertheless elegant and clean. Leveraging this result, the authors connect the of identifiability for a two layer network to the uniqueness of partially symmetric tensor decompositions. This framework allows them to derive new, explicit conditions under which specific deep PNN architectures, such as pyramidal and bottleneck networks, are generically identifiable. The paper also provides improved, linear bounds on the activation degree thresholds required for identifiability and extends its analysis to PNNs with bias terms via a homogenization technique.

**Questions:**

- From what I understand, all the proofs of inedibility rely on this "general" configuration weights, that are everywhere except on some zero measure subset. While this is very true for initialization, I wonder if network that has been trained will remain in this general configuration or not?
- Do authors have any ideas to generalize their results to new activations that are not homogenous, namely GeLU, tanh, etc? Or the proof method here completely breaks down?
- While the algebra shown here is quite interesting, the polynomial basis is really not a good abstraction for real activations. But something like Hermite polynomials are in fact a natural basis there. Is it conceivable to extend these proof methods to some other type of basis rather than polynomials?
- While the issue of identifiability is very interesting, another topic of high interesting to ML audience would be the complexity aspect. meaning, even if we know a network is identifiable, can we find it (reconstruct it) in a reasonable time? Can we turn one of these i Another aspect would be, the  tightly related topic is also the sample complexity? As in, how many samples do we need to identify a network?  Can authors comment on this?
- Could the authors please comment on the relationship between their proof strategy and Fefferman's? Is there any link at all or not?

**Ethical Concerns:**

["NO or VERY MINOR ethics concerns only"]

**Final Justification:**

Both based on the original submission and throughout the discussion, I maintain my highly favorable view (5:accept, unchanged from the original rating) of this submission and consider it to be a valuable contribution.

As stated in my original review, I only raised some points regarding references and contextualizing the work, plus  some questions that were more to clarify the work and some novelty aspects of the work. Several of the discussion points, including those in response to my review, deem to be interesting points worth being integrated into the work, which I presume the authors will do if accepted.

**Limitations:**

yes

**Paper Formatting Concerns:**

no issues

**Quality:**

3

**Strengths And Weaknesses:**

# strengths

- The paper provides a rigorous and comprehensive analysis of deep PNN identifiability. The specific conditions derived for pyramidal and bottleneck networks (Corollaries 18 and 19) are novel and valuable to researchers in this specific domain.

- The work goes beyond mere existence proofs. It provides explicit, checkable inequalities (e.g., Proposition 13) that link architecture choices (layer widths, activation degrees) to the identifiability property. The connection to tensor decomposition is well-executed and powerful.  I found the link between identifiability and uniqueness of tensor decomposition highly interesting and novel (although I am not sure if it is generally novel).

- In general settings, The paper provides an improvement on the activation degree thresholds required for identifiability, showing a linear dependence on layer width rather than the previously conjectured quadratic one (Corollary 22). This is a clear, measurable advance over very recent work in the field.

- I found paper to be quite well-written and logically structured. The definitions are clear, and the arguments, while technical, are laid out in a way that is followable by a reader.

# Weaknesses
Two of the key weaknesses have to do with contextualization and background of this work:
- A similar argument to the localization theorem is present in "Reconstructing a neural net from its output" by Fefferman (1994). Fefferman established an analogous recursive, layer-by-layer proof strategy for the identifiability of deep sigmoidal networks. While the mathematical tools are different and the setup (sigmoid vs polynomial activations), the core conceptual principle of reducing a deep problem to a sequence of shallow ones is not new. The failure to cite and discuss this foundational work that predates this work by 3 decades, seems to be a major oversight.
- The "homogeneization procedure" for handling bias terms is presented as a novel approach. Namely in lines 56-57: "Moreover, we also address the case of PNNs with biases (which was overlooked in previous theoretical studies)." In fact, this method is a standard, textbook technique known as using homogeneous coordinates, widely applied across computer science and machine learning books for decades. The novelty lies in proving it preserves identifiability for PNNs, not in inventing the method itself. This distinction is important to make and the current phrasing is somewhat misleading.
- Limited Scope of Application: The results are specific to PNNs with monomial activations. While this is an important theoretical model, its direct applicability to the broader landscape of modern neural networks (which primarily use ReLU, GeLU, etc.) is limited. Given that earlier works, such as Fefferman's work from 30 years ago on sigmoidal activation, there ought to be more general ways of identifiability that go beyond polynomial networks.

---

> ### Author Rebuttal · Authors · 2025-07-29
>
> We would like to thank the Reviewer for the careful evaluation of our work and for the valuable suggestions for improvements. In the following, we provide responses and clarifications to the weaknesses or questions raised by the Reviewer.
>
> Q1: **Relationship to the argument present in Fefferman (1994)**: We agree with the Reviewer that the property proved by Fefferman is similar to our localization result for polynomial networks. We want nonetheless to emphasize the difference between the objects in consideration. Fefferman's paper indeed focuses on the case of a very specific analytic function (the tanh activation), which is far from being polynomial, and makes use of clever analytic techniques (study of singularities), which only work, though, for activation functions with specific properties (real analytic, finite limit at infinity, meromorphic extension to $\mathbb C$). Also note that Fefferman's arguments would not work in our situation, since monomial activations do not fulfill his requirements (having finite limit at infinity in one direction of the complex plane). In contrast, our proof is based on purely algebraic arguments, well suited to tackle the monomial case. We would also like to add that an exciting future research direction is to tackle the analytic activation case using general (non-monomial) polynomial activations. This would indeed encompass Fefferman's result, and our results on monomial polynomial activations constitute a crucial first step in this analysis.
>
> We also would like to emphasize that the very powerful Fefferman's result provides only a sufficient condition for identifiability. Theorem 9 in Fefferman's work [23] excludes a lot of parameters (though a discrete set) and is not very practical. In our case, not only the proofs are constructive, but also they provide if and only if condition for uniqueness, provided the degree of the activation is high enough (no collinear rows in weight matrices and no zero columns).
>
> We would like to thank the reviewer for mentioning this important paper. Indeed, although Fefferman's paper was already cited in the introduction of the original paper (reference [23]), there is not enough credit given to it, and we will correct this in the revised version.
>
> [23] C. Fefferman. Reconstructing a neural net from its output. Revista Matemática Iberoamericana, 10(3):507–555, 1994.
>
>
> Q2: **Novelty of the homogenization procedure**: We agree with the Reviewer that homogeneization itself is a fairly standard technique in machine learning and mathematics. We highlight, as the Reviewer correctly notes, that the contribution in our paper lies in leveraging it to demonstrate the uniqueness of the resulting deep PNN architecture by reducing it to a homogeneous PNN. The challenge comes from the fact that neither the weight matrices nor the input signals and hidden activations can be assumed to be distributed according to an absolutely continuous measure, which precludes the use of various standard arguments. This difficulty brought on considerable work to tackle NNs with biases even in the case of 2-layer NNs (see for example [68]). Nonetheless, we appreciate the Reviewer's remark that the current phrasing be potentially misleading in the original paper, and we will therefore revise it in the final manuscript to make this distinction clear to the reader.
>
> [68] P. Awasthi et al. Efficient algorithms for learning depth-2 neural networks with general ReLU activations. Adv. Neur. Inf. Proc. Syst., 34:13485–13496, 2021.
>
>
> Q3: **Limited scope of application**: We note that considering PNNs with monomial activations in fact is a crucial first step in tackling general polynomial activations -- which in turn can approximate all classical activation functions over compact sets. This is a direction we will investigate for a follow up work. For more details, see also the answer to Q5 just below, and also the answer to Q2 by Reviewer Ux2R.
>
>
> Q4: **Reliance on a general configuration of weights**: We highlight that our paper include both uniqueness (fixed architecture) and identifiability (generic uniqueness) results. The  conditions for uniqueness of a given architecture are in fact reasonably weak in general: we only require that (for high enough degrees) the weight matrices (with weaker conditions for the first and last layers) have no collinear (or, in the more pathological case, zero) rows or columns, otherwise the network would have redundant neurons that could be suppressed without changing its response. The generic identifiability conditions require the matrices to be sampled from any distribution that is absolutely continuous with respect to the Lebesgue measure. This requirement is fairly weak, and serves to exclude weights that fall in such pathological cases (e.g., some weight matrices with zero/collinear rows/columns). The conditions of uniqueness at fixed architecture are in fact helpful to diagnose the resulting trained network: if it can accurately represent the training data, and yet its weight matrices do not satisfy simple conditions such as having no collinear columns, then this means that the considered architecture could be potentially reduced to a smaller one. If we properly train a PNN to approximate data representable by an identifiable PNN architecture, then the weights should not converge networks with such "degenerate" weight matrices because otherwise the trained network would not be able to represent the data coming from the original identifiable one. These reasons lead us to believe that these conditions are indeed reasonably mild. We will include a more detailed discussion of the implication of these assumptions in the revised version of the paper.
>
> Q5: **Generalizing the results to non-homogeneous activations (GeLU, tanh, etc)**: On the one hand, our proof techniques are considerably different than those of previous works focused on tanh or sigmoidal activations; thus, directly generalizing them for such activation functions appears to be highly nontrivial. On the other hand, considering monomial activation functions is a necessary and crucial step to tackle the PNNs with general polynomial activations. This general polynomial case, which is of significant practical interest, can be potentially tackled with an extension of our proof techniques, where our identifiability results for the case of monomial activations should be an essential part of the analysis. See also the answer to Q2 by Reviewer Ux2R.
>
> Q6: **Extension to Hermite polynomials or other bases**: This is a very interesting question. This is related to the study of the identifiability of PNNs whose activation functions are general polynomials. As mentioned in the previous response, our results on monomial activations serve as a crucial first step to tackle the uniqueness of PNNs with general polynomial activations, which can encompass Hermite polynomial bases. This is a topic we are interested in pursuing in a future work. We will clarify this perspective in the revised version of our paper.
>
> Q7: **Complexity aspects**: Thank you for this insightful question. An important and distinctive aspect of our proofs is that they are constructive, and the parameters of a PNN can be computed from the canonical polyadic decomposition of its associated tensor. While the order of this tensor can be potentially high, under some conditions there exist very efficient algorithms based on linear algebra to compute the factor matrices (see, for example, [Batselier et al., 2016]). While the question of sample complexity is also of great practical interest, our analysis concerns identifiability in the noiseless setting, which is a first stepping stone to study the finite sample case. We find these questions very exciting and plan to tackle algorithmic and sample complexity issues in upcoming work, for instance, following an approach similar to the one used in [Oymak et al., 2021], which leverage the calculation of moment tensors from empirical data.
>
> [Batselier et al., 2016] K. Batselier et al. Symmetric tensor decomposition by an iterative eigendecomposition algorithm. Journal of Computational and Applied Mathematics, 308, 69-82. (2016)
>
> [Oymak et al., 2021] S. Oymak et al. Learning a deep convolutional neural network via tensor decomposition. Information and Inference: A Journal of the IMA, 10(3), 1031-1071 (2021).
>
>
> Q8: **Link between our proof and the one in Fefferman (1994)**: Please see the response to Q1 just above.

---

> ### Comment · Reviewer_vryW · 2025-08-01
>
> From my reading of the paper, I had the strong impression that there is a very fundamental limitation of applying the ideas in present paper to an activation that is not homogenous. As noted in your response here, even adding a bias term was quite challenging. Is my initial understanding that extension of current results to non-homogeneous PNN's will be a major leap, and presumably would require substantial new technical  innovations? Or do you believe such a step is a natural next step? are authors claiming that the present results on homogeneous PNNs can be naturally extended for identifiability of a general PNNs?

---

> > ### Author Response · Authors · 2025-08-02
> >
> > The reviewer is right that this is a major leap, however, what we mean is that contrary to the study of general analytic functions, which might require completely different proof techniques (for instance, Fefferman’s proof relies on the study of the properties of singularities of analytic functions, which has no clear connection to our technical tools), we believe that such an extension can still be pursued within the tensor/algebro-geometric framework used in our paper, see for example the very recent paper [Shahverdi et al., 2025] that managed to tackle the case of activations being sum-of-monomials (which is not homogeneous) using the general framework of algebraic-geometric tools that was employed in the study of the monomial case, in particular, the results of [Finkel et al, 2025], which are superseded by our paper. Thus, while it would indeed be a major leap and require a significant amount of work, we believe such an extension to be feasible without resorting to completely different technical tools, and as such we believe that our results tackling the monomial case can be a crucial first step in this direction.

---

> > > ### Comment · Reviewer_vryW · 2025-08-03
> > >
> > > Can authors comment on the originality of the proof technique that is used in proving the identifiability of a two-layer network with tensor decomposition and invoking theorems? As noted earlier, I found this connection quite fascinating, but I am not sure if this is the first time such a connection is employed or not.

---

> > > > ### Author Response · Authors · 2025-08-04
> > > >
> > > > We would like to thank the Reviewer for taking the time to read our responses and for the clarifications. The uniqueness of homogeneous PNNs in the 2-layer (shallow) case is already known to specialists in the field, with the links between homogeneous PNNs and tensor decompositions being established in [7]. However, [7] was interested in expressivity--a complementary question to identifiability--and we did not find a criteria such as our Proposition 38 (and its generic case, Proposition 13), which give bounds on the degrees for uniqueness/identifiability to hold.
> > > >
> > > > The conditions in these propositions are derived from the uniqueness of the partially symmetric tensor $[[W_2,W_1,W_1,\dots,W_1]]$, which were obtained using a reshaping argument to reduce it to a case of an order-3 tensor for which the classical Kruskal’s criterial could be applied. From such conditions we can obtain a bound on the minimal degree for uniqueness of 2-layer hPNNs using a lower bound on the Kruskal rank of a Khatri-Rao power. Both the reshaping argument and this lower bound are known in the tensor literature (see [Chiantini et al., 2017, Section 4.2] and [77, Corollary 1.18]), but we have not seen them used in the study of neural networks. By leveraging these arguents we obtained a proof that is constructive and gives useful insight for the development of algorithms.
> > > >
> > > > However, we emphasize that Proposition 38 (resp. Proposition 13) are supporting results: a highlight of our localization theorem lies in providing an *if and only if* condition such that *any* uniqueness (resp. identifiability) result for 2-layer PNNs can be used to establish the uniqueness (resp. identifiability) of deep PNNs. Thus, while Kruskal-based criteria are simple and elegant, more powerful uniqueness/identifiability results for the 2-layer case will immediately give us more powerful results for deep PNNs (see Remark 17).
> > > >
> > > > [Chiantini et al., 2017] Chiantini, L., Ottaviani, G., and Vannieuwenhoven, N. Effective criteria for specific identifiability of tensors and forms. SIAM Journal on Matrix Analysis and Applications, 38(2), 656-681, 2017.
> > > >
> > > > [77] Ignat Domanov and Lieven De Lathauwer. On the uniqueness of the canonical polyadic decomposition of third-order tensors—Part II: Uniqueness of the overall decomposition. SIAM Journal on Matrix Analysis and Applications, 34(3):876–903, 2013

---

### Official Review · Reviewer_Mfr2 · 2025-07-03

**Clarity:** 3
**Significance:** 3
**Originality:** 3
**Rating:** 5
**Confidence:** 2

**Summary:**

The paper considers identifiability of the parameters of deep polynomial neural networks. Identification is an important, and often less studied, property of NNs as it is directly linked to the potential to interpret the NNs weights. The authors show that for an equivalence class up to permutations, deep PNN are identified if and only if each 2-layer subnetwork is also identified, extending existing results for 2-layer networks. The paper's main result follows from (1) a rank condition to ensure identifiability in the 2-layer case and (2) an induction argument to extend it to the L layer case.  The authors show how their results can be applied to a variety of structures and given conditions to check whether a PNN structure can be identified.

**Questions:**

Some smaller questions beyond the two questions in the weakness section:

1. In definition 8 why do we additionally introduce the requirement that the parameters cannot be chosen on sets of measure zero? Is this also a requirement for definition 6 and is the set of parameters in the equivalence class defined by Lemma 4 measure zero?

2. Is Proposition 13 new? I would be helpful to relate it to previous work (e.g. [7]) and to the recurrent example and how it relates to the full rank condition of the weight matrices.


* Some typos: for eg. Lemma 32 *of with* architecture

**Ethical Concerns:**

["NO or VERY MINOR ethics concerns only"]

**Final Justification:**

see comment.

**Limitations:**

yes

**Paper Formatting Concerns:**

no concerns

**Quality:**

3

**Strengths And Weaknesses:**

Strengths:

1. The topic of identification of general L-layer deep PNN is important as it is a less studied topic, but crucial to understand interpretability of NNs.

2. The main result that 2-layer uniqueness can be extended to L-layer uniqueness appears correct, slightly surprising at first sight, and potentially a great insight.

3. The paper's review of how the identification results may be used to check different architectures may be potentially useful for researchers in choosing how to construct PNNs.

Weaknesses:
1. A bit more discussion on how interpretability relates to identification, given that the result is up to permutation matrices would help make the paper more relevant. How large is the equivalence class? Are there other equivalence classes for which a similar result could have been proven?
2. The paper presents many results and it is often slightly unclear (beyond Theorem 11) what is their contribution versus building on existing work. Furthermore, what is the relevance of the polynomial activations for the arguments? It seems that the results and proof methods would work generically for different activations, and if such, do similar results exist for NNs with other activations. More discussion on this would be appreciated.

---

> ### Author Rebuttal · Authors · 2025-07-29
>
> We would like to thank the Reviewer for the careful evaluation of our work and for the valuable suggestions for improvements. In the following, we provide responses and clarifications to the weaknesses or questions raised by the Reviewer.
>
> Q1: **How interpretability relates to identification and size of the equivalence class**:  Thank you for the insightful question. In fact, the equivalence class of the parameters of a neural network (i.e., a set of parameter ambiguities which will lead to a representation of the same function) depends  on choice of the activation function. In a more general setting, this has been elegantly formulated through the notion of "intertwiner groups" studied in [34], which characterizes classes of equivalent parameters for various activations.
>
> For the case of polynomial neural networks with monomial activation, the equivalence class  described in Lemma 4 consists of permutations of neurons and rescaling of the input and the output of the activation function. If a PNN is unique/identifiable, then this equivalence class describes all possible representations of the same network with the same number of neurons. For the case of the polynomial activation function of general shape (not just a monomial), the equivalence class reduces to the set of permutations of neurons (which is unavoidable in any MLP), and no rescaling is possible [Shahverdi et al., 2025].
>
> From a practical perspective, if we assume the weight matrices to be properly normalized to fix the scaling ambiguity, identification is linked to interpretability in the sense that for an identifiable network, any alternative representation will be a permuted version of the original one, whereas for any non-identifiable model the parameters and representations could be completely different. Moreover, identifiability plays an important role in model comparison, "stitching" or averaging, which has been used in several recent works [34], [Ito et al., 2025], [Ainsworth et al., 2022]. In such works, one finds a permutation (and possibly rescaling) that best aligns the hidden representations of different pretrained models in order to properly perform model comparison, averaging, uncertainty quantification, among other tasks. For this procedure to yield meaningful results, identification is essential.
>
> We will emphasize the role of identification in the revised manuscript, as well as the influence of the equivalence class of parameters.
>
> [Shahverdi et al., 2025] Shahverdi, V., Marchetti, G. L., and Kohn, K. (2025). Learning on a Razor's Edge: the Singularity Bias of Polynomial Neural Networks. arXiv preprint arXiv:2505.11846.
>
> [34] Charles Godfrey, Davis Brown, Tegan Emerson, and Henry Kvinge. On the symmetries of deep-.-
> learning models and their internal representations. Advances in Neural Information Processing
> Systems, 35:11893–11905, 2022
>
> [Ito et al., 2025] Ito, A., Yamada, M., and Kumagai, A. (2025) Linear Mode Connectivity between Multiple Models modulo Permutation Symmetries. In Forty-second International Conference on Machine Learning.
>
> [Ainsworth et al., 2022] Ainsworth, S. K., Hayase, J., and Srinivasa, S. (2022). Git re-basin: Merging models modulo permutation symmetries. The Eleventh International Conference on Learning Representations.
>
>
> Q2: **Novelty of the results vs. building on prior work, existing results for other activation functions**:
> We thank the Reviewer for this important question. The main novelty of the paper is on two key results, namely, Theorem 11 (our localization theorem) about the uniqueness of deep PNNs, and Corollary 12 on their identifiability, as well as the key results which are used in their proofs. The uniqueness of homogeneous PNNs in the 2-layer case is already known in the literature, but, to the best of our knowledge, there exists no previous localization results that allowed one to leverage them in the study of deep PNNs. In particular, all results on the uniqueness of deep homogeneous PNNs and (shallow and deep) PNNs with biases are completely novel.
>
> Regarding the relevance of polynomial activations, the main advantage is that this is the case of deep NNs with nonlinear activations that are amenable to analysis with powerful methods of algebraic geometry. This is a very active research direction, as witnessed by a spark of recent results on geometry of the related neuromanifolds, expressivity of the networks, optimization landscape and properties of critical points [37]. For our paper, polynomial (monomial) activations are crucial for key steps in the proofs. Moreover, these techniques can serve as a key first step in the study of general polynomial activations, see the recent preprint that appeared after submission of this article [Shahverdi et al., 2025]. We also believe that an extension to other activation functions (for example, analytic) is feasible, for example, by using polynomial approximations, but this would require additional work and major changes in the proof techniques.
>
> Finally, we highlight that some identifiability results for deep networks also exist for other architectures, for example, for networks with ReLU [29,30] or tanh [23] activation functions. However, the proof techniques used in these works are completely different, and do not apply to the case of polynomial activation functions. Moreover, an important characteristic of our proofs is that they are constructive and can be exploited as a first step in the development of algorithms to learn the weight matrices of a PNN. We will also emphasize this distinction in the related work section of the revised manuscript.
>
> [7] Joe Kileel, Matthew Trager, and Joan Bruna. On the expressive power of deep polynomial neural networks. Advances in neural information processing systems, 32, 2019.
>
> [11] Bella Finkel, Jose Israel Rodriguez, Chenxi Wu, and Thomas Yahl. Activation thresholds and expressiveness of polynomial neural networks. arXiv preprint arXiv:2408.04569, 2024.
>
> [23] Charles Fefferman. Reconstructing a neural net from its output. Revista Matemática Iberoamericana, 10(3):507–555, 1994.
>
> [29] Pierre Stock and Rémi Gribonval. An embedding of ReLU networks and an analysis of their identifiability. Constructive Approximation, pages 1–47, 2022.
>
> [30] Joachim Bona-Pellissier, François Malgouyres, and François Bachoc. Local identifiability of deep ReLU neural networks: the theory. Advances in neural information processing systems, 35:27549–27562, 2022.
>
> [37] Giovanni Luca Marchetti, Vahid Shahverdi, Stefano Mereta, Matthew Trager, and Kathlén Kohn.
> An invitation to neuroalgebraic geometry. arXiv preprint arXiv:2501.18915, 2025
>
>
> Q3: **Why identifiability definitions exclude a set of measure zero**: In the definition of identifiability/generic uniqueness (Definition 8), the requirement that for a network to be identifiable uniqueness must hold for all points except possibly for some subset of measure zero serves to eliminate some pathological cases which would appear had the definition allowed the weights to be completely arbitrary. A special case of parameters to exclude is the subnetworks (i.e., networks representable with fewer neurons in at least one layer). This corresponds to weight matrices that have zero rows or columns, or collinear rows. In fact, such networks corresponds to the singular points of the underlying neurovarieties [Shahverdi et al., 2025].
>
> However, we emphasize that our paper does not only treat generic uniqueness. Definition 6 concerns only a single set of parameters, and says whether a specific network representation is unique. The main theorem (Theorem 11) also applies to specific uniqueness, and does not assume genericity of the parameters. We will clarify these definitions and their implications in the revised version of the paper.
>
> Q4: **Relate proposition 13 to results in [7] and rank condition**:  Thank you for this comment and for the suggestion for clarifying the relation between this proposition and previous work and the link to the recurrent example. Indeed, Proposition 13 is related to the identifiability of 2-layer homogeneous PNNs, which was known in the literature (the link between 2-layer homogeneous PNNs and a CP tensor decomposition was already established in [7]). Thus, Proposition 13 (and Proposition 38, corresponding to the case of fixed weight matrices) are direct consequences of the uniqueness of this CP decomposition, which is known in the literature (in fact, stronger conditions are available for identifiability of homogeneous PNNs in the 2-layer case, see Remark 17). We emphasize that the novelty in our paper lies in the constructive proof for the uniqueness and identifiability of deep PNNs (as well as of shallow and deep PNNs with biases), which comes from the uniqueness of the CP decomposition, see the proof of Proposition 38. This is totally novel and, in turn, supports the development of algorithms, since learning deep PNNs can then be performed by computing the CPD of its corresponding tensor. This will be clarified in the revised version of the paper. We will also relate this proposition to the running example, to show how it applies to its corresponding architecture in the revised paper.
>
> Q5: **Typos**: Thank you very much for pointing this out, we will fix this and other typos in the revised paper.

---

> > ### Comment · Reviewer_Mfr2 · 2025-08-05
> >
> > I thank the authors for their insightful responses to my questions. I have raised my score to 5, I am not familiar with the latest research in this subfield so I can not perfectly assess the full extent of the novelty of the results, I would be amenable to raise my score to 6 if other, more knowledgeable, referees agree.

---

### Note · Authors · 2025-08-12

We thank the referees again for their valuable suggestions and for the productive discussions. As final remarks, we would like to emphasize the following points in our answers to the main questions by the referees:

- Novelty of the results vs. building on prior work

The two main results of our paper are Theorem 11 (localization theorem) about the uniqueness of deep PNNs, and Corollary 12 on their identifiability, as well as the key results used in their proofs. A property proved by Feffermann might seem similar to our localization result, but Fefferman's paper focuses on the case of a very specific analytic function (the tanh activation), which is far from being polynomial, and makes use of clever analytic techniques (study of singularities), not working in our context. Thus, we highlight that while some identifiability results for DNNs also exist for other (tanh) architectures, the proof techniques are completely different and do not apply to the case of polynomial activation functions. Moreover, while the uniqueness of hPNNs in the 2-layer case was already known in the literature, we provide a constructive approach (Propositions 38 and 13) to bound the degrees for uniqueness to hold that can be easily turned in a practical algorithm.

- Relevance of our setting based on monomial activations functions

Considering PNNs with monomial activations in fact is a crucial first step in tackling more general polynomial activations. Our monomial setting is a first step in this direction that we managed to analyze with powerful methods from algebraic geometry. We underline that monomial activations are crucial for key steps in the proofs. Moreover, these techniques can serve as a key first step in the study of general polynomial activations (see [Shahverdi et al., 2025]), which can approximate all classical activation functions over compact sets. Thus, we believe an extension of our results to other activation functions (e.g., analytic) is feasible. Nonetheless, it is not straightforward.

- Numerical examples and complexity aspects

We agree that the paper would benefit from numerical examples and complexity study. We emphasize that the constructive nature of our proofs has two practical implications. First, they can be easily turned into practical algorithms based on linear algebra to recover the parameters of a PNN. Second, the complexity analysis can easily be deduced from it since the parameters of a PNN can be computed from the CPD of its associated tensor.

---

### Decision · Program_Chairs · 2025-09-17

**Decision:**

Accept (oral)

**Comment:**

The paper presents a comprehensive theory of identifiability of deep neural networks with (high-degree) monomial activation functions. The central result is a localization theorem (Theorem 11), which establishes that identifiability of this network is equivalent to the identifiability of each consecutive two-layer subnetwork. It establishes constructive conditions based on activation degrees and layer widths, improving previous bounds. The analysis leverages tensor decomposition techniques and extends to networks with bias terms, resolving an open conjecture on neurovariety dimensions. The reviewers do raise concerns over the applicability to modern networks (e.g. with relu, gelu activations) is questionable. Nevertheless, the reviewers still find the result important and hope that follow-up work can be conducted to connect to more applicable networks. I agree with the limited applicability and hope to see follow-up work connectin them with more general activation functions or with the recent results on polynomial representations, i.e. networks without activation functions such as Multilinear Operator Networks or Deep tree tensor networks for image recognition.